# Cross-ancestry genome-wide analysis of atrial fibrillation unveils disease biology and enables cardioembolic risk prediction

Kazuo Miyazawa[1], Kaoru Ito [1]✉, Masamichi Ito[2], Zhaonan Zou [3], Masayuki Kubota[2], Seitaro Nomura [2], Hiroshi Matsunaga[1,2], Satoshi Koyama[1,4,5], Hirotaka Ieki [1,2], Masato Akiyama[6,7], Yoshinao Koike [6,8], Ryo Kurosawa[1], Hiroki Yoshida[1,2], Kouichi Ozaki [1,9], Yoshihiro Onouchi [1,10], BioBank Japan Project*, Atsushi Takahashi[6,11], Koichi Matsuda[12], Yoshinori Murakami [13], Hiroyuki Aburatani [14], Michiaki Kubo[15], Yukihide Momozawa [16], Chikashi Terao [6], Shinya Oki [3], Hiroshi Akazawa[2], Yoichiro Kamatani [6,12]✉ & Issei Komuro [2]✉

Atrial fibrillation (AF) is a common cardiac arrhythmia resulting in increased risk of stroke. Despite highly heritable etiology, our understanding of the genetic architecture of AF remains incomplete. Here we performed a genome-wide association study in the Japanese population comprising 9,826 cases among 150,272 individuals and identified East Asian-specific rare variants associated with AF. A cross-ancestry meta-analysis of >1 million individuals, including 77,690 cases, identified 35 new susceptibility loci. Transcriptome-wide association analysis identified *IL6R* as a putative causal gene, suggesting the involvement of immune responses. Integrative analysis with ChIP-seq data and functional assessment using human induced pluripotent stem cell-derived cardiomyocytes demonstrated ERRg as having a key role in the transcriptional regulation of AF-associated genes. A polygenic risk score derived from the cross-ancestry meta-analysis predicted increased risks of cardiovascular and stroke mortalities and segregated individuals with cardioembolic stroke in undiagnosed AF patients. Our results provide new biological and clinical insights into AF genetics and suggest their potential for clinical applications.

Atrial fibrillation (AF) is the most common cardiac arrhythmia, affecting approximately 46.3 million individuals worldwide[1]. The global prevalence of AF is increasing due to the rapid aging of the general population and intensified search for subclinical AF[2]. Despite progress in diagnostic and therapeutic technologies, a substantial number of patients with AF are admitted with life-threatening complications such as stroke and heart failure[3], causing a considerable burden on patients and public healthcare systems[4]. Besides conventional clinical risk factors such as aging, obesity, hypertension and heart failure, the genetic contribution to the development of AF is also widely recognized. Recent genome-wide association studies (GWASs) have identified more than 100 AF-associated loci, some of which are involved in cardiac developmental, electrophysiological, contractile and structural pathways[5–8]. However, because the vast majority of AF-GWASs have been predominantly performed in European populations, the genetic pathophysiology of AF in non-European

**Fig. 1 | Overview of the study design.** Flowchart of the study, which encompasses the Japanese GWAS with the BBJ first cohort (9,826 AF cases and 140,446 controls), a replication study using the BBJ second cohort (4,602 cases and 44,075 controls), a cross-ancestry meta-analysis with large-scale European GWASs (77,690 cases and 1,167,040 controls in total) and downstream analyses.

populations is not comprehensively understood, and it is difficult to apply polygenic risk scores (PRSs) derived from such GWASs to non-European populations.

Here we sought to explore the genetic architecture of AF in a non-European population and improve the statistical power of AF-GWASs by performing a large-scale Japanese GWAS, followed by a cross-ancestry meta-analysis. Further, we investigated the biological role of the identified AF-associated loci by leveraging gene expression and epigenomic datasets. Additionally, we developed a PRS derived from the cross-ancestry meta-analysis and assessed the impact of the PRS on relevant phenotypes and long-term mortality, which may provide evidence for the clinical utility of AF-PRS and lay the foundation for the realization of precision medicine in AF.

## Results

### Five new risk loci for AF identified in the Japanese GWAS

An overview of the study design is shown in Fig. 1. We performed a GWAS on the case–control dataset from BioBank Japan (BBJ) that comprised 9,826 AF cases and 140,446 controls, using 16,394,105 variants in the autosomes and 423,039 variants in the X chromosome with a minor allele frequency (MAF) > 0.1%. The GWAS identified 31 AF-associated loci with genome-wide significance, of which five were previously unreported (Table 1, Supplementary Table 1, Extended Data Fig. 1 and Supplementary Datasets 1 and 2). The proportion of the variation in AF (the single nucleotide polymorphism (SNP) heritability; $h^2$) explained by the total genome-wide genetic variation detected in the current Japanese GWAS was estimated to be 6.1% (s.e.m. 1.4%), and the liability-scale $h^2$ was estimated at 11.7% (s.e.m. 2.6%) using linkage disequilibrium (LD)-score regression.

To replicate the five newly identified loci, we performed genotyping and association analysis in an independent Japanese cohort including 4,602 cases and 44,075 controls. All of the lead variants were successfully replicated with nominal associations ($P < 0.05$) in the same effect direction (Supplementary Table 2). Among the lead variants in the five new loci, rs202030113 (MAF = 1.2%) and rs778479352 (MAF = 0.25%) were observed only in the East Asian population according to the Genome Aggregation Database v2.1.1 (gnomAD)[9]. rs202030113 is located in the intronic region 3 bp away from an exon-intron boundary of *SYNE1* and is predicted as a splice donor loss with a spliceAI δ score of 0.33 (ref. [10]). A strong signal (odds ratio (OR) for AF development = 2.00, 95% confidence interval (CI) = 1.73–2.31, $P = 1.6 \times 10^{-20}$) was displayed by rs778479352, the lead variant located in an intron of *FGF13*, which is involved in the region of ENCODE[11] accession no. EH38E2771113 (https://screen-v2.wenglab.org/search/?q=EH38E2771113&assembly=GRCh38), where high H3K4me3 and H3K27ac signals were observed, suggesting that rs778479352 might function as a candidate cis-regulatory element.

To identify AF-associated variants independent of the lead variant at each locus, we performed a stepwise conditional analysis, in which 18 independent variants (locus-wide $P < 5.0 \times 10^{-6}$) were additionally detected, increasing the total number of AF-associated signals to 49 (Extended Data Fig. 2 and Supplementary Table 3). We identified ten loci that had multiple independent association signals, especially in the *PITX2-C4orf32* locus with six association signals ($n_{signal} = 2$: *GORAB-PRRX1, CAND2, HAND2-AS1, FANCC, NEBL, AKAP6, ZFHX3* loci, $n_{signal} = 3$: *LINCO2459-TBX5* locus, $n_{signal} = 4$: *NEURL1* locus, and $n_{signal} = 6$: *PITX2-C4orf32* locus). Of these additional signals, three variants were observed only in East Asian populations in gnomAD (rs577463446 at the *FANCC* locus, MAF = 0.5%, OR = 1.58;

## Table 1 | New AF risk loci identified in the Japanese GWAS

| CHR | POS (hg19) | rsID | REF | ALT | AAF | | | β | SE | P value | Nearby gene | Annotated gene[a] | Functional consequence |
|---|---|---|---|---|---|---|---|---|---|---|---|---|---|
| | | | | | BBJ | gnomAD | | | | | | | |
| | | | | | | EUR (non-Finnish) | EUR (Finnish) | | | | | | |
| 6 | 152466619 | rs202030113 | T | C | 0.012 | 0 | 0 | 0.352 | 0.063 | $1.90 \times 10^{-8}$ | *SYNE1* | *SYNE1* | Intronic |
| 12 | 104471663 | rs2930856 | C | T | 0.579 | 0.861 | 0.854 | 0.090 | 0.015 | $5.80 \times 10^{-9}$ | *HCFC2* | *HCFC2* | Intronic |
| 16 | 30619745 | rs1055894680 | C | T | 0.001 | <0.001 | 0.001 | 1.215 | 0.158 | $1.58 \times 10^{-14}$ | *ZNF689* | *ZNF689* | Intronic |
| X | 23399501 | rs73205368 | T | C | 0.284 | 0.045 | 0.015 | 0.089 | 0.012 | $7.50 \times 10^{-13}$ | *PTCHD1* | *PTCHD1* | Intronic |
| X | 137790580 | rs778479352 | T | C | 0.002 | 0 | 0 | 0.692 | 0.075 | $1.62 \times 10^{-20}$ | *FGF13* | *FGF13* | Intronic |

Sentinel variants in new loci with genome-wide significance in the Japanese GWAS (9,826 cases and 140,446 controls). Two-sided *P* values were computed using a logistic regression model. CHR, chromosome; rsID, reference SNP cluster ID; POS, position (hg19); REF, reference allele; ALT, alternate allele; AAF, alternate allele frequency; SE, standard error. [a]The gene annotated by Open Targets.

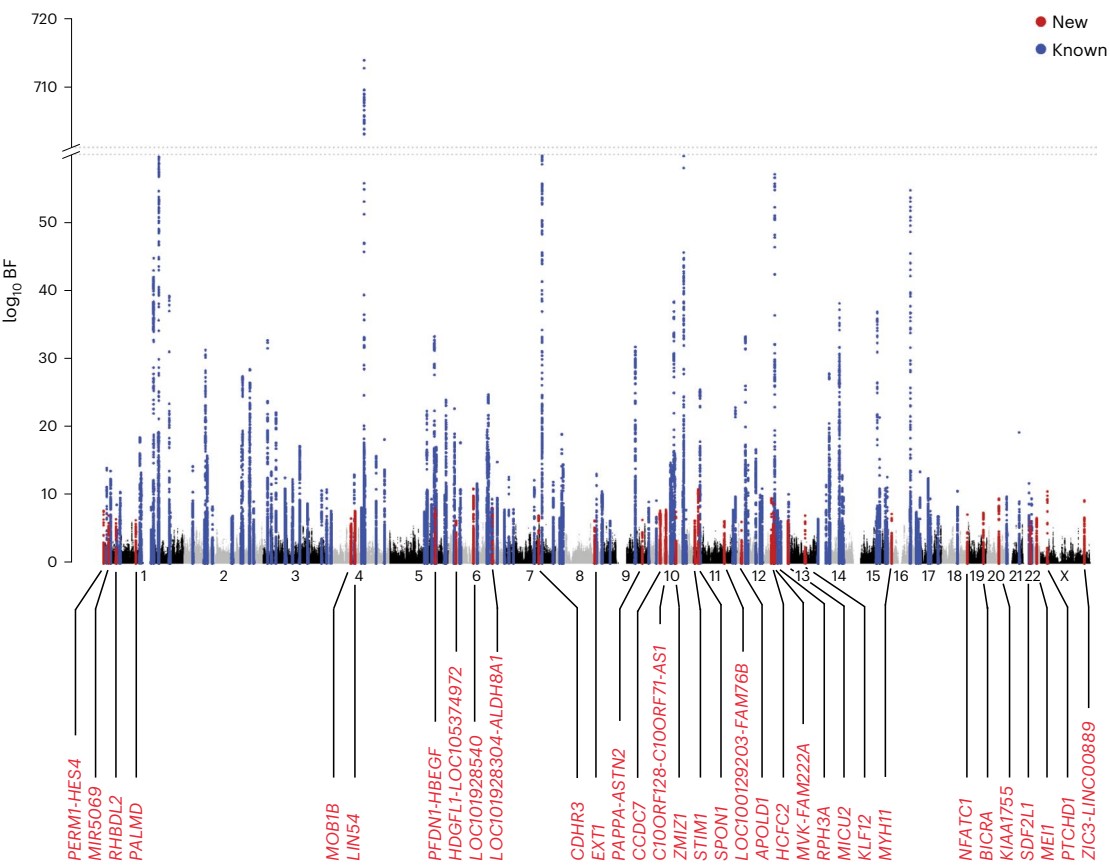

**Fig. 2 | Manhattan plot for the cross-ancestry meta-analysis.** The results of the cross-ancestry meta-analysis (77,690 AF cases and 1,167,040 controls) are shown. The $\log_{10}$ BFs on the *y* axis are shown against the genomic positions (hg19) on the *x* axis. Association signals that reached a genome-wide significance level ($\log_{10}$ BF > 6) are shown in blue if previously reported loci and in red if new loci.

rs965277670 at *NEBL* locus, MAF = 0.6%, OR = 1.65; rs201901902 at *NEURL1* locus MAF = 0.5%, OR = 1.68).

### Cross-ancestry meta-analysis identified 33 new risk loci for AF

To improve the statistical power to detect further genetic associations with AF, we conducted a cross-ancestry meta-analysis by combining the current Japanese GWAS (BBJ) and two European GWASs: a large-scale meta-analysis of European populations (EUR)[7] and biobank data of FinnGen data release 2 (FIN). Together, the three datasets yielded 77,690 cases (BBJ: 9,826, EUR: 60,620 and FIN: 7,244) and 1,167,040 controls (BBJ: 140,446, EUR: 970,216 and FIN: 56,378). We tested a total of 5,158,449 variants with MAF ≥ 1% and identified 150 AF-associated

loci with genome-wide significance ($\log_{10}$ Bayes factor (BF) > 6; Fig. 2, Supplementary Table 4 and Supplementary Datasets 3 and 4). Of these loci, 33 have not been reported previously, including three new loci detected in the current Japanese GWAS. In total, we identified 35 new loci through the current Japanese GWAS and cross-ancestry meta-analysis (Table 2).

Of the 3,637 variants in LD ($r^2 > 0.8$) with 150 lead variants, 19 missense variants were observed (Supplementary Table 5). Among new loci, we found a missense variant, rs848208 (p.Ala970Val), in the *SPEN* gene, encoding a hormone-inducible transcriptional coregulator that activates and represses downstream targets. It was reported that SPEN-deficient zebrafish embryos developed bradycardia,

**Table 2 | New AF risk loci identified in the cross-ancestry meta-analysis**

| CHR | POS (hg19) | REF | ALT | rsID | Nearby gene | Annotated gene[a] | Functional consequence | log₁₀BF | BBJ | | | | EUR | | | | FIN | | | |
|---|---|---|---|---|---|---|---|---|---|---|---|---|---|---|---|---|---|---|---|---|
| | | | | | | | | | AAF | β | SE | P value | AAF | β | SE | P value | AAF | β | SE | P value |
| 1 | 918617 | G | A | rs4970418 | PERM1, HES4 | PLEKHN1 | Intergenic | 7.647 | 0.076 | 0.062 | 0.028 | $2.90\times10^{-2}$ | 0.167 | 0.044 | 0.010 | $7.54\times10^{-6}$ | 0.175 | 0.102 | 0.025 | $4.10\times10^{-5}$ |
| 1 | 16199051 | T | C | rs9782984 | MIR5096 | SPEN | ncRNA intronic | 6.970 | 0.723 | −0.075 | 0.017 | $1.54\times10^{-5}$ | 0.883 | −0.055 | 0.014 | $8.40\times10^{-5}$ | 0.835 | −0.035 | 0.025 | $1.72\times10^{-1}$ |
| 1 | 39385714 | G | A | rs75414548 | RHBDL2 | NDUFS5 | Intronic | 6.439 | 0.060 | 0.068 | 0.032 | $3.49\times10^{-2}$ | 0.077 | 0.068 | 0.015 | $4.35\times10^{-6}$ | 0.059 | 0.106 | 0.040 | $8.18\times10^{-3}$ |
| 1 | 100149308 | G | A | rs1933723 | PALMD | PALMD | Intronic | 6.298 | 0.698 | 0.034 | 0.016 | $3.75\times10^{-2}$ | 0.677 | 0.036 | 0.007 | $5.21\times10^{-7}$ | 0.664 | 0.027 | 0.020 | $1.82\times10^{-1}$ |
| 4 | 71776935 | A | C | rs125125202 | MOB1B | DCK | Intronic | 6.547 | 0.621 | −0.039 | 0.016 | $1.35\times10^{-2}$ | 0.623 | −0.034 | 0.007 | $1.27\times10^{-6}$ | 0.542 | −0.030 | 0.019 | $1.89\times10^{-2}$ |
| 4 | 83910712 | T | G | rs6841049 | LIN54 | LIN54 | Intronic | 7.628 | 0.342 | −0.044 | 0.016 | $5.58\times10^{-3}$ | 0.567 | −0.037 | 0.007 | $6.01\times10^{-8}$ | 0.608 | −0.019 | 0.019 | $3.19\times10^{-1}$ |
| 5 | 139703286 | T | C | rs17118812 | PFDN1, HBEGF | PFDN1 | Intergenic | 8.078 | 0.385 | 0.059 | 0.015 | $1.37\times10^{-4}$ | 0.276 | 0.036 | 0.007 | $1.86\times10^{-6}$ | 0.357 | 0.043 | 0.020 | $3.00\times10^{-2}$ |
| 6 | 22598259 | C | T | rs7664436 | HDGFL1, LOC105374972 | HDGFL1 | Intergenic | 6.187 | 0.229 | 0.040 | 0.018 | $2.19\times10^{-2}$ | 0.282 | 0.031 | 0.007 | $2.04\times10^{-5}$ | 0.222 | 0.071 | 0.023 | $1.62\times10^{-3}$ |
| 6 | 76164589 | C | A | rs12209223 | LOC101928540 | FILIP1 | ncRNA exonic | 10.909 | 0.135 | 0.088 | 0.021 | $4.34\times10^{-5}$ | 0.108 | 0.059 | 0.011 | $7.42\times10^{-8}$ | 0.137 | 0.082 | 0.027 | $2.59\times10^{-3}$ |
| 6 | 135119089 | C | T | rs4896104 | LOC101928304, ALDH8A1 | ALDH8A1 | Intergenic | 7.975 | 0.829 | −0.052 | 0.020 | $8.17\times10^{-3}$ | 0.556 | −0.037 | 0.007 | $7.51\times10^{-8}$ | 0.645 | −0.036 | 0.020 | $7.14\times10^{-2}$ |
| 7 | 105612736 | A | G | rs2727757 | CDHR3 | CDHR3 | Intronic | 6.943 | 0.570 | 0.060 | 0.016 | $1.13\times10^{-4}$ | 0.273 | 0.030 | 0.008 | $5.49\times10^{-5}$ | 0.292 | 0.051 | 0.021 | $1.47\times10^{-2}$ |
| 8 | 118863412 | A | T | rs17430357 | EXT1 | EXT1 | Intronic | 6.245 | 0.230 | 0.027 | 0.018 | $1.20\times10^{-1}$ | 0.180 | 0.040 | 0.009 | $4.85\times10^{-6}$ | 0.160 | 0.082 | 0.026 | $1.35\times10^{-3}$ |
| 9 | 119181794 | G | A | rs17303101 | PAPPA, ASTN2 | TRIM32 | Intergenic | 6.328 | 0.086 | 0.035 | 0.026 | $1.86\times10^{-1}$ | 0.290 | 0.034 | 0.007 | $5.27\times10^{-6}$ | 0.254 | 0.077 | 0.022 | $3.80\times10^{-4}$ |
| 10 | 32772734 | C | T | rs11527634 | CCDC7 | CCDC7 | Intronic | 7.692 | 0.303 | −0.048 | 0.016 | $3.30\times10^{-3}$ | 0.113 | −0.051 | 0.011 | $1.92\times10^{-6}$ | 0.103 | −0.084 | 0.031 | $7.23\times10^{-3}$ |
| 10 | 50485434 | G | A | rs76460895 | C10orf128, C10orf71-AS1 | TMEM273 | Intergenic | 7.810 | 0.096 | −0.083 | 0.026 | $1.43\times10^{-3}$ | 0.053 | −0.066 | 0.015 | $1.25\times10^{-5}$ | 0.057 | −0.138 | 0.041 | $7.16\times10^{-4}$ |
| 10 | 80898969 | G | T | rs1769758 | ZMIZ1 | ZMIZ1 | Intronic | 7.505 | 0.715 | 0.052 | 0.017 | $2.72\times10^{-3}$ | 0.490 | 0.034 | 0.008 | $5.38\times10^{-6}$ | 0.501 | 0.062 | 0.019 | $1.11\times10^{-3}$ |
| 11 | 3890059 | C | T | rs7126870 | STIM1 | STIM1 | Intronic | 6.216 | 0.661 | −0.036 | 0.016 | $2.27\times10^{-2}$ | 0.489 | −0.030 | 0.007 | $5.10\times10^{-6}$ | 0.484 | −0.042 | 0.019 | $2.50\times10^{-2}$ |
| 11 | 14036189 | G | A | rs10500790 | SPON1 | SPON1 | Intronic | 10.831 | 0.341 | 0.062 | 0.016 | $8.40\times10^{-5}$ | 0.376 | 0.035 | 0.007 | $3.45\times10^{-7}$ | 0.361 | 0.081 | 0.020 | $3.61\times10^{-5}$ |
| 11 | 95089882 | C | T | rs517938 | LOC100129203, FAM76B | SESN3 | Intergenic | 6.157 | 0.219 | −0.022 | 0.018 | $2.30\times10^{-1}$ | 0.670 | −0.037 | 0.007 | $1.80\times10^{-7}$ | 0.603 | −0.028 | 0.019 | $1.47\times10^{-1}$ |
| 12 | 12886027 | G | A | rs10845620 | APOLD1 | GPR19 | Intronic | 6.031 | 0.104 | 0.064 | 0.026 | $1.26\times10^{-2}$ | 0.132 | 0.049 | 0.010 | $2.32\times10^{-6}$ | 0.134 | 0.037 | 0.028 | $1.77\times10^{-1}$ |
| 12 | 104492003 | A | G | rs2629755 | HCFC2 | HCFC2 | Intronic | 9.505 | 0.420 | −0.088 | 0.015 | $6.35\times10^{-9}$ | 0.141 | −0.038 | 0.009 | $6.71\times10^{-5}$ | 0.152 | −0.045 | 0.026 | $8.69\times10^{-2}$ |
| 12 | 110082115 | T | C | rs1344543 | MVK, FAM222A | UBE3B | Intergenic | 8.675 | 0.446 | −0.097 | 0.015 | $1.46\times10^{-10}$ | 0.537 | −0.011 | 0.007 | $9.59\times10^{-2}$ | 0.431 | −0.012 | 0.019 | $5.20\times10^{-1}$ |
| 12 | 113196733 | G | A | rs11614295 | RPH3A | RPH3A | Intronic | 8.669 | 0.417 | −0.111 | 0.017 | $9.88\times10^{-11}$ | 0.312 | −0.009 | 0.008 | $2.12\times10^{-1}$ | 0.252 | −0.034 | 0.022 | $1.16\times10^{-1}$ |
| 13 | 22111521 | C | A | rs11841562 | MICU2 | MICU2 | Intronic | 6.240 | 0.324 | 0.026 | 0.016 | $1.11\times10^{-1}$ | 0.404 | 0.030 | 0.007 | $8.82\times10^{-6}$ | 0.388 | 0.064 | 0.019 | $8.57\times10^{-4}$ |
| 13 | 74520186 | T | A | rs1886512 | KLF12 | KLF12 | Intronic | 6.943 | 0.195 | 0.047 | 0.018 | $1.08\times10^{-2}$ | 0.357 | 0.036 | 0.007 | $2.81\times10^{-7}$ | 0.360 | 0.024 | 0.020 | $2.29\times10^{-1}$ |
| 16 | 15902715 | G | A | rs9284324 | MYH11 | MYH11 | Intronic | 7.200 | 0.194 | −0.057 | 0.019 | $3.39\times10^{-3}$ | 0.314 | −0.035 | 0.007 | $1.22\times10^{-6}$ | 0.335 | −0.044 | 0.020 | $2.92\times10^{-2}$ |
| 18 | 77156537 | C | G | rs8096658 | NFATC1 | NFATC1 | Intronic | 7.120 | 0.305 | 0.054 | 0.017 | $1.18\times10^{-3}$ | 0.488 | 0.038 | 0.007 | $1.33\times10^{-7}$ | 0.446 | 0.010 | 0.019 | $6.19\times10^{-1}$ |
| 19 | 48142746 | A | C | rs11881441 | BICRA | NOP53 | Intronic | 7.384 | 0.818 | 0.045 | 0.020 | $2.23\times10^{-2}$ | 0.660 | 0.038 | 0.007 | $5.93\times10^{-7}$ | 0.708 | 0.051 | 0.021 | $1.35\times10^{-2}$ |
| 20 | 36841914 | G | A | rs3746471 | KIAA1755 | KIAA1755 | Exonic | 9.429 | 0.430 | 0.050 | 0.015 | $7.96\times10^{-4}$ | 0.469 | 0.035 | 0.007 | $2.30\times10^{-7}$ | 0.437 | 0.057 | 0.019 | $2.94\times10^{-3}$ |
| 22 | 21999229 | C | G | rs5754508 | SDF2L1 | CCDC116 | Downstream | 6.077 | 0.360 | 0.069 | 0.018 | $1.35\times10^{-4}$ | 0.191 | 0.036 | 0.009 | $1.00\times10^{-4}$ | 0.288 | 0.035 | 0.021 | $9.70\times10^{-2}$ |
| 22 | 42189407 | T | G | rs139557 | MEI1 | MEI1 | Intronic | 6.577 | 0.697 | −0.025 | 0.016 | $1.33\times10^{-1}$ | 0.676 | 0.038 | 0.007 | $1.69\times10^{-7}$ | 0.682 | 0.070 | 0.020 | $5.73\times10^{-4}$ |
| X | 23399501 | T | C | rs73205368 | PTCHD1 | PTCHD1 | Intronic | 10.514 | 0.285 | 0.089 | 0.012 | $7.50\times10^{-13}$ | 0.051 | 0.009 | 0.024 | $7.12\times10^{-1}$ | 0.021 | 0.148 | 0.054 | $5.74\times10^{-3}$ |
| X | 137418967 | C | T | rs1891095 | ZIC3, LINC00889 | ZIC3 | Intergenic | 9.242 | 0.079 | 0.091 | 0.020 | $5.93\times10^{-6}$ | 0.180 | 0.038 | 0.011 | $7.50\times10^{-4}$ | 0.192 | 0.086 | 0.019 | $9.31\times10^{-6}$ |

Sentinel variants in new loci with genome-wide significance in the cross-ancestry meta-analysis (77,690 cases and 1,167,040 controls). Ancestry-adjusted associations (log₁₀ BF) were computed using the MANTRA software. CHR, chromosome; POS, position (hg19); REF, reference allele; ALT, alternate allele; rsID, reference SNP cluster ID; AAF, alternate allele frequency; SE, standard error; ncRNA, noncoding RNA. [a]The gene annotated by Open Targets.

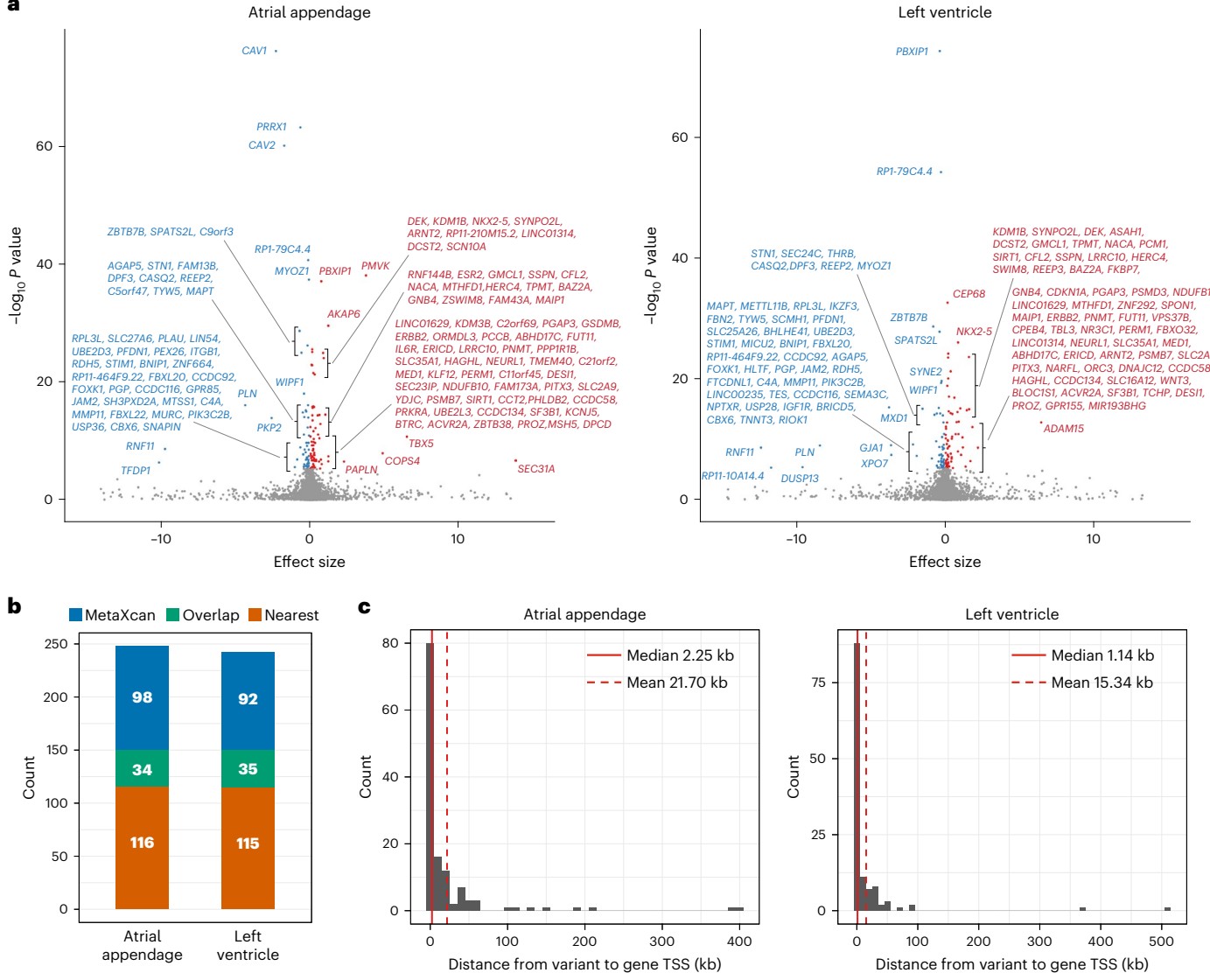

**Fig. 3 | Transcriptome-wide association analysis. a**, Volcano plot showing individual genes with the effect size and the $-\log_{10}P$ value for TWAS based on the atrial appendage (left) and left ventricular tissues (right) from GTEx. Effect sizes and $P$ values were computed using the MetaXcan software to assess the association between predicted gene expression based on the GTEx data and AF. Bonferroni correction was applied to account for the number of genes tested in each tissue ($P < 0.05/10,414$ for atrial appendage and $<0.05/9,702$ for left ventricle). Significant genes are highlighted with red (positive effect of predicted gene expression on AF) and blue (negative effect of predicted gene expression on AF). **b**, Number of genes located closest to the lead variants (red), identified by TWAS (blue), and overlapped between them (green) in atrial appendage and left ventricular tissues. **c**, Distribution of physical distances from the AF-associated variant included in the prediction model to the TSS of the canonical transcript for each candidate gene identified by TWAS in atrial appendage (left) and left ventricular tissues (right).

atrioventricular block and heart chamber fibrillation with downregulation of connexin 43 expression[12], which is a well-known component of gap junctions and is associated with the cardiac conduction system[13]. Another missense variant in a new locus, rs3746471 (p.Arg1045Trp), is located in the *KIAA1755* gene, which is reported to be associated with heart rate[14] and heart rate variability[15]. Given the robust relationship between autonomic nervous dysfunction and AF[16], *KIAA1755* is a potential target gene for neural modulation contributing to AF management; however, the biological association between *KIAA1755* and AF has not been fully examined.

**Prioritization of associated genes and transcription factors**

We performed a transcriptome-wide association study (TWAS) using the identified loci in the cross-ancestry meta-analysis and GTEx data[17] to identify candidate genes associated with AF. Given the enrichment

of AF-associated loci in heart tissue (Supplementary Note and Supplementary Fig. 1), we used gene expression data from GTEx in the atrial appendage and left ventricle as a reference. TWAS prioritized 132 and 127 candidate causal genes substantially associated with AF in the atrial appendage and left ventricle, respectively (Fig. 3a and Supplementary Table 6). Intriguingly, we found that *IL6R* is one of the candidate genes associated with AF in the atrial appendage ($\beta_{IL6R} = 0.221$, $P = 2.147 \times 10^{-9}$). The prediction model of *IL6R* expression included rs10908837 (MAF = 42%, $\log_{10}$ BF = 7.237 in the cross-ancestry meta-analysis), which is located in an intron of *IL6R*. Furthermore, to assess the cis and trans effects of AF-associated variants on the candidate genes, we calculated the physical distances from the variants to the transcription start site (TSS) of the candidate genes. Only 34 and 35 genes overlapped between the nearest genes to the lead variants and the candidate genes identified by TWAS (Fig. 3b), and the median physical

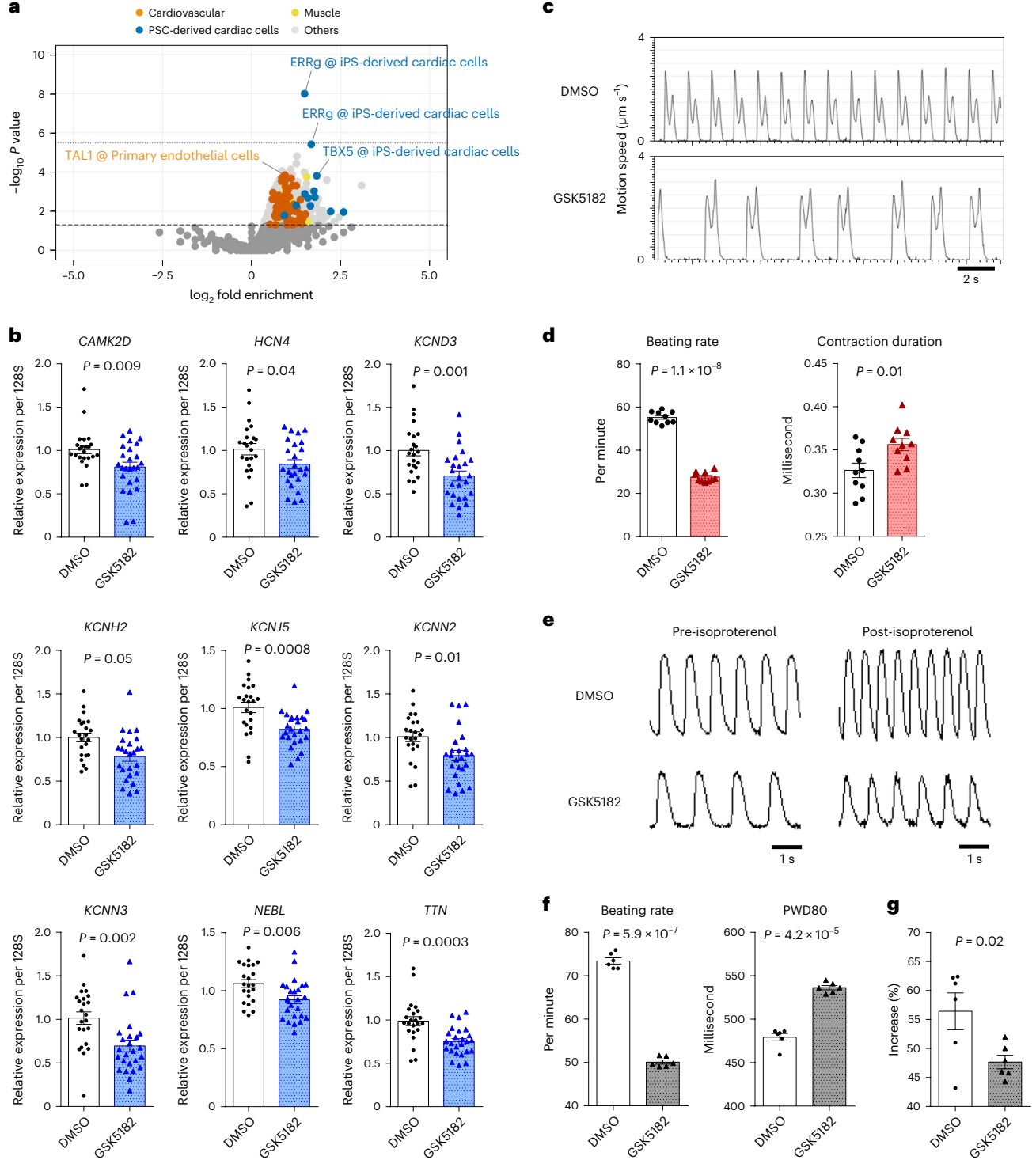

**Fig. 4 | Functional analysis of ERRg using iPSCMs. a**, Volcano plot analysis. Each point represents a ChIP-seq experiment and is highlighted as red for cardiovascular cells, green for induced pluripotent stem cell-derived cardiac cells, blue for muscle cells and gray for other cell types. *P* values were calculated with two-tailed Fisher's exact probability test. The *x* axis shows $\log_2$-transformed fold enrichment of transcription factor in 150 AF-associated loci, compared to 150 randomly selected genomic regions. The *y*-axis shows $\log_{10}$-transformed *P* value for enrichment. The dashed line indicates the significance threshold level of *P* = 0.05, and the dotted line indicates *P* = 0.05/15,109. **b**, Comparison of gene expression changes in iPSCMs with and without GSK5192 (an inverse agonist of ERRg) administration (*n* = 25 and *n* = 23, respectively). Ion channels and sarcomere genes were selected from the downstream genes of ERRg based on the

binding profiles of ChIP-seq data using target genes function in ChIP-Atlas. **c,d**, Motion analysis of iPSCMs using the SI8000 Cell Motion Imaging System. The *y* axis represents the magnitude of cellular motion over time. GSK5182-treated iPSCMs show a decrease in spontaneous beating rate and irregularity (**c**). Beating rate and contraction duration analyzed using the SI8000 Cell Motion Imaging System (*n* = 10) (**d**). **e–g**, Calcium handling analysis of iPSCMs. Calcium transient signal was recorded before and after isoproterenol administration (**e**). We measured averaged peak counts (beating rates) and peak with duration at 80% repolarization (PWD80) (*n* = 6) (**f**). The increases in beating rate were compared before and after isoproterenol administration (*n* = 6) (**g**). In **b**, **d**, **f** and **g**, data are presented as mean ± s.e.m., and *P* values from a two-sided Student's *t* test are shown.

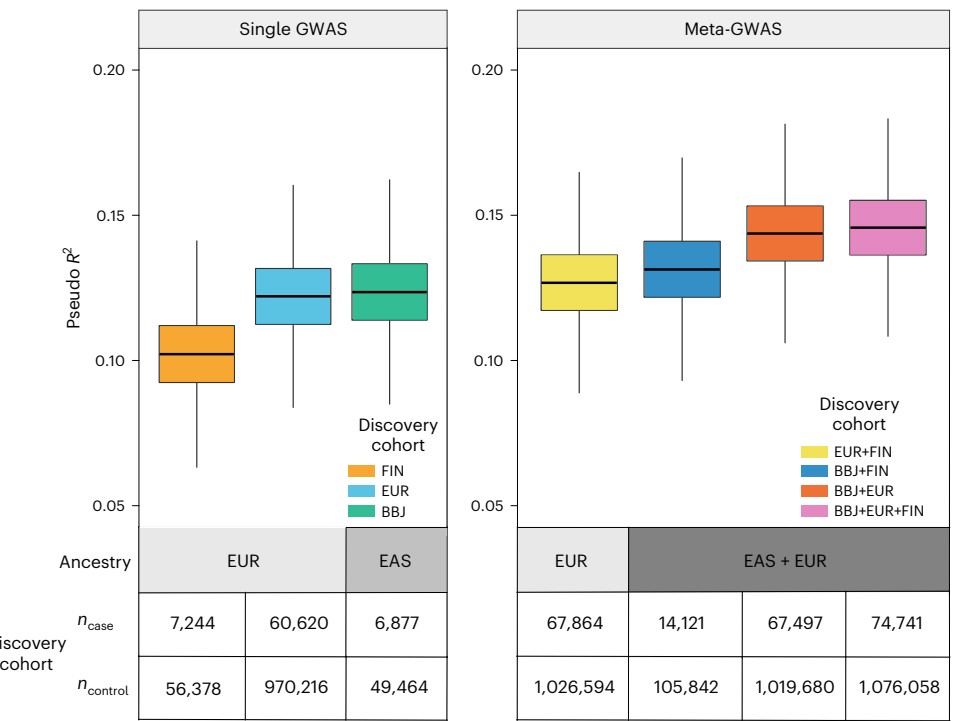

**Fig. 5 | PRS performance (Nagelkerke's pseudo $R^2$) in the Japanese test cohort (2,953 cases and 21,194 controls).** The results of three PRS models derived from a single GWAS are shown in the left panel ($n_{total}$ = 63,622 for FIN; $n_{total}$ = 1,030,836 for EUR; $n_{total}$ = 56,341 for BBJ). The results of four PRS models derived from a meta-GWAS are shown in the right panel ($n_{total}$ = 1,094,458 for EUR + FIN;

$n_{total}$ = 119,963 for BBJ + FIN; $n_{total}$ = 1,087,177 for BBJ + EUR; $n_{total}$ = 1,150,799 for BBJ + EUR + FIN). The distribution of pseudo $R^2$ was estimated from $5 \times 10^4$ times bootstrapping. The box plot center line represents the median, the bounds represent the first and third quartile, and the whiskers reach to 1.5 times the interquartile range.

distances were 2.25 kb and 1.14 kb (Fig. 3c) in the atrial appendage and left ventricle, respectively. This relationship between AF-associated variants and candidate genes is comparable to a previous study in which the distances from the noncoding GWAS signals to the target genes were assessed based on the chromatin state and three-dimensional contacts[18]. Exceptionally, only one gene, *FBN2*, was more than 500 kb away from the variant in the left ventricle (512 kb). This result indicates that, although disease-associated genes are not necessarily closest to the lead variants, most candidate genes are influenced by cis effects of AF-associated variants. Finally, we performed Gene Ontology enrichment analysis using the candidate genes identified by TWAS and found several substantially enriched pathways, such as cardiac developmental, conduction and cardiomyocyte contractile or structure (Extended Data Fig. 3).

Next, we sought to identify transcription factors that bind to AF-associated loci and orchestrate the expression of causative genes involved in AF development. We performed enrichment analysis using the ChIP-Atlas dataset[19], which comprises several high-throughput ChIP-seq experiments (15,109 experiments, 1,028 transcription factors). We found that estrogen-related receptor gamma (ERRg) binding was substantially enriched in AF-associated loci with Bonferroni-corrected significance level of $P = 3.3 \times 10^{-6}$ (0.05/15,109) (Fig. 4a and Supplementary Table 7). Indeed, ERRg ChIP-seq peaks overlapped with AF-associated loci around genes encoding cardiac ion channels (*CAMK2D*, *KCNJ5*, *KCNH2* and *HCN4*), where active histone marks such as H3K27ac and H3K4me3 in induced pluripotent stem cell (iPSC)-derived cardiac cells were also observed (Extended Data Fig. 4). To demonstrate that ERRg is functionally involved in the pathogenesis of AF, we performed a functional analysis of ERRg using human induced pluripotent stem cell-derived cardiomyocytes (iPSCMs). We first evaluated changes in gene expression after administration of an inverse agonist of ERRg, GSK5182 (ref. 20); ion channels and sarcomere

genes were selected from the downstream genes of ERRg based on the binding profiles of ChIP-seq data using Target Genes function in ChIP-Atlas. We found that gene expression was substantially decreased after ERRg administration (Fig. 4b). Furthermore, GSK5182-treated iPSCMs revealed a trend toward decreased spontaneous beating rate and notable irregularity and prolonged contraction duration (Fig. 4c,d). Similarly, the calcium transient duration was also found to be prolonged (Fig. 4e,f), and the increase in beating rate by isoproterenol was attenuated by GSK5182 administration (Fig. 4g). Such changes in beating rate and action potential duration have been reported in iPSCMs derived from patients with AF[21,22]. These results collectively suggest that ERRg is critically involved in the pathogenesis of AF through the regulation of expression of target genes, including ion channels, in cardiomyocytes.

**Performance of PRS derived from cross-ancestry meta-GWAS**

PRS offers potential for risk stratification of complex traits and diseases based on genetic data. However, the transferability of PRS from diverse populations to a population of another ancestry remains challenging. Therefore, we examined the performance of a PRS derived from various combinations of summary statistics in the Japanese population. We split our case–control samples into derivation, validation and test datasets, and constructed 376 combinations of the summary statistics of three GWASs (BBJ, EUR and FIN) with parameters for PRS derivation. Based on the PRS performance in the validation cohort, we determined the parameters that showed the best performance for each combination of summary statistics (BBJ, FIN, EUR, BBJ + FIN, BBJ + EUR, EUR + FIN and BBJ + EUR + FIN) (Supplementary Table 8) and assessed the performance of the best model in the test cohort (Fig. 5, Extended Data Figs. 5 and 6 and Supplementary Table 9). For the PRS derived from a single population GWAS, as concordant with the population specificity, the PRS derived from BBJ

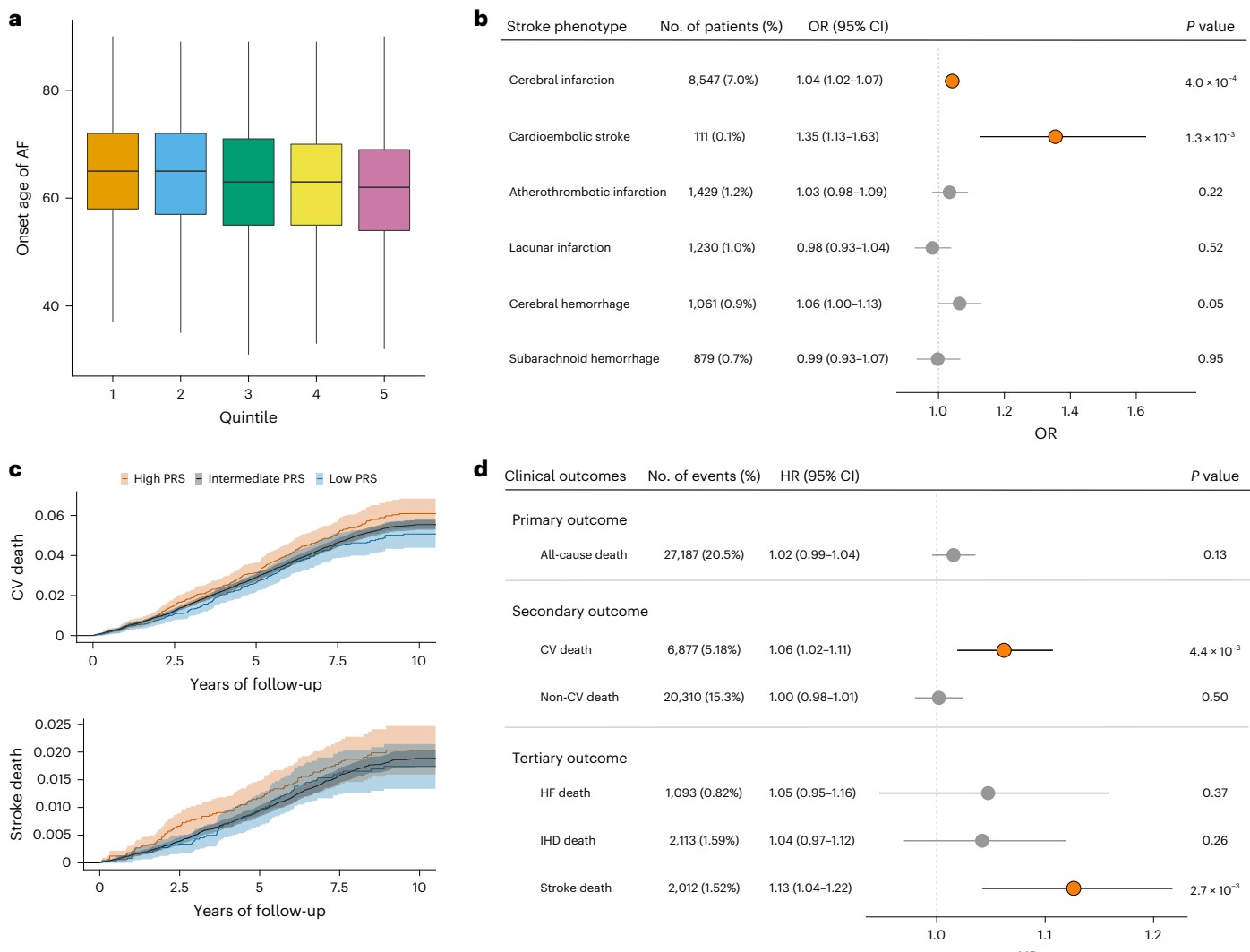

**Fig. 6 | Impact of AF-PRS on relevant phenotypes and long-term mortality.**
**a**, Association between AF-PRS and onset age of AF. The onset age of AF in individuals with data available ($n = 7,458$) is shown based on the AF-PRS quintiles. The number of individuals in each quintile is 1,491 to 1,492. The center line of the box plot represents the median, the bounds represent the first and third quartile, and the whiskers reach to 1.5 times the interquartile range. **b**, Association between AF-PRS and stroke phenotypes in individuals without AF. *P* values were calculated for PRS by logistic regression analysis, and the significance was set at $P = 8.3 \times 10^{-3}$ (0.05/6). Each dot represents an estimate on the OR scale with an error bar indicating the 95% CI for 1 s.d. increase in AF-PRS. Significant and nonsignificant associations are shown in orange and gray, respectively.

**c**, Kaplan–Meier estimates of cumulative events from cardiovascular mortality (upper) and stroke death (lower) are shown with a band of 95% CI. Individuals are classified into high PRS (top 10th percentile, red), low PRS (bottom 10th percentile, blue) and intermediate (others, green). **d**, Effects of AF-PRS on long-term mortality. *P* values were calculated for PRS by Cox proportional hazard analysis and the significance was set at $P = 8.3 \times 10^{-3}$ (0.05/6). Each dot represents an estimated HR with an error bar indicating the 95% CI for a 1 s.d. increase in AF-PRS. Significant and nonsignificant associations are shown in orange and gray, respectively. CV, cardiovascular; HF, congestive heart failure; IHD, ischemic heart disease.

showed higher performance trend than those from EUR (pseudo $R^2 = 0.122$ in EUR versus 0.124 in BBJ, $P = 0.681$) and FIN (pseudo $R^2 = 0.102$ in FIN, $P < 4 \times 10^{-4}$) despite the smaller sample size, although there was no statistically significant difference in the PRS performance between BBJ and EUR. Among the PRS derived from the meta-GWAS, we found significant superiority of the PRS derived from BBJ + EUR compared to those from FIN + EUR (pseudo $R^2 = 0.144$ in BBJ + EUR versus 0.131 in FIN + EUR, $P < 4 \times 10^{-4}$) even though the number of cases was similar. Among all models, the PRS derived from three studies with multi-ancestry and the largest sample size (BBJ + EUR + FIN) showed the highest performance (pseudo $R^2 = 0.146$, 95% CI = 0.115–0.170, area under the curve of receiver operating characteristic = 0.738, 95% CI = 0.726–0.745).

## Impact of AF-PRS on relevant phenotypes and outcomes

To assess the potential of the PRS for clinical applications, we investigated the association between PRS and the onset age of AF in individuals from our BBJ case samples ($n = 7,458$). We observed that the onset age decreased as the PRS increased, and individuals with the top 1% PRS were estimated to be approximately 4 years younger at AF onset compared to the remaining individuals (Fig. 6a and Extended Data Fig. 7a,b). Moreover, we examined whether AF-PRS could explain the phenotypic variability of stroke in individuals without a diagnosis of AF. We performed logistic regression analysis in 121,351 control samples in our dataset, and found significant associations of the PRS with increased risks of cerebral infarction (OR (95% CI) = 1.042 (1.018–1.065), $P = 4.0 \times 10^{-4}$) and cardioembolic stroke (OR (95% CI) = 1.355

(1.126–1.630), $P = 1.3 \times 10^{-3}$) after Bonferroni correction (Fig. 6b). Notably, we observed the largest impact of the PRS on cardioembolic stroke among those with other stroke phenotypes, indicating that AF-PRS may reveal clinically undetectable AF (that is, subclinical AF) or AF-related conditions such as prothrombotic or hypercoagulable state, in individuals without AF.

To further explore the clinical utility of AF-PRS, we assessed the impact of PRS on mortality using long-term follow-up data in BBJ. The Kaplan–Meier estimates of cumulative mortality rate were increased in individuals with a high PRS, especially in cardiovascular- and stroke-related mortality (Fig. 6c,d and Extended Data Fig. 8). Moreover, we performed Cox regression analysis, and as shown in Fig. 6d, no significant association between AF-PRS and all-cause death was found, but a trend was observed (hazard ratio (HR) per 1 s.d. of PRS = 1.02, 95% CI = 0.99–1.04, $P = 0.13$). The secondary outcome indicates that this trend was highly specific to cardiovascular death, which was substantially associated with AF-PRS (HR (95% CI) = 1.06 (1.02–1.11), $P = 4.4 \times 10^{-3}$ for cardiovascular disease; HR (95% CI) = 1.00 (0.98–1.01), $P = 0.50$ for noncardiovascular disease). Furthermore, the tertiary outcome suggests stroke death as a leading factor that impacts the association between AF-PRS and cardiovascular deaths (HR (95% CI) = 1.13 (1.04–1.22), $P = 2.7 \times 10^{-3}$). In contrast to evidence from clinical studies, the association between AF-PRS and heart failure death did not reach statistical significance in the present study (HR (95% CI) = 1.05 (0.95–1.16)) $P = 0.37$). Among 132,737 individuals for whom mortality data were available, the number of events for heart failure death was 1,093 (0.82%), which was approximately half of stroke events ($n = 2,012$) and even less than 20% of cardiovascular events ($n = 6,877$). Thus, it was assumed that the standard deviation for heart failure death was larger due to the smaller number of events in our cohort, which resulted in a relatively wide CI that might make it difficult to reach statistical significance.

### Cross-trait genetic liability of AF

AF is frequently concomitant with various cardiovascular diseases, such as valvular heart disease, heart failure and stroke. These cardiovascular diseases, including AF, partially share the underlying pathophysiology and are mutually associated with the development of each other, whereas the causality between AF and cardiovascular diseases is not comprehensively elucidated. Therefore, we estimated the causal effect of AF on a wide range of cardiovascular diseases using two-sample Mendelian randomization (MR), where the exposure was AF and all the distinct AF-associated variants from the cross-ancestry meta-analysis were used as instrumental variables. Consistent with the clinical evidence, we observed significant genetic liability of AF to the development of several cardiovascular diseases such as heart failure, cardiomyopathy, stroke and transient ischemic attack (Extended Data Fig. 9a and Supplementary Table 10). Additionally, we found the causal effect of AF on valvular disease (OR (95% CI) = 1.139 (1.133–1.630), $P = 9.4 \times 10^{-4}$ for rheumatic valvular disease; OR (95% CI) = 1.183 (1.112–1.258), $P = 1.1 \times 10^{-7}$ for valvular heart disease), indicating that hemodynamic instability and structural remodeling underlying AF may contribute to the development of valvular diseases.

AF is also known as a consequent phenotype accumulated by multiple atherosclerotic- and metabolic-related traits. Large observational studies have identified these traits as significant risk factors associated with AF[23,24], but the causal relationship between them has not been fully assessed due to potential mediators or confounders of these associations. Therefore, we performed an MR analysis to thoroughly investigate the causality of quantitative traits. We represented the exposures as quantitative traits and selected the distinct variants associated with each trait as instrumental variables. As expected[25], height and BMI were significant predictors for AF (OR (95% CI) = 1.398 (1.164–1.679), $P = 3.3 \times 10^{-4}$; OR (95% CI) = 1.133 (1.061–1.209), $P = 1.8 \times 10^{-4}$, respectively). Furthermore, among atherosclerotic- and metabolic-related traits, we found blood pressure as the only trait with a causal effect on AF development (OR (95% CI) = 1.400 (1.285–1.525), $P = 1.2 \times 10^{-14}$ for systolic blood pressure; OR (95% CI) = 1.455 (1.330–1.591), $P = 2.1 \times 10^{-16}$ for diastolic blood pressure; OR (95% CI) = 1.267 (1.161–1.381), $P = 9.2 \times 10^{-8}$ for pulse pressure; Extended Data Fig. 9b and Supplementary Table 10).

## Discussion

We conducted a large-scale GWAS with approximately 10,000 AF cases in the Japanese population and identified 31 genome-wide significant loci associated with AF. This includes five new loci, where disease-relevant rare and highly East Asian-specific variants were found in the *SYNE1* and *FGF13* loci, suggesting the involvement of functional alteration in the nuclear envelope and ion channels as a mechanism underlying AF. *SYNE1* encodes nesprin-1 (spectrin repeat) protein and, together with the Sad1p/UNC84-domain-containing proteins (SUN1/2), compose the nuclear envelope protein complex via its nucleoplasmic domains to lamin A/C. Mutations in *LMNA* and *SYNE1* have been identified in patients with severe muscle dystrophy and dilated cardiomyopathy[26,27]. Mutations in *SYNE1* cause defects in nuclear morphology, myoblast differentiation and heart development[28], altering the nuclear envelope protein complex that contributes to the structural substrate in atrial arrhythmogenesis. *FGF13* encodes a member of the fibroblast growth factor family, which possesses broad mitogenic and cell survival activities. FGF13 directly binds to the C-terminus of the main cardiac sodium channel (Na$_V$1.5) in the sarcolemma, and FGF13 knockdown in rat cardiomyocytes exhibited a loss of function of Na$_V$1.5-reduced Na$^+$ current density, decreased Na$^+$ channel availability and slowed Na$_V$1.5-reduced Na$^+$ current recovery from inactivation[29]. This evidence of conduction disturbance in cardiomyocytes indicates that *FGF13* is an important target gene associated with AF.

Furthermore, we performed the largest cross-ancestry meta-analysis for AF to date, where 150 genome-wide significant loci were identified, resulting in the discovery of 35 new loci. By integrating these loci with transcriptomic and epigenomic data, we prioritize candidate genes and transcription factors associated with AF. Transcriptome-wide analysis linked AF-associated loci to target genes and particularly revealed *IL6R* as a candidate gene associated with AF. Despite increasing evidence for the role of inflammation in AF pathophysiology[30], only suggestive association between *IL6R* and AF ($P = 5.0 \times 10^{-4}$) has so far been reported[31], and the genetic contribution of inflammatory process to AF development has not been fully elucidated. Our transcriptome-wide analysis revealed a significant association between *IL6R* and AF development, shedding light on the inflammatory signaling as a key pathway in the pathogenesis of AF and a therapeutic target. Additionally, our approach based on the ChIP-seq dataset clearly implicated ERRg as a candidate transcription factor associated with AF. In previous work, ERRg knockdown mice exhibited cardiomyopathy with an arrest of cardiac maturation through transcriptional regulation of genes involved in mitochondrial energy transduction, contractile function and ion transport[32], but the association between AF and ERRg had not been fully examined. Our results from functional studies using iPSCMs indicated a new transcriptional network orchestrated by ERRg in the pathophysiology of AF.

During the last decade, there has been a growing interest in predicting complex diseases or traits using genetic data. PRS is expected to provide a clinical utility to enhance disease risk prediction, whereas previous studies demonstrated comparable or less performance and a weak additive effect of PRS to the established risk prediction models[33,34]. Additionally, the lack of cross-ancestry portability of PRS has also been reported due to the predominant proportion of individuals of European descent in the current GWASs[35]. In this study, we found shared allelic effects of AF-associated variants and genetic correlations between Japanese and European populations (Supplementary Note and Supplementary Fig. 2). Therefore, we exhaustively examined AF-PRS using various combinations of GWASs and multiple parameters to maximize the predictive performance; AF-PRS achieved (1)

a higher performance when applied to the same population as the derivation-GWAS population regardless of the sample size in the single derivation-GWAS category, (2) a higher performance when it was derived from a cross-ancestry meta-GWAS including the Japanese population compared to that derived from a meta-GWAS in a non-Japanese population even with a similar or smaller sample size of derivation-GWAS and (3) the best performance when it was derived from the cross-ancestry meta-GWAS including the Japanese population and with the largest sample size. Furthermore, recent studies have shown the potential utility of PRS in a variety of clinical settings, such as diagnostic refinement[36] and prediction of disease progression[37]. Our study also demonstrated that, in addition to the predictive ability for AF itself, AF-PRS segregated individuals with AF-related phenotypes, such as early onset of AF and cardioembolic stroke, and those with increased risks of long-term cardiovascular and stroke mortalities. This indicated that the cumulative genetic risk for AF could be an indicator for early therapeutic intervention, including anticoagulation in at-risk individuals as a primary prevention of stroke. Taken together, our results have several implications for the clinical utility of AF-PRS, which will be clues for the realization of future precision medicine.

Finally, MR analysis revealed evidence of a causal relationship between AF and relevant diseases or traits, which supports the results from clinical observational studies. In particular, blood pressure was the only trait that showed significant causality among atherosclerotic- and metabolic-related traits, which indicates that blood pressure is an important modifiable risk factor, and the intensive management of blood pressure may reduce the risk of AF development.

In conclusion, our large-scale Japanese and cross-ancestry genetic analyses identified 35 new risk loci and provided insights into the distinct and shared genetic architecture of AF between Japanese and Europeans. Integrative analysis of transcriptome and epigenome data highlighted candidate genes and implicated a transcription factor involved in the mechanism of disease development. Furthermore, analyses of disease prediction and long-term survival demonstrated the clinical utility of the AF-PRS. These data highlight the importance of AF genetics in clinical settings and provide useful evidence for the implementation of genomic medicine.

## Online content

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

[1]Laboratory for Cardiovascular Genomics and Informatics, RIKEN Center for Integrative Medical Sciences, Yokohama, Japan. [2]Department of Cardiovascular Medicine, Graduate School of Medicine, The University of Tokyo, Tokyo, Japan. [3]Department of Drug Discovery Medicine, Kyoto University Graduate School of Medicine, Kyoto, Japan. [4]Program in Medical and Population Genetics, Broad Institute of Harvard and MIT, Cambridge, MA, USA. [5]Cardiovascular Research Center, Massachusetts General Hospital, Boston, MA, USA. [6]Laboratory for Statistical and Translational Genetics, RIKEN Center for Integrative Medical Sciences, Yokohama, Japan. [7]Department of Ocular Pathology and Imaging Science, Kyushu University Graduate School of Medical Sciences, Fukuoka, Japan. [8]Department of Orthopedic Surgery, Hokkaido University Graduate School of Medicine, Sapporo, Japan. [9]Medical Genome Center, Research Institute, National Center for Geriatrics and Gerontology, Obu, Japan. [10]Department of Public Health, Chiba University Graduate School of Medicine, Chiba, Japan. [11]Department of Genomic Medicine, Research Institute, National Cerebral and Cardiovascular Center, Suita, Japan. [12]Department of Computational Biology and Medical Science, Graduate School of Frontier Sciences, The University of Tokyo, Tokyo, Japan. [13]Division of Molecular Pathology, Institute of Medical Science, The University of Tokyo, Tokyo, Japan. [14]Genome Science & Medicine Laboratory, Research Center for Advanced Science and Technology, The University of Tokyo, Tokyo, Japan. [15]RIKEN Center for Integrative Medical Sciences, Yokohama, Japan. [16]Laboratory for Genotyping Development, RIKEN Center for Integrative Medical Sciences, Yokohama, Japan. *A list of authors and their affiliations appears at the end of the paper. ✉e-mail: kaoru.ito@riken.jp; kamatani.yoichiro@edu.k.u-tokyo.ac.jp; komuro-tky@umin.ac.jp

**BioBank Japan Project**

**Koichi Matsuda[13], Yoshinori Murakami[14] & Yoichiro Kamatani[6,13]**

A full list of consortium members is provided in the Supplementary Note.

## Methods

### Study populations

This study was approved by ethics committees of the RIKEN Center for Integrative Medical Sciences, the Institute of Medical Sciences and the University of Tokyo. Informed consent was obtained from all participants. All study participants were Japanese who were registered in the BBJ project (https://biobankjp.org/). The BBJ is a hospital-based national biobank project that collects DNA and serum samples and clinical information from cooperative medical institutes. Approximately 200,000 patients with any of the 47 target diseases were enrolled between 2003 and 2007. All study participants were at least 18 years old.

For GWAS quality control (QC), we excluded samples with a call rate <0.98 and related individuals with PI_HAT > 0.2 by PLINK 2.0 (20 Aug 2018 version). We then excluded samples with a heterozygosity rate > +4 s.d. To identify population stratification, we performed principal component analysis (PCA) using PLINK 2.0 and excluded outliers from the Japanese cluster. For the case samples in GWAS, we selected individuals with AF or atrial flutter diagnosed by a physician based on the general medical practices or documented on a 12-lead electrocardiogram. The demographic features of the case–control cohort are shown in Supplementary Table 11.

The samples in the replication study were registered in the BBJ second cohort, which comprised DNA samples and clinical information of approximately 80,000 new patients with the 38 target diseases collected between 2013 and 2018 to expand research outcomes from the first cohort. We applied the same inclusion criteria to the clinical information of the participants and excluded related individuals estimated by PI_HAT and PCA outliers from the East Asian population. Finally, 48,677 individuals (4,602 cases and 44,075 controls) were included in the replication study.

### Genotyping, imputation and quality control

GWAS participants were genotyped using the Illumina HumanOmniExpress Genotyping BeadChip or a combination of Illumina HumanOmniExpress and HumanExome BeadChips. For genotype QC, we excluded variants with (1) SNP call rate <0.99, (2) MAF < 0.01 and (3) Hardy–Weinberg equilibrium $P < 1.0 \times 10^{-6}$. We prephased the genotypes using EAGLE and imputed dosages with the 1,000 Genome Project Phase 3 (1 KG Phase 3; May 2012)[38] reference panel with 1,037 Japanese in-house reference panel from BBJ using minimac3. For the X chromosome, prephasing was performed in both males and females, and imputation was performed separately for males and females. Dosages of variants in X chromosomes for males were assigned between zero and two.

In the replication study, all participants were genotyped using Illumina Asian Screening Array. We excluded variants meeting any of the following criteria: (1) SNP call rate <98%, (2) a minor allele count of <5 and (3) Hardy–Weinberg equilibrium $P < 1.0 \times 10^{-6}$. Post-QC genotype data were prephased using SHAPEIT2 and imputed using minimac4 with the 1 KG Phase 3 reference panel and 3,256 Japanese in-house reference panel from BBJ. Prephasing and imputation of the X chromosome were performed using the same pipeline applied for autosomes.

### Genome-wide association study

In the Japanese GWAS, association was performed by logistic regression analysis assuming an additive model with adjustment for age, age$^2$, sex and top 20 principal components (PCs) using PLINK 2.0. We selected variants with minimac3 imputation quality score of >0.3 and MAF ≥ 0.001. For the X chromosome, we conducted association analyses in males and females separately and integrated the results using an inverse-variance weighted fixed-effects model implemented in METASOFT (v2.0.1). Heterogeneity between studies was calculated using Cochran's $Q$ test. We filtered variants with strong heterogeneity ($P_{het} < 1.0 \times 10^{-4}$). The genome-wide significance threshold was defined at $P < 5.0 \times 10^{-8}$ for variants with MAF ≥ 1% and $P < 5.71 \times 10^{-9}$ for those with MAF < 1% (0.05/8,753,038 variants). Although the genomic

inflation factor ($\lambda_{GC}$) was 1.12, LD score regression indicated that the inflation was primarily due to polygenic effects (LD score regression intercept = 1.02; Supplementary Fig. 3a). Adjacent genome-wide significant SNPs were grouped into one locus if they were within 1 Mb of each other. We defined a locus as follows: (1) extracted genome-wide significant variants ($P < 5 \times 10^{-8}$) from the association result, (2) added 500 Mb to both sides of these variants and (3) merged overlapping regions. If the locus did not contain coordinates with previously reported genome-wide significant variants (that is, all variants with $P < 5 \times 10^{-8}$ in the previously reported locus), the region was annotated as being new. We mapped variants to nearby genes and functionally annotated genes using Open Targets (https://www.opentargets.org/), in which the pair of variant and gene with the highest variant-to-gene score was selected.

To identify independent association signals in the loci, we conducted a stepwise conditional analysis for genome-wide significant loci defined as described in the GWAS. First, we performed logistic regression conditioning on the lead variants of each locus. We set a locus-wide significance at $P < 1.0 \times 10^{-5}$ and repeated this procedure until none of the variants reached locus-wide significance for each locus.

### LD score regression

We performed LD score regression (version 1.0.0) using selected SNPs with MAF ≥ 0.01 and without the major histocompatibility complex region. For the regression, we used the East Asian LD scores provided by the authors (https://github.com/bulik/ldsc/).

### Cross-ancestry meta-analysis

Summary results from two European AF GWASs (EUR and FIN) were obtained from a previously published website (http://csg.sph.umich.edu/willer/public/afib2018)[7] and from the FinnGen research project website (https://www.finngen.fi/en), respectively. We calculated the LD score regression intercept for each study and confirmed that these two studies were well calibrated (LD score regression intercept for EUR = 1.052 (s.e.m. = 0.012) and FIN = 1.033 (s.e.m. = 0.010); Supplementary Fig. 3b,c). We also calculated the genetic correlation and found a significant genetic correlation between EUR and FIN ($r_g$ = 0.918, s.e.m. = 0.035, $P = 3.9 \times 10^{-155}$).

To account for ancestral heterogeneity among the three studies, we applied the MANTRA algorithm in the cross-ancestry meta-analysis[39], which allows for heterogeneity between diverse ancestry groups and improves performance compared to fixed-effects meta-analysis and random-effects meta-analysis. Variants with MAF ≥ 1% in both the Japanese and European populations were selected for association. We considered SNPs with $\log_{10}$ BF > 6 to be genome-wide significant.

### Transcriptome-wide association study

We performed a TWAS using MetaXcan v0.3.512 (ref. 40), which estimates the association between predicted gene expression levels and a phenotype of interest using summary statistics and gene expression prediction models. We used precomputed prediction models of gene expression in atrial appendage and left ventricular tissues with LD reference data in GTEx v8 and the summary statistic of the cross-ancestry AF-GWAS as input. Bonferroni significance level was set at $P = 4.8 \times 10^{-6}$ (= 0.05/10,414) for the atrial appendage and $P = 5.2 \times 10^{-6}$ (= 0.05/9,702) for the left ventricle to account for the number of genes tested in each tissue. To assess the relationship between AF-associated variants and the candidate genes, we first extracted an AF-associated variant with the lowest association $P$ value among those included in the prediction model obtained from GTEx PredictDB[41] for each candidate gene and then calculated the physical distances from the variant to the TSS of the canonical transcript for each candidate gene. Furthermore, we performed Gene Ontology enrichment analysis using FUMA web application v1.3.7 (ref. 42) with false discovery rate correction considering the number of gene sets tested per category.

## Enrichment analysis of transcription factors

To assess the enrichment of transcription factors in AF-associated loci, we defined AF-associated loci as regions within 500 Mb upstream and 500 Mb downstream of the AF-associated lead variants or proxies with $r^2 > 0.8$ in European samples of 1 KG. We then searched for overlaps of peak-call data archived in the ChIP-Atlas dataset with AF-associated loci and control regions selected from all genomic regions by permutation test. $P$ values were calculated with the two-tailed Fisher's exact probability test (the null hypothesis is that the two regions overlap with the ChIP-Atlas peak-call data in the same proportion). The epigenetic landscapes around cardiac ion channel-related genes were visualized using ChIP-Atlas peak browser and integrative genomics viewer[43].

## Functional analysis of ERRg using iPSCMs

For functional assessment of ERRg, we first prepared iPSCMs, which were established at the University of Tokyo (IRB 11044), and cultured and maintained them in Essential 8-flex medium (ThermoFisher, A2858501). iPSCMs were used between around 20 and 50 passages. The iPSCMs were differentiated into cardiomyocytes[44] with minor modifications. Briefly, all iPSCM lines were differentiated with Asclestem Cardiac Differentiation Media (Nacalai Tesque, 13166-05) until day 12 and maintained with glucose-deficient DMEM (ThermoFisher, A1443001) with sodium DL-lactate (Wako Fujifilm, 128-00056) supplementation for 4 d. The purified cardiomyocytes were replated on gelatin-coated plates in DMEM media supplemented with 10% FBS (Nacalai Tesque, 08458-45). Before downstream assays, the iPSCMs were passaged onto gelatin-coated plates around day 28. An inverse agonist of ERRg, GSK5182 (Selleck, S3449), was dissolved in DMSO and administered to the iPSCMs at 10 μM for 4 d before gene expression and functional analysis. To measure gene expression, total RNA was extracted using TRIzol reagents (ThermoFisher, 15596026) according to the manufacturer's instructions. RNA samples were reverse-transcribed using QuantiTect Reverse Transcription Kit (QIAGEN, 205313). Quantitative real-time PCR was performed using THUNDERBIRD Probe quantitative real-time PCR Mix (Toyobo, QPS-101). Relative expression levels of target genes were normalized to the expression of an internal control gene (*RPS28*) using the comparative Ct method. The primers used for quantitative real-time PCR are listed in Supplementary Table 12. For motion analysis of iPSCMs, the contractile characteristics of iPSCMs were analyzed using SI8000 Cell Motion Imaging System (SONY)[44,45]. The video of synchronously beating iPSCMs was captured, and the motion of each detection point was converted into a vector for quantitative analysis. Cellular motion was analyzed based on the sum of the vector magnitudes. Video images were taken 4 d after drug administration (GSK5182 versus control), and the spontaneous beating rate and the duration of contraction were calculated. To examine calcium handling, we performed a calcium transient assay, in which iPSCMs were plated on a gelatin-coated 96-well plate in DMEM containing 10% FBS. After drug administration, the cells formed a homogenously beating monolayer sheet and were incubated with Cal520AM (AAT, 21130) diluted in Fluoro-Brite medium (ThermoFisher, A1896701) containing 10% FBS for 1 h at 37 °C and 5% $CO_2$. After staining, the medium was replaced with 90 μl of FluoroBrite medium containing 2% FBS. Calcium transient signals were recorded by FDSS/μCell (Hamamatsu Photonics K.K.)[46]. Light source (L11601-01) was used with an output excitation wavelength of 480 nm and an emission of 540 nm, at a sampling rate of 16 Hz for 30 s. Then, 100 nM isoproterenol (Wako Fujifilm, 553-69841) was added to the medium and the calcium transient was recorded again 30 min later. We measured averaged peak counts (beating rates (per min)) and peak width durations at 80% repolarization (PWD80 (ms)). All data analysis was performed using GraphPad Prism 7.04 (GraphPad Software).

## PRS derivation and performance

First, we divided our dataset into the following three groups: (1) a discovery group to derive and validate PRS (6,890 cases and 49,451 controls), (2) a test group to assess PRS performance (2,953 cases and 21,194 controls) and (3) a group for the survival analysis (70,645 controls) (Extended Data Fig. 10). To secure independence between the PRS derivation and validation, we used a tenfold cross-validation approach. Next, we randomly split a discovery group into ten subgroups and used nine of these subgroups for PRS derivation and the remaining one for PRS validation. For each derivation cohort, we performed GWASs in combination with one Japanese and two European GWASs—(1) a population-specific GWAS (BBJ, EUR and FIN), (2) European meta-GWAS (EUR + FIN) and (3) the cross-ancestry meta-GWAS (BBJ + EUR, BBJ + FIN and BBJ + EUR + FIN). Meta-analyses were performed using the fixed effect and the random effect models by METASOFT software. We derived PRS using the pruning and thresholding method and the LDpred2 algorithm. For the pruning and thresholding method, in addition to the meta-analysis models, we applied the $P$ value thresholds as 0.5, $5.0 \times 10^{-2}$, $5.0 \times 10^{-4}$, $5.0 \times 10^{-6}$ and $5.0 \times 10^{-8}$, and the $r^2$ thresholds as 0.8, 0.5 and 0.2. For LDpred2, the variants were restricted to HapMap3 SNPs as recommended[47], and we ran the LDpred2-grid model with the parameters: $p$ (proportion of causal variants) in a sequence of five values from $10^{-4}$ to 1 on a log-scale and sparse option (true or false). We did not tune the parameter for the SNP heritability $h^2$ because the different samples in each derivation cohort did not enable us to determine the optimal $h^2$. For the LD reference, we used 1KG East Asian (EAS) and 1KG European (EUR) populations according to each cohort population as follows: (1) 1KG EAS for a cohort with only East Asian (BBJ), (2) 1 KG EUR for cohorts with only European (EUR, FIN and EUR + FIN) and (3) both 1KG EAS and 1KG EUR for cohorts with multiple ancestries (BBJ + EUR, BBJ + FIN and BBJ + EUR + FIN). Subsequently, we calculated PRSs in the withheld validation cohorts and repeated this procedure ten times by changing the withheld validation cohorts. Finally, we constructed 376 PRSs in total; the PRS with the best performance for each cohort is shown in Fig. 5. The performance of the PRS was measured as (1) Nagelkerke's pseudo $R^2$ obtained by modeling age, sex, the top 20 PCs and normalized PRS and (2) the area under the curve of the receiver operator curve in the same model as Nagelkerke's pseudo $R^2$. The best model/parameter set for each combination model (BBJ, EUR, FIN, BBJ + EUR, BBJ + FIN, EUR + FIN and BBJ + EUR + FIN) was determined by averaging Nagelkerke's pseudo $R^2$ (Supplementary Table 9). Using the best models and parameters determined in the derivation and validation cohorts, we calculated the PRSs and assessed their performance for the independent test cohort. To evaluate the PRS performance in the test cohort, we performed bootstrap over the samples in the test cohort with $5.0 \times 10^4$ replicates and assessed Nagelkerke's pseudo $R^2$ and the area under the curve of the receiver operator curve in each bootstrap group. Before the comparison of the combination models, we evaluated the performance of the base model, which included age, sex and the top 20 PCs (Supplementary Table 9). Next, to compare the performance of the PRS derived from each combination model, we calculated the pairwise difference of Nagelkerke's pseudo $R^2$ ($\Delta R^2$) between each pair of models (two of seven models; 21 combinations) and obtained the two-sided bootstrap $P$ value by counting the number of $\Delta R^2 \leq 0$ or $\Delta R^2 > 0$ and then multiplied the lower value by the minimum estimated $P$ value ($2 \times 1/(5.0 \times 10^4) = 4 \times 10^{-5}$: two-sided). The significance was set at $P = 2.3 \times 10^{-3}$ (0.05/21).

## Association of AF-PRS with relevant phenotypes

We extracted AF case samples with available data on age at AF onset ($n = 7,458$, the median age of AF onset was 63 years of age (IQR = 56–71)) and constructed a linear regression model of age at AF onset including AF-PRS as a dichotomous variable (individual with high PRS (the top 1%, 5%, 10% and 20%) versus those with the remaining PRS) to estimate the difference in the age of AF onset between them adjusted by sex and the top 20 PCs. For the association analysis with stroke phenotypes, we performed a logistic regression analysis adjusted by the use of anti-coagulants or antiplatelets in addition to age, sex and the top 20 PCs,

because antithrombotic therapy is associated with a decreased risk of ischemic stroke and an increased risk of hemorrhagic stroke. We selected the control samples with available data on antithrombotic therapy (n = 121,351), among whom we found 14,120 stroke phenotypes: 8,547 cerebral infarction, 111 cardioembolic strokes, 1,429 atherothrombotic infarction, 1,230 lacunar infarction, 1,061 cerebral hemorrhages and 879 subarachnoid hemorrhages.

## Survival analysis

The Cox proportional hazards model was used to assess the association between AF-PRS and long-term mortality. We obtained survival follow-up data with the ICD-10 code for 132,737 individuals from the BBJ dataset[48,49]. The causes of death were classified into three categories according to ICD-10 codes as follows: (1) primary outcome for all-cause death, (2) secondary outcome for cardiovascular death (100–199) and noncardiovascular death (not 100–199) and (3) tertiary outcome for heart failure death (150), ischemic heart disease death (120–125) and stroke death (160, 161, 163 and 164). The median follow-up period was 8.4 years (IQR 6.8–9.9). The Cox proportional hazards model was adjusted for sex, age, the top 20 PCs and disease status. Analyses were performed with the R package survival v.2.44, and survival curves were estimated using the R package survminer v.0.4.6, with modifications.

## Mendelian randomization

We extracted summary statistics from the UK Biobank (http://www.nealelab.is/uk-biobank/). To avoid sample overlap, we selected AF-associated variants from the cross-ancestry meta-analysis in combination with BBJ and FIN, although the statistical power to detect the associations with AF decreased. To select independent variants for exposure, genome-wide significant variants ($P < 5 \times 10^{-8}$) were pruned ($r^2 < 0.01$; LD window of 10,000 kb; using European samples of 1KG for LD reference)[50]. For the assessment of the causal effect of AF on cardiovascular diseases, we excluded variants associated with cardiovascular risk factors such as hypertension, cholesterol, diabetes mellitus and smoking and those with cardiovascular diseases from the list of instrument variables to avoid the pleiotropic effects of them using PhenoScanner V2 (http://www.phenoscanner.medschl.cam.ac.uk/). Then, we performed MR analysis using TwoSampleMR package in R v4.0.3, with AF-associated variants as instrument variables and variants associated with cardiovascular diseases as outcome variables. Next, we assessed the causal effects of quantitative traits related to anthropometry, metabolites, serum protein, kidney function, liver function, hematocyte count and blood pressure on AF. To exclude variants with pleiotropic effects, we also used PhenoScanner to identify variants associated with risk factors for AF such as hypertension, diabetes mellitus and obesity from the list of instrument variables, unless exposure was a risk factor itself. Then, we performed MR analysis, where variants associated with quantitative traits were used as instrument variables and AF-associated variants as outcome variables. Causal estimates were based on the inverse-variance-weighting (IVW) method. To exclude horizontal pleiotropic outliers, we performed MR-PRESSO for instrument variables[51]. We also calculated Cochran's Q statistics for heterogeneity between the causal effects using IVW and the MR-Egger intercept for directional pleiotropy.

## Reporting summary

Further information on research design is available in the Nature Portfolio Reporting Summary linked to this article.

## Data availability

Summary statistics of Japanese GWAS and the cross-ancestry meta-analysis and the data for the calculation of PRS derived from the current study are publicly available in the National Bioscience Database Center (research ID: hum0014, https://humandbs.biosciencedbc.jp/en/). The cross-ancestry GWAS summary statistics and polygenic score are also available through the NHGRI-EBI GWAS catalog (study accession: GCST90204201, https://www.ebi.ac.uk/gwas/downloads/summary-statistics) and Polygenic Score catalog (https://www.pgscatalog.org/, score ID: PGS002814), respectively. The phenotype information can be provided by the BBJ project upon request (https://biobankjp.org/english/index.html).

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

## Acknowledgements

We thank the staff of BBJ for their assistance in collecting samples and clinical information. We thank Y. Kaneko (the Uniersity of Tokyo) for technical assistance for functional experiments using iPSCMs. We also thank A.P. Morris (University of Liverpool) for providing us with MANTRA software and valuable advice.

This research was funded by the Japan Agency for Medical Research and Development (AMED) (JP22ek0210164 to K.I., S.N. and I.K., JP21tm0724601 to K.I., S.N. and I.K., JP20km0405209 and JP20ek0109487 to K.M., K.I., S.N., I.K., JP20ek0109440 and JP22ek0210172 to S.N. and I.K., JP18km0405209 to I.K., JP21ek0109543 to S.N. and JP22bm1123011 to S.N.), MSD Life Science Foundation (to K. Miyazawa), the Japan Society for the Promotion of Science (a Grand-in-Aid for Scientific Research (S) to I.K., a Grant-in-Aid for Scientific Research (A) to K.I. and S.N., a Grant-in-Aid for Scientific

Research (B) to K.I., a Grant-in-Aid for Early-Career Scientists to K. Miyazawa, and R.K. and H.I., and JSPS Fellows to Z.Z.), Research Funding for Longevity Sciences from the NCGG (21–23 to K.O. and K.I.), the Japan Science and Technology Agency (NBDC and PRESTO to S.O.) and Sakakibara Memorial Research Grant from the Sakakibara Heart Foundation (to H.M.). BBJ was supported by the Tailor-Made Medical Treatment Program of the Ministry of Education, Culture, Sports, Science and Technology (MEXT) and AMED under grant numbers JP22km0605001, JP17km0305002 and JP17km0305001.

## Author contributions

K. Miyazawa, K.I., C.T., H. Akazawa, Y. Kamatani and I.K. conceived and designed the study. C.T., K. Matsuda, Y.M., M. Kubo and Y. Kamatani collected and managed the BBJ sample. Y.M., A.T., M. Kubo and Y. Kamatani performed the genotyping. K. Miyazawa, K.I., M.I., Z.Z., M. Kubota, H.M., C.T., S.O. and Y. Kamatani performed the statistical analyses. K. Miyazawa, K.I., M.I., Z.Z., M. Kubota, S.N., H.M., S.K., H.I., M.A., Y. Koike, R. K., H.Y., K.O., Y.O., C.T. and S.O. contributed to data processing, analysis and interpretation. K.I., H. Aburatani, H. Akazawa, Y. Kamatani and I.K. supervised the study. K. Miyazawa, K.I., and M.H. wrote the manuscript, and many authors provided valuable edits.

## Competing interests

The authors declare no competing interests associated with this manuscript.

## Additional information

**Extended data** is available for this paper at https://doi.org/10.1038/s41588-022-01284-9.

**Correspondence and requests for materials** should be addressed to Kaoru Ito, Yoichiro Kamatani or Issei Komuro.

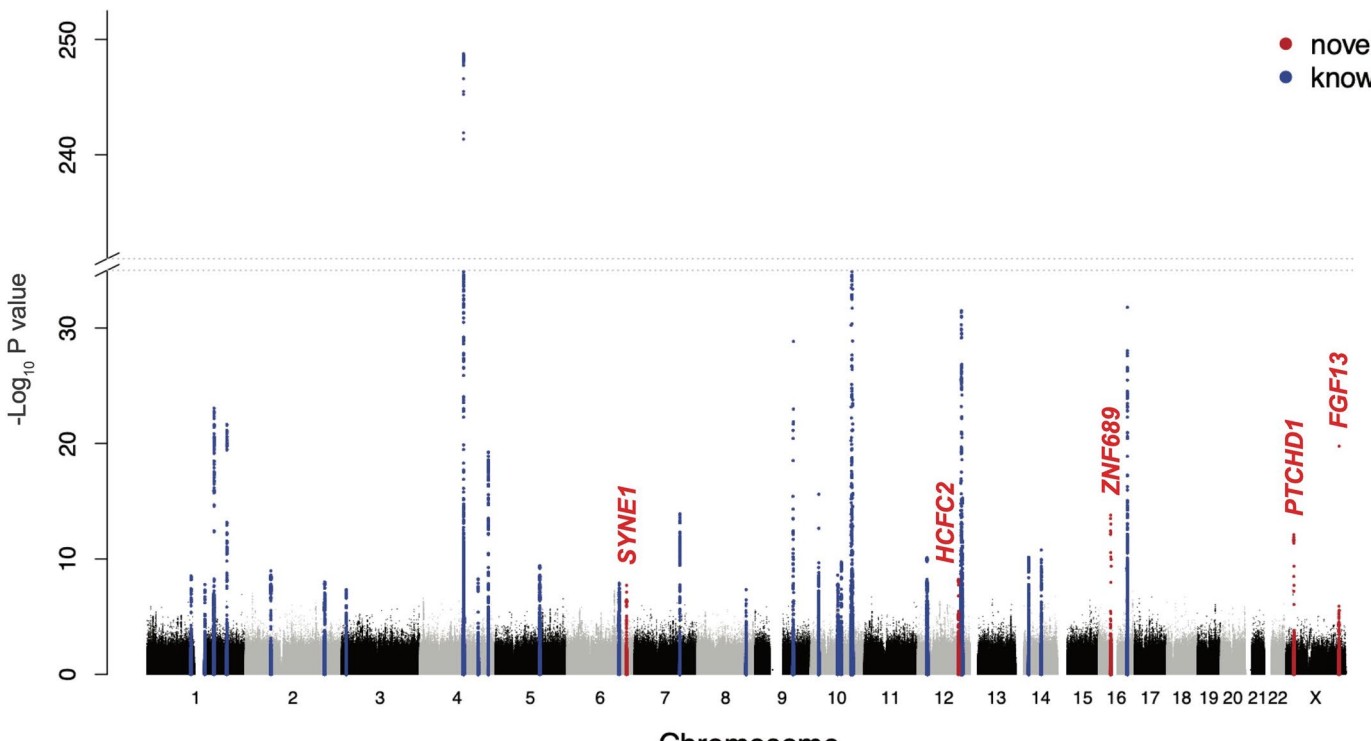

**Extended Data Fig. 1 | Manhattan plot for the Japanese GWAS.** The results of the Japanese GWAS (9,826 AF cases and 140,446 controls) are shown. The negative $\log_{10} P$ values on the $y$-axis are shown against the genomic positions (hg19) on the $x$-axis. Association signals that reached a genome-wide significance level ($P < 5.0 \times 10^{-8}$) are shown in blue if previously reported loci and in red if novel loci. Two-sided $P$ values were calculated using a logistic regression model. GWAS, genome-wide association study; AF, atrial fibrillation.

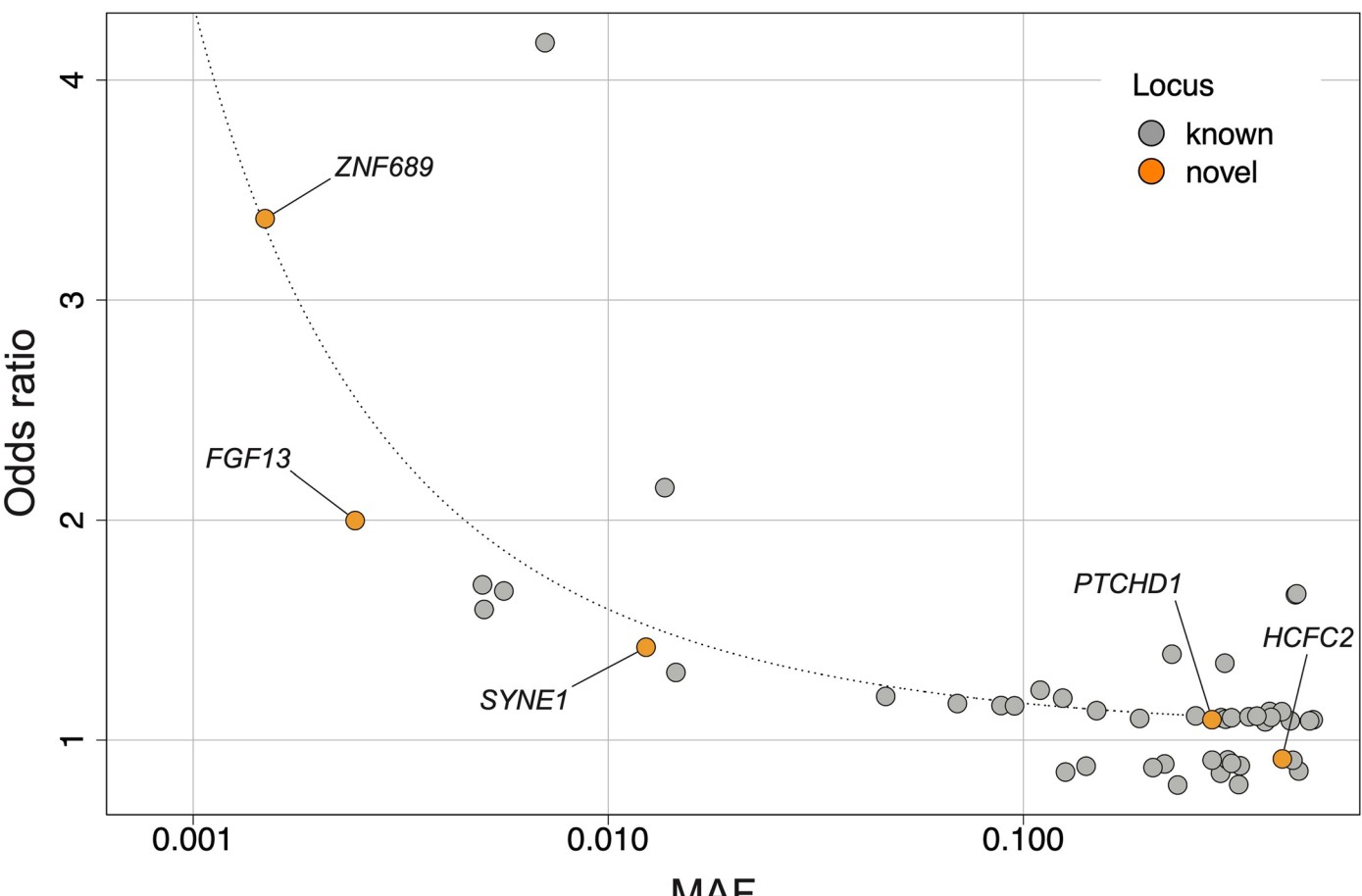

**Extended Data Fig. 2 | Independent association signals in AF development.** The ORs for AF development of 49 independent signals in the Japanese GWAS (31 lead variants and 18 independent variants) were plotted against the risk allele frequencies. Novel variants are highlighted in orange with annotated genes. The dotted line indicates 80% detection power at a significance threshold of 5.0 × 10$^{-8}$. OR, odds ratio.

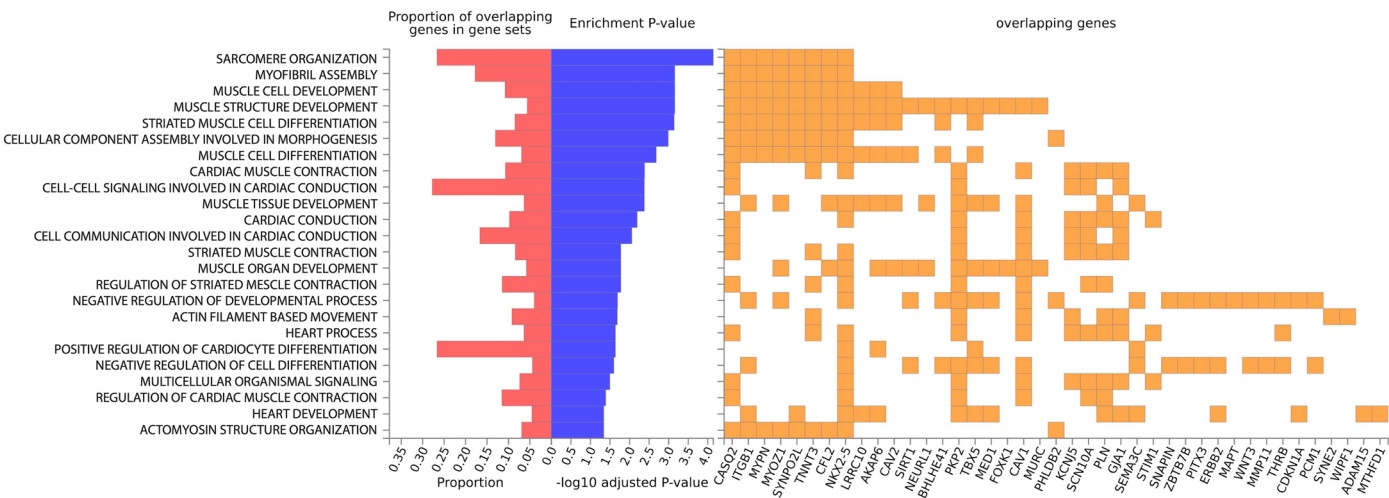

**Extended Data Fig. 3 | Pathway analysis of candidate genes.** Gene ontology analysis enriched in the candidate genes identified by TWAS using hypergeometric tests in FUMA web application. The significance level accounts for multiple testing of gene sets using a Benjamini-Hochberg correction, and gene sets with adjusted P value ≤ 0.05 are displayed.

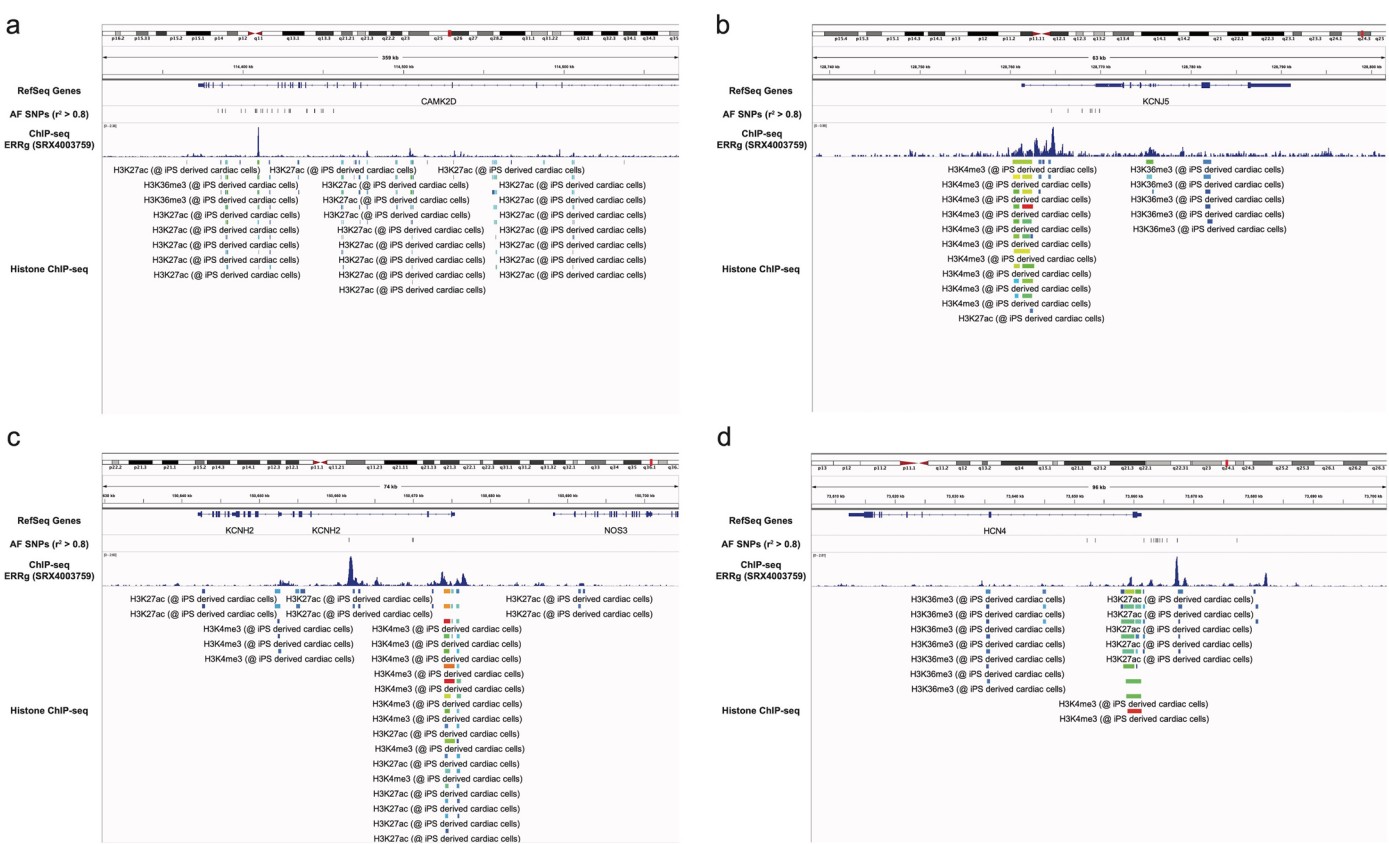

**Extended Data Fig. 4 | Ion channel-related gene and ERRg. a**, *CAMK2D* locus. **b**, *KCNJ5* locus. **c**, *KCNH2* locus. **d**, *HCN4* locus. Plots show AF-associated variants ($r^2 > 0.8$ in European samples in 1KG) and ChIP-seq track of the ERRg experiment (SRX4003759) and histone modification markers in iPSC-derived cardiac cells. iPSC, induced pluripotent stem cells.

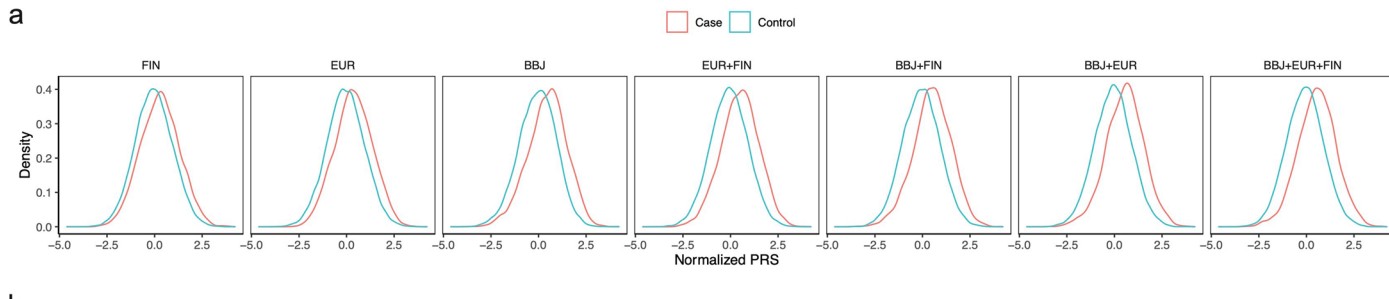

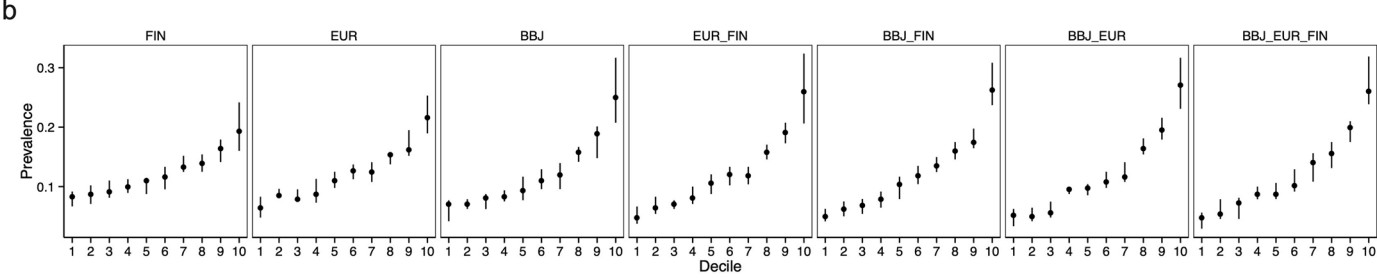

**Extended Data Fig. 5 | PRS distribution and AF prevalence.** PRS distribution and AF prevalence in the test cohort are shown ($n_{case}$ = 2,949 and $n_{control}$ = 21,081). **a**, Distribution of PRS in the case and control samples for each combination of GWAS. **b**, Prevalence of AF based on the AF-PRS deciles in each combination of GWAS. The number of individuals in each decile is 2,403. Data are presented as medians and 95% CI. PRS, polygenic risk score; CI, confidence interval.

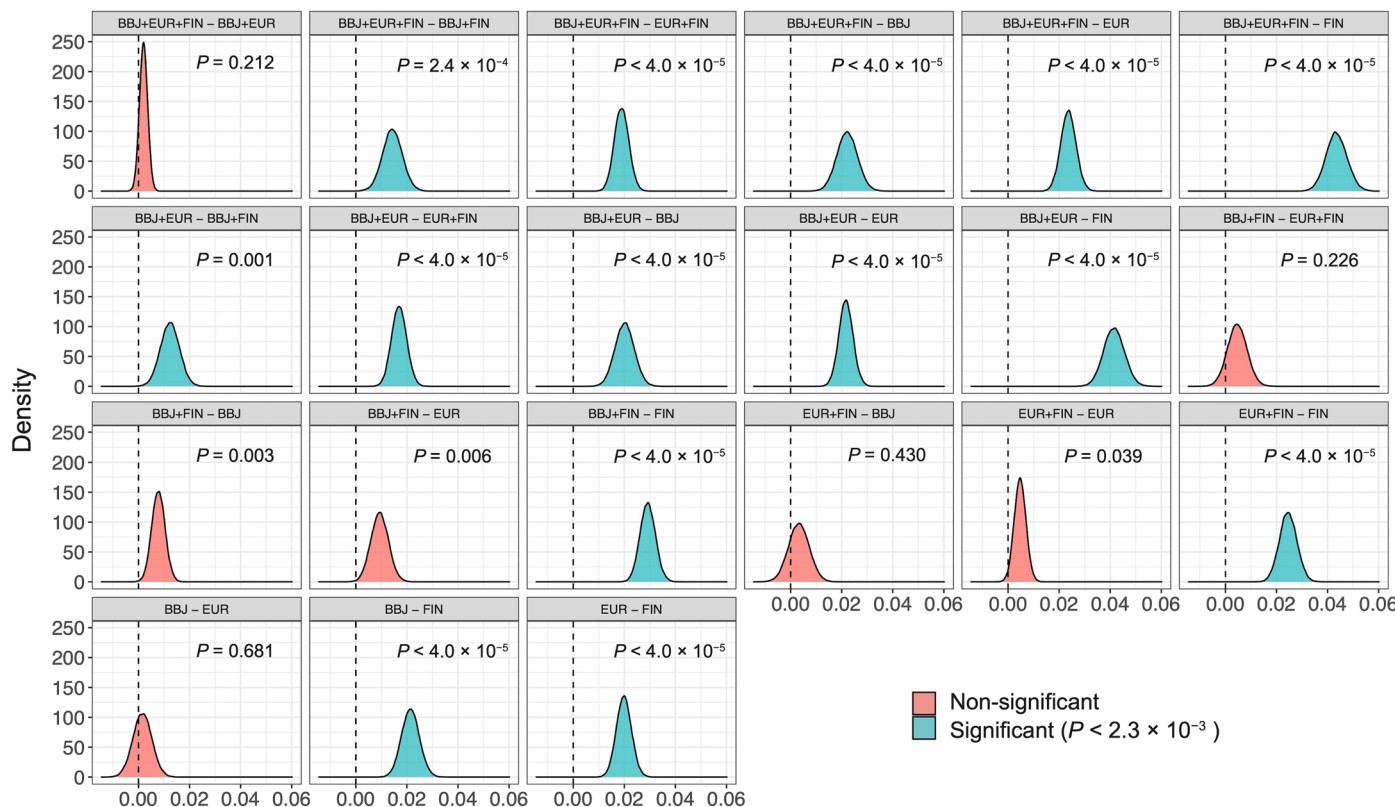

**Extended Data Fig. 6 | Pairwise comparison of PRS performance.** Distribution of the pairwise difference of Nagelkerke's pseudo $R^2$ ($\Delta$ pseudo $R^2$: Pseudo $R^2_{Score}$ Y – Pseudo $R^2_{Score}$ X, X and Y are found at the top of the panel) between each pair of GWAS models. The distributions were obtained by bootstrapping $5.0 \times 10^4$ times. Two-sided bootstrap $P$ values were calculated by counting the number of $\Delta$ pseudo $R^2 \leq 0$ or $\Delta$ pseudo $R^2 > 0$ and then multiplying the lower value by the minimum estimated $P$ value ($2 * 1 / (5.0 \times 10^4) = 4 \times 10^{-5}$: two-sided). The significance was set at $P = 2.3 \times 10^{-3}$ (0.05/21).

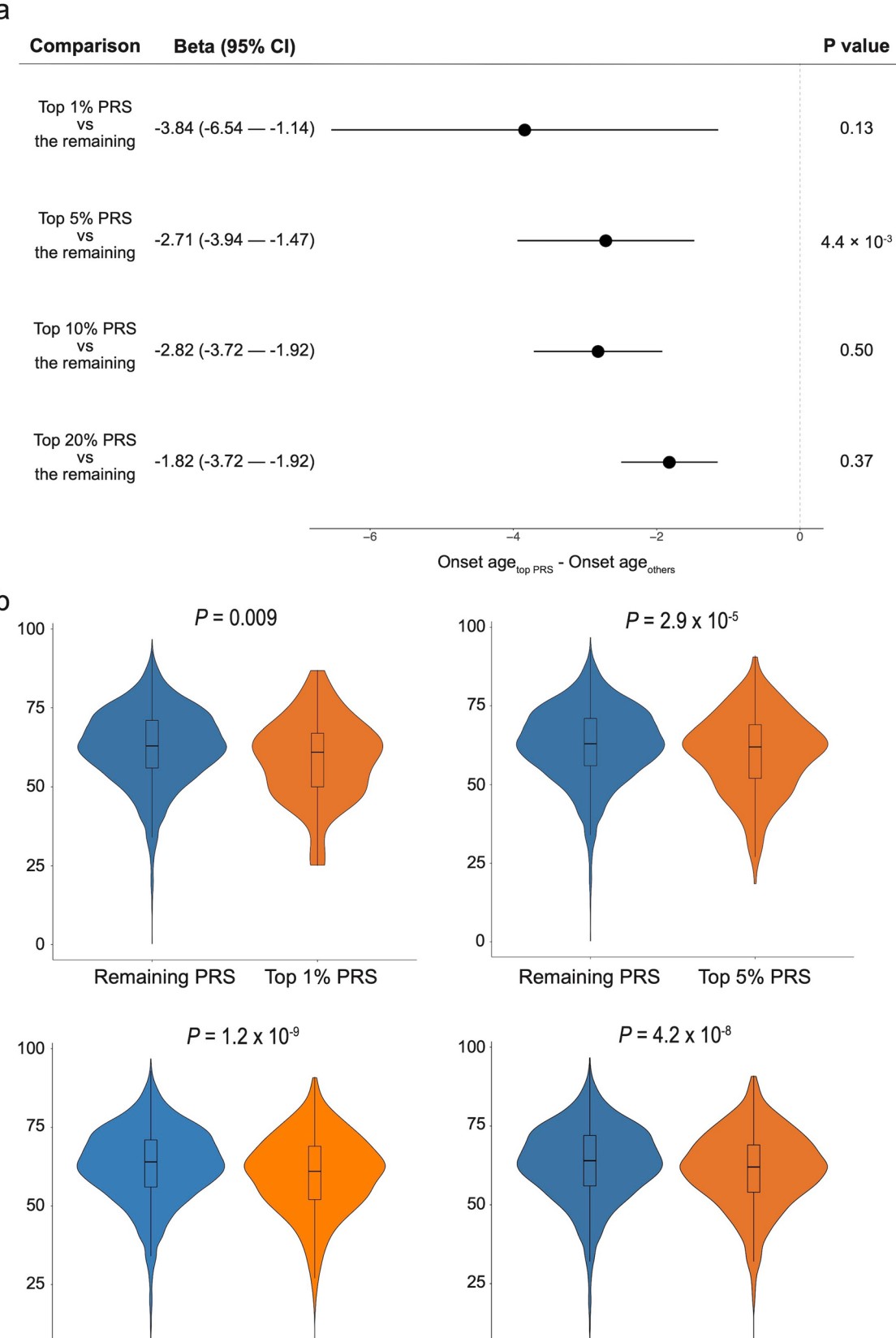

**Extended Data Fig. 7 | See next page for caption.**

**Extended Data Fig. 7 | Impact of AF-PRS on the age of AF onset. a**, Difference in the age of AF onset between individuals with high PRS (the top 1%, 5%, 10%, and 20%) and those with the remaining PRS. We constructed linear regression models by adjusting for sex and the top 20 PCs. *P* values were calculated by linear regression analysis comparing individuals with high PRS versus those with the remaining PRS and the significance was set at $P = 8.3 \times 10^{-3}$ (0.05/6). Each dot represents an estimated effect size (β) with an error bar indicating the 95% CI of the estimate obtained from the linear regression models. **b**, Comparison of the age of AF onset between samples with high PRS and those with remaining PRS. Box plot represents the median, the bounds represent the first and third quartile, and the whiskers reach to 1.5 times the interquartile range. In **a** and **b**, data are shown for individuals based on PRS ($n_{top1\%} = 75$ vs. $n_{remaining} = 7,383$; $n_{top5\%} = 373$ vs. $n_{remaining} = 7,085$; $n_{top10\%} = 746$ vs. $n_{remaining} = 6,712$; $n_{top20\%} = 1,492$ vs. $n_{remaining} = 5,966$).

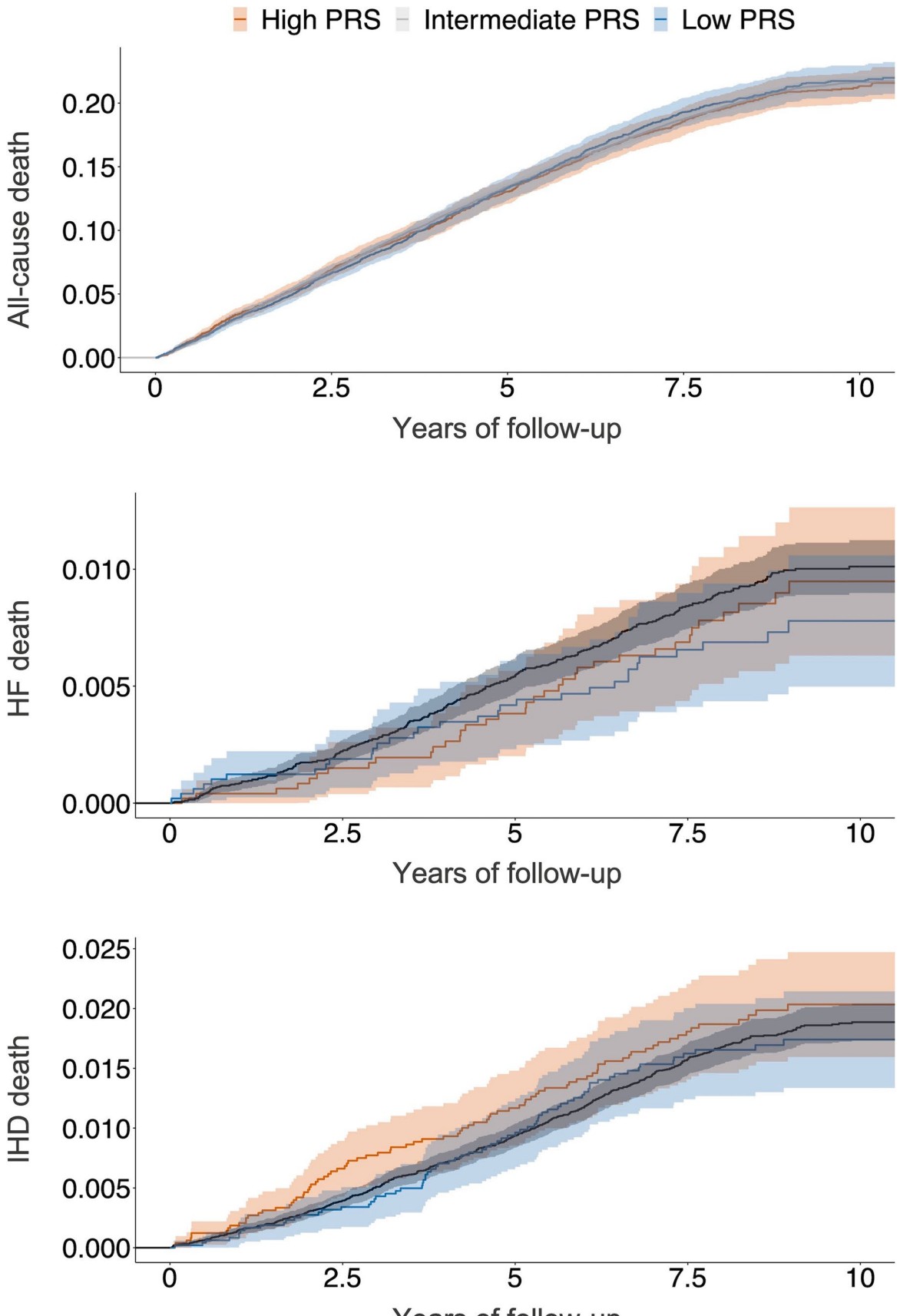

**Extended Data Fig. 8 | Impact of AF-PRS on long-term mortality.** Kaplan-Meier estimates of cumulative events from all-cause death (top), heart failure death (middle), and ischemic heart disease death (bottom) are shown with a band of 95% CI. Individuals are classified into high PRS (top 10 percentile, red), low PRS (bottom 10 percentile, blue), and intermediate (others, green).

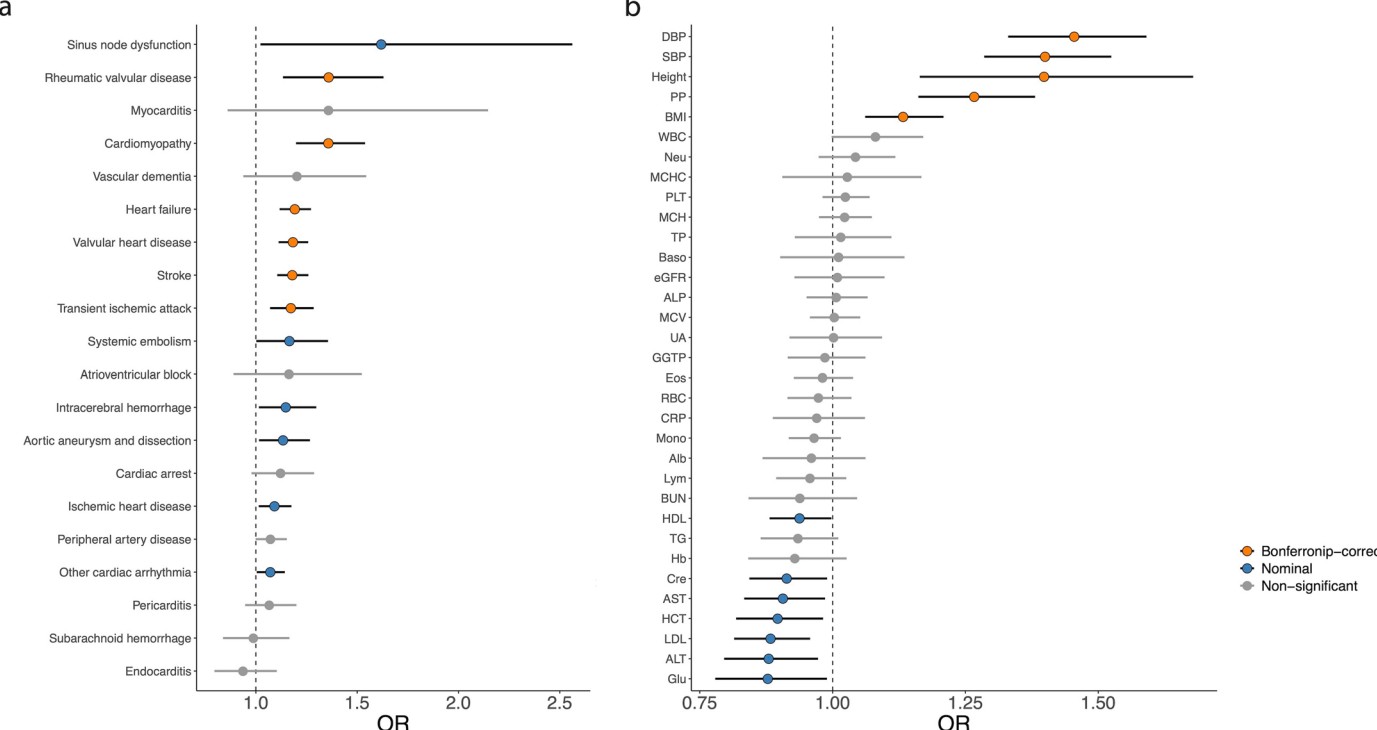

**Extended Data Fig. 9 | Mendelian randomization analysis. a**, Causal effect of AF on cardiovascular diseases. **b**, Causal effect of quantitative traits on AF development. Each dot represents a causal estimate on the OR scale with an error bar indicating the 95% CI of the estimate. We analyzed the MR results to estimate causal effects using associated variants without pleiotropic effects. The $P$ values were determined using the IVW two-sample MR method and the significance was set at $P = 2.5 \times 10^{-3}$ (0.05/20) for **a**, and $P = 1.5 \times 10^{-3}$ (0.05/33) for **b**, respectively. The sample sizes for individual traits and the P values of MR analyses are shown in Supplementary Table 10. MR, Mendelian randomization; IVW, inverse-variance-weighting; DBP, diastolic blood pressure; SBP, systolic blood pressure; PP, pulse pressure; BMI, body mass index; Neu, neutrophil count; MCH, mean corpuscular

hemoglobin; WBC, white blood cell count; PLT, platelet count; TP, total protein; MCHC, mean corpuscular hemoglobin concentration; eGFR, estimated glomerular filtration rate; MCV, mean corpuscular volume; UA, uric acid; Eos, eosinophil count; ALP, alkaline phosphatase; Baso, basophil count; GGT, gamma-glutamyl transferase; CRP, C-reactive protein; HDL, high-density lipoprotein cholesterol; Mono, monocyte count; BUN, blood urea nitrogen; TG, triglyceride; Cre, serum creatinine; RBC, red blood cell; Hb, hemoglobin; AST, aspartate aminotransferase; LDL, low-density lipoprotein cholesterol; Lym, lymphocyte count; Alb, albumin; HCT, hematocrit; ALT, alanine aminotransferase; Glu, blood sugar.

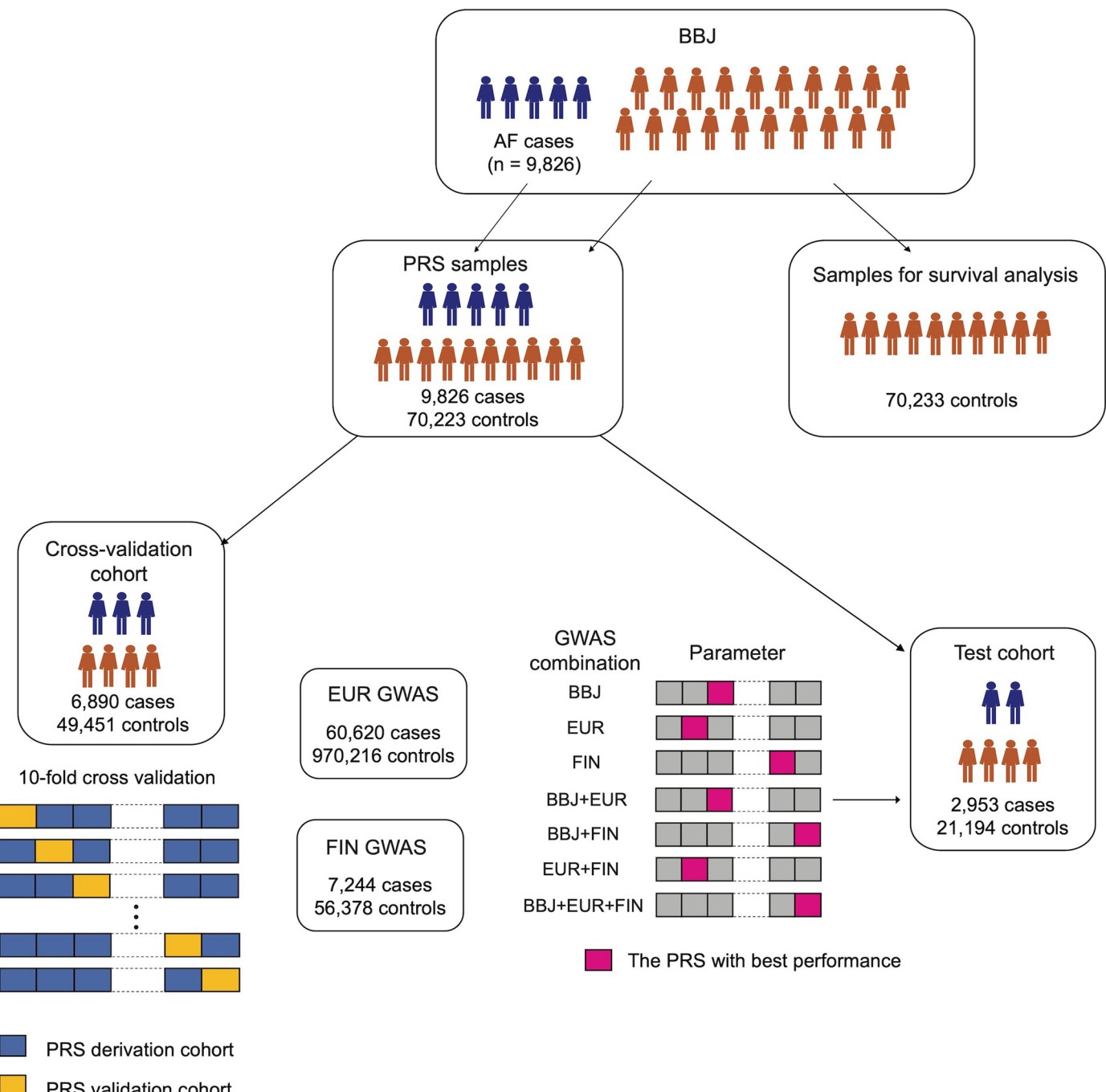

**Extended Data Fig. 10 | Analytical scheme for PRS derivation, validation, and test cohorts, and survival analysis.** Schematic representation for cross-validation, performance testing of AF-PRS in the independent test cohort, and survival analysis for AF-PRS.

# Reporting Summary

## Statistics

For all statistical analyses, confirm that the following items are present in the figure legend, table legend, main text, or Methods section.

| n/a | Confirmed | |
|---|---|---|
| ☐ | ☒ | The exact sample size (*n*) for each experimental group/condition, given as a discrete number and unit of measurement |
| ☐ | ☒ | A statement on whether measurements were taken from distinct samples or whether the same sample was measured repeatedly |
| ☐ | ☒ | The statistical test(s) used AND whether they are one- or two-sided<br>*Only common tests should be described solely by name; describe more complex techniques in the Methods section.* |
| ☐ | ☒ | A description of all covariates tested |
| ☐ | ☒ | A description of any assumptions or corrections, such as tests of normality and adjustment for multiple comparisons |
| ☐ | ☒ | A full description of the statistical parameters including central tendency (e.g. means) or other basic estimates (e.g. regression coefficient) AND variation (e.g. standard deviation) or associated estimates of uncertainty (e.g. confidence intervals) |
| ☐ | ☒ | For null hypothesis testing, the test statistic (e.g. *F*, *t*, *r*) with confidence intervals, effect sizes, degrees of freedom and *P* value noted<br>*Give P values as exact values whenever suitable.* |
| ☒ | ☐ | For Bayesian analysis, information on the choice of priors and Markov chain Monte Carlo settings |
| ☒ | ☐ | For hierarchical and complex designs, identification of the appropriate level for tests and full reporting of outcomes |
| ☐ | ☒ | Estimates of effect sizes (e.g. Cohen's *d*, Pearson's *r*), indicating how they were calculated |

*Our web collection on statistics for biologists contains articles on many of the points above.*

## Software and code

Policy information about availability of computer code

| Data collection | No software was used for data collection. |
|---|---|
| Data analysis | We used open source softwares for the analysis, as listed below.<br><br>bgzip/tabix (0.2.6), http://www.htslib.org/doc/tabix.html;<br>Minimac3 (2.0.1), https://genome.sph.umich.edu/wiki/Minimac3;<br>Minimac4 (1.0.0), https://genome.sph.umich.edu/wiki/Minimac4;<br>Eagle (2.4.1), https://data.broadinstitute.org/alkesgroup/Eagle;<br>SHAPEIT2 (r837), https://mathgen.stats.ox.ac.uk/genetics_software/shapeit/shapeit.html;<br>ANNOVAR (2017Jun01), http://annovar.openbioinformatics.org;<br>Open Targets, https://www.opentargets.org;<br>PLINK (2.0), https://www.cog-genomics.org/plink;<br>LDSC (1.0.0), https://github.com/bulik/ldsc;<br>GCTA-GREML, http://cnsgenomics.com/software/gcta/#BivariateGREMLanalysis;<br>Genetic Association Study Power Calculator, https://csg.sph.umich.edu/abecasis/gas_power_calculator;<br>MANTRA (2.0), provided by the author;<br>METASOFT (2.0), http://genetics.cs.ucla.edu/meta;<br>METAL (2011-03-25), https://genome.sph.umich.edu/wiki/METAL;<br>bedtools (2.25.0), https://bedtools.readthedocs.io/en/latest;<br>LocusZoom (1.4), http://locuszoom.sph.umich.edu;<br>R (3.5.1), https://www.r-project.org;<br>python (4.0.3), https://www.python.org;<br>DEPICT (1, release 194), https://data.broadinstitute.org/mpg/depict; |

SpliceAI (v.1.3.1), https://github.com/Illumina/SpliceAI;
Popcorn software (1.0), https://github.com/brielin/Popcorn;
MetaXcan (0.3.5), https://github.com/hakyimlab/MetaXcan/wiki;
FUMA (v1.3.7), https://fuma.ctglab.nl;
LDpred2, https://privefl.github.io/bigsnpr/articles/LDpred2.html;
TwoSampleMR, https://github.com/MRCIEU/TwoSampleMR;
MRPRESSO (1.0), https://github.com/rondolab/MR-PRESSO;
PhenoScanner V2, http://www.phenoscanner.medschl.cam.ac.uk;
GraphPad Prism 7.04, https://www.graphpad.com/scientific-software/prism;
coxph in the R survival package v2.44

Further details are described in the Method section.

For manuscripts utilizing custom algorithms or software that are central to the research but not yet described in published literature, software must be made available to editors and reviewers. We strongly encourage code deposition in a community repository (e.g. GitHub). See the Nature Portfolio guidelines for submitting code & software for further information.

## Data

Policy information about availability of data

All manuscripts must include a data availability statement. This statement should provide the following information, where applicable:
- Accession codes, unique identifiers, or web links for publicly available datasets
- A description of any restrictions on data availability
- For clinical datasets or third party data, please ensure that the statement adheres to our policy

We used publicly available data   as listed below.

Summary results of the European AF meta-analysis are publicly available on http://csg.sph.umich.edu/willer/public/afib2018.
Summary results from the FinnGen release 2 data can be accessed through application on https://www.finngen.fi/en/access_results.
Summary statistics of quantitative trait loci in the BioBank Japan are publicly available on http://jenger.riken.jp/en/ (JENGER).
Summary statistics of the UK Biobank are publicly available on http://www.nealelab.is/uk-biobank.

ggnomAD, https://gnomad.broadinstitute.org;
ClinVar, https://www.ncbi.nlm.nih.gov/clinvar;
1000 Genomes Project, http://www.1000genomes.org;
HapMap project, http://hapmap.ncbi.nlm.nih.gov;
GWAS catalog, https://www.ebi.ac.uk/gwas;
dbSNP, https://www.ncbi.nlm.nih.gov/snp;
PubTator, https://www.ncbi.nlm.nih.gov/research/pubtator;
GTEx, https://gtexportal.org/home;
ChIP-Atlas, https://chip-atlas.org;
ENCODE, https://www.encodeproject.org

Further details are described in the Method section.

The summary statistics of the Japanese GWAS and the cross-ancestry meta-analysis, and the data for the calculation of PRS derived from the current study are publicly available in the National Bioscience Database Center (research ID: hum0014, https://humandbs.biosciencedbc.jp/en/). The phenotype information can be provided by the BioBank Japan project upon a request (https://biobankjp.org/english/index.html).

# Field-specific reporting

Please select the one below that is the best fit for your research. If you are not sure, read the appropriate sections before making your selection.

☒ Life sciences          ☐ Behavioural & social sciences          ☐ Ecological, evolutionary & environmental sciences

For a reference copy of the document with all sections, see nature.com/documents/nr-reporting-summary-flat.pdf

# Life sciences study design

All studies must disclose on these points even when the disclosure is negative.

| Sample size | Because we aimed to create the largest sample size in the Japanese population in order to gain the statistical power, we included as many case and control individuals as possible in our analysis. |
|---|---|
| Data exclusions | We excluded individuals according to the standard quality control procedure of GWAS. Further details are described in the Method section. |
| Replication | Our Japanese GWAS identified 31 genome-wide significant loci, where 26 previously reported loci were replicated. We also performed a replication study for the newly identified five loci using an independent Japanese cohort, the BioBank Japan second cohort (4,602 cases and 44,075 controls), and confirmed that all signals were successfully replicated with nominal associations ($P < 0.05$) in the same effect direction. |
| Randomization | Randomization is not applicable, because this is a population based case-control analysis. |

| Blinding | Blinding is not applicable, because this is a population based case-control analysis. |
| --- | --- |

# Reporting for specific materials, systems and methods

We require information from authors about some types of materials, experimental systems and methods used in many studies. Here, indicate whether each material, system or method listed is relevant to your study. If you are not sure if a list item applies to your research, read the appropriate section before selecting a response.

## Materials & experimental systems

| n/a | Involved in the study |
| --- | --- |
| ☒ | ☐ Antibodies |
| ☐ | ☒ Eukaryotic cell lines |
| ☒ | ☐ Palaeontology and archaeology |
| ☒ | ☐ Animals and other organisms |
| ☐ | ☒ Human research participants |
| ☒ | ☐ Clinical data |
| ☒ | ☐ Dual use research of concern |

## Methods

| n/a | Involved in the study |
| --- | --- |
| ☒ | ☐ ChIP-seq |
| ☒ | ☐ Flow cytometry |
| ☒ | ☐ MRI-based neuroimaging |

## Eukaryotic cell lines

Policy information about cell lines

| Cell line source(s) | All cell lines were iPCs from the University of Tokyo (IRB #11044). Peripheral blood mononuclear cells were obtained from healthy volunteers with their consent, and iPS cells were established using episomal plasmid vectors encoding the reprogramming factors. |
| --- | --- |
| Authentication | All cell lines were confirmed to be normal by karyotype analysis, and pluripotency was confirmed by RT-PCR and immunostaining of stem cell markers such as SOX2, POU5F1, and NANOG . All cell lines were matched to the original donor by genotype, using STR anaysis. |
| Mycoplasma contamination | All cell lines were tested with Mycoplasma using MycoAlert Plus Mycoplasma detection kit (Lonza) and were found negative. |
| Commonly misidentified lines (See ICLAC register) | NA |

## Human research participants

Policy information about studies involving human research participants

| Population characteristics | Population characteristics such as the number of male and female subjects, the mean and standard deviation of the age in the Japanese GWAS are provided in Supplementary Table 11. |
| --- | --- |
| Recruitment | The BioBank Japan (BBJ) project<br>The BBJ project (https://biobankjp.org/english.) is a hospital-based biobank. Participants were recruited at 12 medical institutes throughout Japan. Detailed information is described in the following papers: Nagai A. et al. Overview of the BioBank Japan Project: Study design and profile. J Epidemiol 27, S2–S8 (2017), Hirata M. et al. Cross-sectional analysis of BioBank Japan clinical data: A large cohort of 200,000 patients with 47 common diseases. J Epidemiol 27, S9–S21 (2017). |
| Ethics oversight | Our study was approved by the appropriate Institutional Review Board at each facility and the informed consent was obtained from all participants in each study. We obtained approval from ethics committees of (1) RIKEN Center for Integrative Medical Sciences, (2) the Institute of Medical Sciences, The University of Tokyo, (3) Kyoto University, (4) National Cancer Center Japan, (5) Nagoya University, (6) Aichi Cancer Center, (7) Osaka University. |

Note that full information on the approval of the study protocol must also be provided in the manuscript.

