## [Peer Review File · Nature Genetics]

Peer Review Information

Manuscript Title: Cross-ancestry genome-wide analysis of atrial fibrillation provides insights into disease biology and enables polygenic prediction of cardioembolic risk

Corresponding author name(s): Dr Kaoru Ito

Reviewer Comments & Decisions:

Decision Letter, initial version:
--

30th November 2021

Dear Dr. Ito,

Your Article "Trans-ancestry genome-wide analysis of atrial fibrillation provides new insights into disease biology and enables polygenic prediction of cardioembolic risk" has been seen by three referees. You will see from their comments below that, while they find your work of potential interest, they have raised substantial concerns that must be addressed. In light of these concerns, we cannot accept the manuscript for publication at this time, but we would be interested in considering a substantially revised version that addresses the referees' criticisms.

We hope you will find the referees' comments useful as you decide how to proceed. If you wish to submit a substantially revised manuscript, please bear in mind that we will be reluctant to approach the referees again in the absence of major revisions.

To guide the scope of the revisions, the editors discuss the referee reports in detail within the team, including with the chief editor, with a view to identifying key priorities that should be addressed in revision, and sometimes overruling referee requests that are deemed beyond the scope of the current study. In this case, we ask that you carefully address all technical queries related to the association analyses and their interpretation and extend the follow-up analyses where feasible as proposed by the referees. In particular, we encourage you to seek a suitable East Asian replication sample to validate the novel loci identified in the Japanese ancestry GWAS, and we ask that you revise the sex-stratified GWAS, co-localization, TWAS, and PRS analyses taking all referee comments into account. We hope you will find this prioritized set of referee points to be useful when revising your study. Please do not hesitate to get in touch if you would like to discuss these issues further.

If you choose to revise your manuscript taking into account all reviewer and editor comments, please highlight all changes in the manuscript text file. At this stage we will need you to upload a copy of the manuscript in MS Word .docx or similar editable format.

*2) If you have not done so already please begin to revise your manuscript so that it conforms to our Article format instructions, available [here](http://www.nature.com/ng/authors/article_types/index.html). Refer also to any guidelines provided in this letter.

[redacted]

If you wish to submit a suitably revised manuscript we would hope to receive it within 3-6 months. If you cannot send it within this time, please let us know. We will be happy to consider your revision so long as nothing similar has been accepted for publication at Nature Genetics or published elsewhere. Should your manuscript be substantially delayed without notifying us in advance and your article is eventually published, the received date would be that of the revised, not the original, version.

Nature Genetics is committed to improving transparency in authorship. As part of our efforts in this direction, we are now requesting that all authors identified as 'corresponding author' on published papers create and link their Open Researcher and Contributor Identifier (ORCID) with their account on the Manuscript Tracking System (MTS), prior to acceptance. ORCID helps the scientific community achieve unambiguous attribution of all scholarly contributions. You can create and link your ORCID from the home page of the MTS by clicking on 'Modify my Springer Nature account'. For more

information please visit please visit www.springernature.com/orcid.

Thank you for the opportunity to review your work.

Sincerely,
Kyle

Kyle Vogan, PhD
Senior Editor
Nature Genetics
<https://orcid.org/0000-0001-9565-9665>

Referee expertise:

Referee #1: Genetics, cardiovascular diseases

Referee #2: Genetics, cardiovascular diseases

Referee #3: Genetics, cardiovascular diseases

Reviewers' Comments:

Reviewer #1:
Remarks to the Author:

Miyazawa et al present GWAS findings on atrial fibrillation (AF) in the Japanese population along with an extended trans-ethnic meta-analysis with AFGen and FinnGen AF data. Compared to previous AF GWAS in Japanese individuals (about ~8,200 AF cases and ~29,000 controls; Low et al. Nat Genet 2017) sample size is now increased to 9,826 AF cases and ~140,000 controls. With the addition of ~2,000 new cases, they identify 5 new loci on top of the 26 previously reported loci (Low et al 2017). The trans-ethnic meta-analysis identified 150 loci (33 novel; some of them overlapping with the more restricted analysis in the Japanese population). Overall, the manuscript is well-written, and the downstream in-silico analyses, while pretty standard, are state-of-the-art.

However, there are multiple points that require clarification. In order of appearance:

- For the sex-stratified GWAS, I would prefer to see a formal interaction analysis of SNP x sex using both the interaction term and an omnibus test of main effect and interaction effect. From a statistical perspective, this would be the correct way of analysing the data.
- The authors report a correlation > 1 for the comparison of male and female GWAS, which to many readers may seem impossible. Of course, this can happen in LD score regression analyses. However, seeing an explanation on why this can happen would help the reader.

- The authors state that for the lead SNP in the CUX2 locus (rs12644625) there is a difference between male ($p=10E-33$) and female ($p=10E-3$). Is this true for all SNPs at that locus or just for the lead SNP? It would be of interest to see the differences for the whole signal and not only for single SNPs.
 - For the whole sex-stratified GWAS, it would be good to state the sample sizes for male and female in the paragraph to demonstrate that the differences are not merely due to the difference in power in the two analyses.
 - For the missense variants discovered in the trans-ethnic meta-analysis, please present the r^2 value with the lead SNP in the locus.
 - The authors tried to look for AF-associated variants overlapping with quantitative trait loci and found multiple SNPs that showed associations also with QTLs. However, for a more formal analysis, a Bayesian method like coloc should be used to see whether the signals truly overlap.
 - TWAS analysis: the authors state that "the median physical distances from the lead variants to the candidate genes were greater than 100 kb. Notably, 17.4% and 25.6% of the candidate genes were located 500 kb away from the lead variants". This seems to be a lot of trans effects. The authors should compare these figures to previously published figures and put them in context.
 - The authors emphasise that their constructed AF PRS was significantly associated with an increased risk of cardiovascular death, which is pretty much expected. I suggest they also include the association with all-cause death and non-cardiovascular death as a control.
 - The authors should make sure and confirm there is no sample overlap between FinnGen and AFGen cohorts.
 - For the credible set analysis, how was a locus defined? This is missing from the methods.
 - Throughout the manuscript, the r^2 values used vary widely. For instance, in the "pleiotropy" paragraph in the methods, the authors use two different r^2 cutoffs (0.5 and 0.6 for QTLs) with no apparent rationale. The reviewer acknowledges that changing the r^2 value to an uniform value throughout the manuscript is not feasible, but the authors should explain the rationale for using a specific cutoff in the methods (even if the explanation is just because it's the default setting of the software used).
- Also, in the MR analysis, the authors use an r^2 cutoff of 0.05 which is exceptionally high (0.01 or 0.001 are used nowadays).
- In the TWAS, the authors base their findings on GTEx7 which is outdated. GTEx8 weights should be readily available for analysis and used in the updated analysis.

Reviewer #2:
Remarks to the Author:

Miyazawa and colleagues describe the results of their genome-wide association study (GWAS) meta-analyses of atrial fibrillation (AF) in the Japanese population and in a trans-ancestry population of Europeans and Japanese samples, identifying five and 35 novel loci, respectively. Downstream in silico analyses of the novel loci show several biologically and clinically relevant genes and pathways, and genetic risk scores and Mendelian randomization analyses show that AF is a (causal) risk factor for a variety of cardiovascular disorders. However, no functional (in vitro or in vivo) work is done to reveal the biological mechanisms underlying AF. The manuscript is well written and used methodologies and presented results are of high quality. Conclusions mostly flow smoothly from described results.

I have some suggestions regarding the analyses and manuscript, and would like the authors to clarify a number of points:

- 1) Five novel loci are found in the Japanese ancestry GWAS that have not been found previously. Since some of these have a low frequency in the Japanese population and are virtually absent from other populations, it is difficult to determine whether they may be false positive hits. I would suggest to use an independent (East Asian) replication dataset to confirm these novel loci. Also I would suggest to describe in the Discussion section why you believe these are true positive hits.
- 2) Sex stratified GWAS for AF: The authors describe that several loci show a statistically stronger association (i.e. much smaller P-values) in males than females. However, the study in males contains ~2 times more cases. Please incorporate this information in this section. In addition, I would be interested in the number of novel loci found among the significant loci in the sex-stratified GWAS.
- 3) Several downstream analyses use not only the index/lead variant for each locus but also SNPs in LD. However, the r^2 used to determine SNPs in LD is not consistent across the manuscript. Please explain why a specific threshold is chosen for a certain analysis.
- 4) Lines 239-243, distance between the lead SNP and potential causal gene: how do these numbers compare to what has been described in literature?
- 5) Lines 258 and 302-303: a P-value threshold of 0.05 is used while a large number of tests have been performed. How do the authors warrant the lack of multiple testing correction?
- 6) Mendelian randomization section: the authors refer to clinical causality vs. genetic causality – I would suggest to rephrase this sentence because MRs use a genetic instrument to determine the (clinically relevant) causality between traits/diseases. In addition, I wonder if the authors excluded variants with pleiotropic effects?
- 7) Discussion lines 371-373: while I agree with the authors that the PRS shows clear groups with different AF risks, the conclusion that it will help identifying individuals at risk for precision medicine is not something I can agree with.
- 8) Table 2 shows the novel loci after the trans-ancestry meta-analysis. However, if I understand it correctly, many loci are significant in the EUR samples only, which come from a study that is published few years ago. Why are these loci described as novel?
- 9) Figure 4: How do these numbers compare to a model including only the covariates/PCs? Also please note the Y-axis label contains a typo.

Minor comments:

- 1) The numbers of samples between the Japanese-only and BBJ in the trans-ancestry meta-analysis differ. Please elaborate on this difference.
- 2) Several analyses show $P < 2.2 \times 10^{-16}$. Please provide the actual P-values for these analyses.
- 3) Line 197 "739 variants": Please explain where this number comes from, is this the number of lead variants and proxies across the 150 loci?
- 4) Lines 207-209: Several clusters are mentioned but I have difficulties to understand why the authors focus specifically on these.
- 5) Line 229: The way this sentence is currently phrased it seems like the mouse knockdown was part of the current study, while this is not the case.
- 6) Line 251 TWAS association between AF and IL6R: Please include a bit more information, such as variant, P-value.
- 7) Lines 315-317: Low numbers of heart failure deaths are provided as potential reason for not finding a significant effect. How do these numbers compare to CVD and stroke deaths (which both show a significant association)?
- 8) Methods line 431: this sentence described loci instead of variants being in LD (or not). It is unclear to me what the authors mean with this, please clarify.
- 9) Credible set analysis line 488: please explain why you did not use FIN + BBJ
- 10) PRS methods section: It is not entirely clear to me how many PRS were constructed. Is it 6 subgroups x 5 P-value thresholds x 3 r^2 thresholds = 90, or many more because different subsets of samples were used for test/validation purposes?
- 11) I noticed that throughout the manuscript the authors use 10 or 20 principal components. How were they chosen and why are different numbers used?
- 12) Figure 5: panels c and e contain colors (grey vs red) that seem to be significant vs non-significant results but this is not specified.

Reviewer #3:

Remarks to the Author:

A. Summary of key results

The authors report results from the largest GWAS of atrial fibrillation (AF) ever performed in the

Japanese population, including 9826 cases and 150272 controls. The GWAS identified 5 new susceptibility loci. A trans-ancestry meta-analysis including published GWAS for a total of 77690 AF cases identifies a total of 35 novel loci. The investigators then follow-up on the GWAS finding using standard annotation of loci and identify several new genes of potential interest. No functional studies are reported. A polygenic risk score (PRS) is then derived from the trans-ancestry meta-analysis and is shown to be associated with an increased risk of long-term cardiovascular and stroke mortality, and segregated individuals with cardioembolic stroke in undiagnosed AF patients.

B. Originality and significance

Although prior larger GWAS of AF have been reported, this work reports the largest GWAS in Japanese, and should as such be considered novel given the importance of diversity in genetics research especially with regards to potential clinical use. The trans-ancestry meta-analysis with the current sample size is novel.

C. Data & methodology

- Methodology. Generally very appropriate and well described, with the following minor exceptions:

-- Assuming sample independence, the authors should test the PRS of Khera et al (PMID 30104762; https://personal.broadinstitute.org/ryank/AtrialFibrillation_PRS_LDpred_rho0.003_v3.zip) in their Japanese testing cohort. How does it perform in the Japanese population, compared to the European testing set reported by Khera et al. ? Does the PRS derived from the current Japanese GWAS perform better?

-- PRS derivations have been performed using classic thresholding and pruning. The authors can consider also using LDpred or LDpred2 for PRS derivation.

-- The association of PRS with hemorrhagic stroke is interesting. What is the mechanism? Can it be mediated by anticoagulant therapy ? Did the authors correct for antithrombotic drug use?

-- How are gene names selected for locus identification in Tables 1 and 2 and Figure 1? More generally, the authors should consider using opentarget to map locus to genes in addition to the described annotation.

-- In the trans-ancestry meta-analysis, the authors included the GWAS of Nielsen et al (PMID 30061737) but not of Roselli et al (PMID 29892015). The latter publication reported a meta-analysis of both Roselli and Nielsen including non-overlapping samples. Do the authors have access to this latter larger meta-analysis summary statistics and can it be included? More importantly, did the authors make sure the 35 novel loci were not identified by this combined meta-analysis of Roselli and Nielsen (supplementary table 16 in PMID 29892015: https://www.ncbi.nlm.nih.gov/pmc/articles/PMC6136836/bin/NIHMS986675-supplement-Supplementary_Table_16.xlsx).

- Data presentation should be improved, as follows:

-- The authors should be more concise in the results section and consider modifying the manuscript to a letter format. Important findings seem to be flooded by less interesting results and lengthy descriptions. Specifically, the following 3 sections should be shortened or shifted to the supplement: "Sex-stratified GWAS for AF", "Shared allelic effects between Japanese and European populations and fine mapping/credible set analyses in the trans-ancestry meta-analysis", "Pleiotropic effects and functional pathways of AF-associated loci"

-- Figure 2d: Are the P-values corrected for multiple testing? It seems strengths of association are otherwise very modest ($-\log_{10} P$ ranging from 1.5 to 2). Move to supplement?

-- Figure 4: Describe what the outcome is in the figure and/or legend.

-- Figure 5b as represented here is not easy to understand. If it adds to panel A, then please describe further and clarify visually. If it does not add to panel A, simply remove or move to supplement.

D. Appropriate use of statistics and treatment of uncertainties

No further comment

E. Conclusions: robustness, validity, reliability

The discussion section is rather short compared to the results section. I suggest shortening the results section (or modifying to a letter format for conciseness)

No further comment

F. Suggested improvements: experiments, data for possible revision

No further comment

G. References: appropriate credit to previous work?

Appropriate.

H. Clarity and context: lucidity of abstract/summary, appropriateness of abstract, introduction and conclusions

Good abstract. Suggest moving supplementary figure 1 (Overview of the study design) to the main text.

Author Rebuttal to Initial comments

See Attached PDF

**Dear Editor and Reviewers,**

We are grateful for the opportunity to submit our revised manuscript. We appreciate the editor and the
reviewers for providing us with such high-quality review comments and a chance for revision. We
have substantially revised our manuscript according to the comprehensive and constructive feedback.
We believe these changes make our study clearer, more robust, and more informative. We hope you
will find our revised manuscript suitable for publication in Nature Genetics. The major changes in the
revised manuscript and point-by-point responses to all comments from the reviewers as given below.

**Major changes**

**1. Replication study for newly identified loci in the Japanese population**

As suggested by the Editor and Reviewer #2, we conducted a replication study for the five association
signals newly identified in our current Japanese GWAS. Fortunately, we had access to an
independent Japanese dataset the BioBank Japan (BBJ) second cohort, which comprised DNA
samples and clinical information of approximately 60,000 new patients with the 38 target diseases
collected between 2013 and 2018 to expand research outcomes from the BBJ first cohort. We then
performed association analyses leveraging 48,677 samples (4,602 cases and 44,075 controls) and
confirmed that all of the association signals were successfully replicated with nominal association ($P <$
0.05) in the same direction (**Supplementary Table 2**).

**2. Transcriptome-wide association analysis (TWAS) with the latest database**

Based on the reviewers' comments, we re-performed TWAS using GTEx v8, the latest release of the
tissue-specific gene expression database, which resulted in substantial increases in the number of
candidate genes associated with AF from 86 to 132 in the atrial appendage and from 74 to 127 in the
left ventricle (**Fig. 2a**). Furthermore, to more precisely assess the *cis*- and *trans*-effect of variants on
candidate genes, we calculated physical distances from an AF-associated variant included in the
prediction model to the transcription start site of the canonical transcript for each candidate gene.
Thereby, we found that the median physical distances were 2.25 kb and 1.14 kb in the atrial
appendage and left ventricle, respectively (**Fig. 2c**), which showed the shorter-acting effects of AF-
associated variants compared to those in our previous results, while they were comparable to the
results from a previous study.

**3. Analyses of AF-associated transcription factor**

During revision, when we checked the ChIP-seq experiment (SRX4003759), which was the basis for
the ERG identified as a result of the transcription factor-enrichment analysis using the ChIP-Atlas
dataset, we found that the experiment reported the result of the human induced pluripotent stem cell-
derived cardiomyocytes (iPSCMs) experiment for ERRg, but not ERG. Thus, we reached the
developer of the ChIP-Atlas dataset and confirmed that ERG assigned to SRX4003759 was their
curation error of ERRg. Therefore, to confirm that the result from the enrichment analysis using the
ChIP-Atlas dataset was truly reliable, we added several functional experiments of ERRg using
iPSCMs; (1) We showed that the expression levels of ion channel and sarcomere genes associated
with AF were significantly reduced after treatment with an inverse agonist of ERRg (GSK5182) (**Fig.**

**4b**). (2) GSK5182-treated iPSCMs showed irregular rhythm and prolonged contraction time,
consistent with the mechanism of AF pathogenesis (**Fig. 4c,d**). (3) GSK5182-treated iPSCMs also
showed prolonged calcium transient duration and attenuated isoproterenol-induced tachycardia (**Fig.**
**4e,f**). These results suggest that ERRg is an important transcriptional regulator in the pathogenesis of
AF.

**Other changes**

We performed additional analyses for the sex-stratified GWAS, colocalization, and PRS according to
the reviewers' comments, which supported the conclusions obtained from our previous analyses. As
Reviewer #3 suggested, however, our important results were buried in less interesting results and
lengthy descriptions. Therefore, we decided to shorten the result section by moving some analyses to
Supplementary Notes, such as the sex-stratified GWAS, comparison of variant effects among
ancestries, 99% credible set analysis, pleiotropic analysis using colocalization, and tissue and gene-
sets enrichment analysis. We instead added a replication analysis for newly identified loci and
functional analyses for AF-associated transcription factors, and also improved the TWAS and PRS
analyses with a more precise assessment. We believe these alterations now clearly highlight the
pivotal results of our study in the manuscript.

**Referee expertise:**

**Referee #1: Genetics, cardiovascular diseases**

**Referee #2: Genetics, cardiovascular diseases**

**Referee #3: Genetics, cardiovascular diseases**

We thank all the reviewers for taking considerable time to provide high-quality comments on our
manuscript.

**Reviewers' Comments:**

**Reviewer #1:**

**Remarks to the Author:**

**Miyazawa et al present GWAS findings on atrial fibrillation (AF) in the Japanese population**
**along with an extended trans-ethnic meta-analysis with AFGen and FinnGen AF data.**
**Compared to previous AF GWAS in Japanese individuals (about ~8,200 AF cases and ~29,000**
**controls; Low et al. Nat Genet 2017) sample size is now increased to 9,826 AF cases and**
**~140,000 controls. With the addition of ~2,000 new cases, they identify 5 new loci on top of the**
**26 previously reported loci (Low et al 2017). The trans-ethnic meta-analysis identified 150 loci**
**(33 novel; some of them overlapping with the more restricted analysis in the Japanese**
**population). Overall, the manuscript is well-written, and the downstream in-silico analyses,**
**while pretty standard, are state-of-the-art. However, there are multiple points that require**
**clarification. In order of appearance:**

We appreciate your careful assessment of our manuscript and are grateful for the numerous
constructive comments. Accordingly, we revised our manuscript to enhance its validity and clarity.
The following is a point-by-point response to each of your comments.

**Reviewer #1, Major comment #1**

**For the sex-stratified GWAS, I would prefer to see a formal interaction analysis of SNP x sex**
**using both the interaction term and an omnibus test of main effect and interaction effect. From**
**a statistical perspective, this would be the correct way of analysing the data.**

We thank you for the comment and agree with this appropriate advice. Accordingly, we performed an
omnibus test using models with and without an interaction term of SNP × sex to detect the interaction
effect on AF development. Chi-square statistics were calculated from the omnibus test on AF-
associated variants with heterogeneity in the meta-analysis of male- and female-GWAS. As a result,
we identified variants with a significant interaction effect with sex in the *PITX2-C4orf32* and *CUX2* loci
(Supplementary Fig. 12c); this supports the contribution of two loci to the gender difference in AF
development.

The corresponding revised text and figure are shown below.

(Lines 33 – 37 in Supplementary Notes)

**Additionally, we performed an interaction analysis to investigate a SNP × sex effect, in which we**
**performed an omnibus test for genome-wide significant variants with heterogeneity between sexes**
**using regression models with or without the interaction term. As a result, we identified variants with a**
**significant interaction effect with sex in the *PITX2-C4orf32* and *CUX2* loci (Supplementary Fig. 12c**
**and Supplementary Table 15).**

(Lines 131 – 136 in Supplementary Notes)

In the interaction analysis, we tested for the interaction effect of a SNP × sex on AF development,
where we performed an omnibus test using models with and without the interaction term. Chi-square
statistics were calculated from the omnibus test. Since limiting to variants with a strong heterogeneity
($P_{\text{het}} < 0.0001$) may lead to some association signals being missed, we analyzed genome-wide
significant variants with a relaxed heterogeneity threshold ($P < 0.05$).

(Supplementary Fig. 12c)

**Supplementary Fig. 12 | Sex-stratified GWAS. c.** The results of the omnibus test in the *PITX2-*
*C4orf32* (left) and *CUX2* loci (right), respectively. The y axis represents $-\log_{10} P$ values obtained from
the omnibus test for AF-associated variants with heterogeneity between sexes, showing against the
genomic position of chromosomes 4 (left) and 12 (right) on the x axis. The red line indicates a
significant threshold adjusted for the number of variants tested ($P = 0.05/399$ for the *PITX2-C4orf32*
locus and $P = 0.05/176$ for the *CUX2* locus).

**Reviewer #1, Major comment #2**

**The authors report a correlation > 1 for the comparison of male and female GWAS, which to**
**many readers may seem impossible. Of course, this can happen in LD score regression**
**analyses. However, seeing an explanation on why this can happen would help the reader.**

Thank you for raising this important point. Since LD score regression is not a bounded estimator, it
can produce estimates outside of $[-1, 1]$ due to sampling variation, e.g., sample sizes and heritabilities
of the GWASs. In our analysis, the genetic correlation between sexes was indeed slightly above 1
(1.095), but it does not mean that our estimation is inaccurate. Therefore, we mentioned the
estimation of genetic correlation using LD score regression in the context. Additionally, we performed
bivariate GREML analysis (<http://cnsgenomics.com/software/gcta/#BivariateGREMLanalysis>)
implemented in GCTA software and it also showed a high genetic correlation between sexes ($r_g =$
0.934 , s.e.m. = 0.076). This result also supports the evidence of the high genetic correlation between
males and females.

We added the following text in Supplementary Notes.

(Lines 7 – 12 in Supplementary Notes)

Thus, we divided our samples into males and females, and performed GWASs separately. First,
to explore the difference in genetic susceptibility to AF between males and females, we
calculated the genetic correlation, which was 0.934 (s.e.m. 0.076) by GCTA-GREML⁶ and 1.095
(s.e.m. 0.112) by LD-score regression⁷, respectively. This result suggests that genetic influences
on AF development are strongly shared between males and females.

(Lines 118 – 123 in Supplementary Notes)

Then, we performed the genetic correlation analysis using LD score regression⁷. As LD-score
regression may estimate a genetic correlation beyond the range of -1 to 1 depending on
sampling variation or disease heritability, we added a bivariate GREML analysis
(<http://cnsgenomics.com/software/gcta/#BivariateGREMLanalysis>) implemented in GCTA
software⁶.

**Reviewer #1, Major comment #3**

**The authors state that for the lead SNP in the CUX2 locus (rs12644625) there is a difference**
**between male (p=10E-33) and female (p=10E-3). Is this true for all SNPs at that locus or just for**
**the lead SNP? It would be of interest to see the differences for the whole signal and not only**
**for single SNPs.**

Thank you for pointing out this important issue. Because of LD structure, it is true that we need to
compare the locus as a region. First, we would like to apologize that rsID of the SNP in the *CUX2*
locus is a typo of rs3809297. Second, in the meta-analysis of male- and female-GWAS, we observed
only a single genome-wide significant SNP (rs3809297) that showed a strong heterogeneity ($P_{\text{het}} =$
2.802×10^{-5}) in the *CUX2* locus, although there are multiple genome-wide significant SNPs with
slightly below strong heterogeneity threshold ($P_{\text{het}} > 1.0 \times 10^{-4}$) (Response Fig. 1). Therefore, we
performed an interaction analysis on genome-wide significant SNPs with a heterogeneity threshold
level of 0.05. Among two loci, we identified multiple variants with significant interaction of SNP \times sex
(Supplementary Table 15).

Below are the corresponding revised text and table.

(Lines 131 – 136 in Supplementary Notes)

In the interaction analysis, we tested for the interaction effect of a SNP \times sex on AF development,
where we performed an omnibus test using models with and without the interaction term. Chi-square
statistics were calculated from the omnibus test. Since limiting to variants with a strong heterogeneity
($P_{\text{het}} < 0.0001$) may lead to some association signals being missed, we analyzed on genome-wide
significant variants with a relaxed heterogeneity threshold ($P < 0.05$).

[Supplementary Table 15 (partially extracted)]

Supplementary Table 15 | Genome-wide significant variants with the evidence of heterogeneity and interaction of sex in the sex-stratified GWAS

CHR	POS	rsID	REF	ALT	AAF	Beta	SE	Association P	Heterogeneity P	Interaction P
PITX2-C4orf32 locus										
4	111587243	rs78582666	G	T	0.140	-0.404	0.028	5.14E-48	2.03E-04	1.03E-04
4	111605325	rs77507205	C	T	0.130	-0.408	0.029	1.71E-45	1.93E-05	7.95E-06
4	111621597	rs2122078	T	C	0.564	0.296	0.016	6.73E-79	1.76E-04	1.09E-04
4	111623276	rs12650829	G	A	0.232	0.298	0.017	8.56E-69	8.73E-06	4.46E-06
4	111624340	rs777109930	T	TTA	0.565	0.297	0.016	1.69E-79	1.28E-04	8.97E-05
4	111625464	rs1375302	C	T	0.436	-0.296	0.016	2.60E-79	1.41E-04	1.12E-04
4	111625465	rs1375303	A	G	0.565	0.296	0.016	2.59E-79	1.41E-04	1.12E-04
4	111627356	rs2595093	A	G	0.290	0.315	0.016	2.31E-86	5.99E-06	4.12E-06
4	111627380	rs75752846	C	T	0.233	0.298	0.017	7.23E-69	7.29E-06	4.68E-06
4	111627489	rs2723313	A	G	0.568	0.292	0.016	5.35E-77	1.29E-04	1.06E-04
CUX2 locus										
12	110297784	rs117203896	C	T	0.091	-0.183	0.032	7.26E-09	1.53E-03	2.79E-04
12	110342598	rs74416240	G	A	0.122	-0.206	0.028	8.49E-14	1.76E-03	1.93E-04
12	110343157	rs16940666	C	T	0.122	-0.206	0.028	8.40E-14	1.76E-03	1.82E-04
12	110343901	rs79088408	A	G	0.878	0.206	0.028	8.31E-14	1.76E-03	1.83E-04
12	110360321	rs16940688	G	A	0.116	-0.207	0.028	2.91E-13	2.05E-04	2.59E-05
12	110390979	rs925368	T	C	0.884	0.209	0.029	2.32E-13	1.99E-04	1.57E-05
12	110480809	rs142915423	C	T	0.180	-0.173	0.024	7.63E-13	1.48E-03	1.49E-04
12	110582338	rs141965732	C	T	0.129	-0.226	0.028	7.31E-16	2.69E-04	8.07E-05
12	110631881	rs117624317	C	T	0.135	-0.227	0.028	2.48E-16	1.86E-03	2.80E-04
12	110675363	rs144825998	C	T	0.129	-0.229	0.028	4.43E-16	2.48E-04	6.54E-05

(Response Fig. 1)

Response Fig. 1 | The CUX2 locus in the sex-stratified GWAS. a,b, The results of association study in the CUX2 locus from the sex-stratified GWAS. The $-\log_{10} P$ -values on the y axis are shown against the genomic position of chromosome 12 on the x axis. The red line indicates a genome-wide significant threshold ($P = 5 \times 10^{-8}$) for the association P -value (a), and a heterogeneity threshold ($P = 0.0001$) for the strong heterogeneity P -value (b). The blue line indicates a heterogeneity threshold with a nominal P -value ($P = 0.05$) (b).

**Reviewer #1, Major comment #4**

**For the whole sex-stratified GWAS, it would be good to state the sample sizes for male and**
**female in the paragraph to demonstrate that the differences are not merely due to the**
**difference in power in the two analyses.**

We apologize for not showing this essential information. In the sex-stratified GWAS, the overall
sample sizes were different. We therefore wrote the numbers in the manuscript. Additionally, to
assess the impact of the sample size on the heterogeneity in the *PITX2-C4orf32* and *CUX2* loci, we
calculated the effective sample sizes with sufficient statistical power to detect genome-wide significant
signals in the sex-stratified GWAS using the Genetic Association Study Power Calculator provided by
the University of Michigan (https://csg.sph.umich.edu/abecasis/gas_power_calculator/), under the
following conditions: a significance level of $P < 5 \times 10^{-8}$ and disease prevalence in the Japanese
population of 0.82% (<https://vizhub.healthdata.org/gbd-compare/>). We used the input data of the lead
variant in the two loci, such as allele frequency and genotype relative risk (Supplementary Table 14).
To achieve 80% power, the effective sample size for the *PITX2-C4orf32* locus was approximately
450, while the *CUX2* locus required 3,000 cases, which shows that we have enough sample sizes
with sufficient statistical power to detect the association signals of the two loci in our analyses.
Accordingly, this result also supports the result that the different association signals in the *PITX2-*
*C4orf32* and *CUX2* loci were due to the genetic heterogeneity between sexes rather than the mere
difference in the sample sizes.

Below are the corresponding texts and table in the revised manuscript.

(Lines 24 – 33 in Supplementary Notes)

Since the sample sizes in the sex-stratified GWAS were different (6,825 cases in the male-GWAS and
3,001 cases in the female-GWAS), we calculated the effective sample sizes with sufficient statistical
power using the Genetic Association Study Power Calculator to examine whether the heterogeneity
between sexes in these two loci was due to the difference in the sample sizes. To achieve 80%
power, the effective sample size for the *PITX2-C4orf32* locus was approximately 450, while the *CUX2*
locus required 3,000 cases. This result shows that the sex-stratified GWAS has enough sample size
with sufficient statistical power to detect the significant association signals of the two loci
(Supplementary Table 14) and supports the genetic heterogeneity between sexes in the two loci
rather than the mere difference in the sample sizes.

(Lines 124 – 131 in Supplementary Notes)

To calculate the effective sample sizes with sufficient statistical power, we used the Genetic
Association Study Power Calculator provided by the University of Michigan
(https://csg.sph.umich.edu/abecasis/gas_power_calculator/), under the following conditions: a
significance level of $P < 5 \times 10^{-8}$ and disease prevalence in the Japanese population of 0.82%
(<https://vizhub.healthdata.org/gbd-compare/>). We also used the allele frequencies and genotype
relative risks of the lead variants in the *PITX2-C4orf32* and *CUX2* loci as an input data

(Supplementary Table 14). The effective sample size was defined as the minimum number of
 samples to achieve 80% power.

 [Supplementary Table 14]

Supplementary Table 14 | Power calculation by GAS Power Calculator

PITX2-C4orf32 locus												
CHR	Start	End	Lead variant						N _{case}		Effective sample size for 80% statistical power	
			CHR	POS	rsID	REF	ALT	AAF	OR	Male-GWAS	Female-GWAS	
4	111467478	112007484	4	111716513	rs12644625	C	T	0.455	1.766	6,825	3,001	450

CUX2 locus												
CHR	Start	End	Lead variant						N _{case}		Effective sample size for 80% statistical power	
			CHR	POS	rsID	REF	ALT	AAF	OR	Male-GWAS	Female-GWAS	
12	110153675	114812626	12	111609727	rs3809297	G	T	0.235	0.7556	6,825	3,001	3,000

 **Reviewer #1, Major comment #5**

**For the missense variants discovered in the trans-ethnic meta-analysis, please present the r²**
 **value with the lead SNP in the locus.**

We apologize for missing the important information. We have filled in the r² values in Supplementary
 Table 5. As we mention in your additional comment #11, we unified r² value of 0.8 in the present
 study to avoid ambiguous interpretation of the results. Therefore, we extracted missense variants that
 fell within the range of r² > 0.8, instead of 0.6.

[Supplementary Table 5 (partially extracted)]

Supplementary Table 5 | Missense variants in linkage disequilibrium with the lead variants detected in trans-ancestry meta-analysis

CHR	POS	rsID	REF	ALT	Gene	r ²	Lead SNP				Functional impact				
							CHR	POS	rsID	SIFT score	SIFT pred	Polyphen2 HDIV score	Polyphen2 HDIV pred	CADD raw	CADD pred
Novel loci															
1	16255644	rs848208	C	T	SPEN	0.989	1	16199051	rs9782984	0.096	Tolerated	0.006	Benign	-0.317	0.608
4	83838262	rs897945	G	T	THAP9	0.964	4	83910712	rs6841049	0.499	Tolerated	0.906	Possibly damaging	-0.478	0.250
20	36841914	rs3746471	G	A	KIAA1755	1.000	20	36841914	rs3746471	0.059	Tolerated	0.003	Benign	2.905	21.900
Known loci															
2	179397561	rs3829747	C	T	TTN	0.992	2	179413110	rs3731748	0	Deleterious	1.000	Probably damaging	4.497	24.300
2	179457147	rs16866406	G	A	TTN	0.992	2	179413110	rs3731748	0.167	Tolerated	0.008	Benign	2.159	17.240
3	12848822	rs11718898	T	C	CAND2	0.932	3	12841804	rs7850482	0.502	Tolerated	0.000	Benign	1.109	11.260
3	38766675	rs6795970	A	G	SCN10A	0.959	3	38767315	rs6801957	1	Tolerated	0.000	Benign	-0.943	0.021
3	89521693	rs35124509	T	C	EPHA3	0.927	3	89489529	rs6771054	0.229	Tolerated	0.000	Benign	2.291	18.100
7	881668	rs6461378	C	T	SUN1	0.841	7	895285	rs6461461	0.736	Tolerated	0.945	Possibly damaging	3.431	23.000
10	75406912	rs34163229	G	T	SYNPO2L	0.967	10	75414344	rs60212594	0.02	Deleterious	0.986	Probably damaging	2.766	21.200
10	75415677	rs60632610	C	T	SYNPO2L	0.983	10	75414344	rs60212594	0.02	Deleterious	0.997	Probably damaging	6.741	32.000
10	75759483	rs12217245	T	C	CAMK2G	0.844	10	75414344	rs60212594	NA	NA	NA	NA	NA	NA
12	57109792	rs2958149	A	G	NACA	1.000	12	57105938	rs2860482	1	Tolerated	0.000	Benign	1.812	15.080
12	57114100	rs2826743	A	G	NACA	1.000	12	57105938	rs2860482	0.423	Tolerated	0.000	Benign	-1.298	0.005
13	113909339	rs2302757	A	G	CUL4A	0.983	13	113864837	rs2316443	0.204	Tolerated	0.009	Benign	1.207	11.790
17	38028634	rs11557467	G	T	ZBP2	0.984	17	38049589	rs7359623	0.045	Deleterious	0.295	Benign	1.537	13.510
17	38062196	rs2305480	G	A	GSDMB	0.829	17	38049589	rs7359623	0.898	Tolerated	0.877	Possibly damaging	-1.352	0.004
17	38062217	rs2305479	C	T	GSDMB	0.960	17	38049589	rs7359623	0.129	Tolerated	1.000	Probably damaging	3.505	23.100
22	26159289	rs133885	G	A	MYO18B	0.892	22	26164079	rs133902	-1	Tolerated	0.000	Benign	1.100	11.220

 **Reviewer #1, Major comment #6**

**The authors tried to look for AF-associated variants overlapping with quantitative trait loci and**
 **found multiple SNPs that showed associations also with QTLs. However, for a more formal**
 **analysis, a Bayesian method like coloc should be used to see whether the signals truly**
 **overlap.**

Thank you for raising this point. We agree with the reviewer's opinion. Accordingly, we performed
 colocalization analyses to assess whether AF-associated loci and QTL shared a causal variant. We
 extracted QTL data based on 150 AF-associated loci identified by the trans-ancestry meta-GWAS,
 and colocalization analysis was performed for each locus using Coloc.abf from the Coloc R package¹.
 The threshold of a posterior probability > 0.8 was applied for significant colocalization by referring to a
 previous study², and we then identified 56 pairs of AF-associated loci and QTL with significant
 colocalization (Supplementary Figure 16 and Supplementary Table 18).

We added the following descriptions, a figure, and a table in the revised manuscript.

(Lines 90 – 97 in Supplementary Notes)

In particular, to explore the biological pathways related to AF development, we assessed
colocalization of AF-associated loci with quantitative trait loci (QTL) for clinical measurements such as
anthropometric, metabolic, kidney-related, and blood pressure data. Colocalization analysis was
performed per 150 AF-associated loci identified by the trans-ancestry meta-GWAS, and we identified
25 AF-associated loci that showed significant colocalization with QTL (Supplementary Figure 16). Of
56 pairs of AF-associated loci and QTL with significant colocalization, 19.6% (11/56) were kidney-
related traits, 16.0% (9/56) were blood pressure-related, and 16.0% (9/56) were metabolic traits
(Supplementary Table 18).

(Lines 183 – 186 in Supplementary Notes)

We extracted QTL data based on 150 AF-associated loci identified by the trans-ancestry meta-
GWAS, and colocalization was performed for each locus using Coloc.abf from the Coloc R package²⁶.
The threshold of a posterior probability > 0.8 was applied for significant colocalization by referring to a
previous study²⁷.

(Supplementary Figure 16)

**Supplementary Fig. 16 | Colocalization of quantitative trait loci with AF-associated signals.**

Heat-map representation of the approximate Bayes factor posterior probability of AF and quantitative
trait loci to share a common causal variant at 150 AF-associated loci (coefficient H4 of the
colocalization analysis, represented on a white-red color scale). The rows show 36 quantitative traits
highlighted in each color of the 9 categories, and the columns show the chromosome and physical
position of the lead variant in AF-associated loci. AF, atrial fibrillation; QTL, quantitative trait loci.

[Supplementary Table 18 (partially extracted)]

Supplementary Table 18 Colocalization of AF-associated loci with QTL for clinical measurement						
Lead SNP			Locus		Phenotype	Posterior probability
CHR	POS	rsID	Start	End		
Novel loci						
1	16199051	rs9782984	15699051	16699051	AST	0.893
5	139703286	rs17118812	139032339	140214751	CK	0.940
					eGFR	0.876
6	76164589	rs12209223	75664589	77067954	Height	0.992
11	3890059	rs7126870	3389383	4390223	HbA1c	0.860
12	12886027	rs10845620	12386027	13386027	HDL-C	0.993
18	77156537	rs8096658	76656537	77656537	BUN	0.999
					eGFR	0.984
					sCr	0.997
					UA	0.837
19	48142746	rs11881441	47628418	48670757	Height	0.874
22	21999229	rs5754508	21499229	22499229	GGT	0.836
					HDL-C	0.898
					RBC	0.823
Known loci						
1	10796866	rs880315	10280727	11302468	AST	0.961
					DBP	0.999
					K	0.998
					MAP	0.999
					Na	0.975
					PP	0.999
2	61676940	rs2694635	61071295	62306999	WBC	0.841
2	65284231	rs2540949	64735333	65886462	PP	0.908
3	12841804	rs7650482	12259080	12841804	CK	0.995

**Reviewer #1, Major comment #7**

**TWAS analysis: the authors state that “the median physical distances from the lead variants to**
 **the candidate genes were greater than 100 kb. Notably, 17.4% and 25.6% of the candidate**
 **genes were located 500 kb away from the lead variants”. This seems to be a lot of trans**
 **effects. The authors should compare these figures to previously published figures and put**
 **them in context.**

Thank you for pointing out this issue. As you mentioned, the proportion of trans-effects of AF-
 associated variants was much higher than we expected, although our results are consistent with
 those of a previous study by Roselli et al. in 2018³, in which they performed a large-scale study for the
 same disease and 26% of candidate genes detected by TWAS were more than 500 kb apart from the
 lead variants. However, given the closer relationship between the non-coding variants and the target
 genes in complex diseases, which was reported in Nasser et al.’s study published in *Nature* 2021⁴,
 we thought that we should assess more precise cis- or trans-effect of AF-associated variants on
 target genes. Previously, we calculated a physical distance from a lead variant to the candidate gene,
 but the lead variants were not necessarily included in the prediction models of gene expression.
 Indeed, the gene expression prediction model, GTEx PredictDB⁵, which was used in our present
 analysis, included cis-SNPs within 1 million bp upstream of the transcription start site and 1 million bp
 downstream of the transcription end site. Therefore, we first extracted an AF-associated variant with
 the lowest association *P* value in the prediction model for each candidate gene, and then calculated

the distances from the variant to the transcription start site of the canonical transcript for each
 candidate gene. As a result, we observed that the median physical distances were 2.25 kb and 1.14
 288 kb in the atrial appendage and left ventricle, respectively (Fig. 3c), which demonstrated shorter-acting
 effects of AF-associated variants compared to those in our previous results, while they were
 comparable to the results from the Nasser et al.'s study⁴.

Below are the corresponding texts and figures in the revised manuscript.

(Lines 141 – 151 in the main manuscript)

Furthermore, to assess the cis- and trans-effect of AF-associated variants on the candidate genes, we
 calculated the physical distances from the variants to the transcription start site (TSS) of the
 candidate genes. Only 34 and 35 genes overlapped between the nearest genes to the lead variants
 and the candidate genes identified by TWAS (Fig. 3b), and the median physical distances were 2.25
 297 kb and 1.14 kb (Fig. 3c) in the atrial appendage and left ventricle, respectively. This relationship
 between the AF-associated variants and the candidate genes is comparable to a previous study in
 which the distances from the non-coding GWAS signals to the target genes were assessed based on
 chromatin state and three-dimensional contacts¹⁸. Exceptionally, only one gene, *FBN2*, was more
 than 500 kb away from the variant in the left ventricle (512 kb). This result indicates that, although
 disease-associated genes are not necessarily closest to the lead variants, the majority of candidate
 genes are affected by the cis-effect of AF-associated variants.

(Lines 448 – 451 in the main manuscript))

To assess the relationship between AF-associated variants and the candidate genes, we first
 extracted an AF-associated variant with the lowest association *P* value among those included in the
 prediction model obtained from GTEx PredictDB⁴¹ for each candidate gene, and then calculated the
 physical distances from the variant to the TSS of the canonical transcript for each candidate gene.

(Fig. 3b,c)

**Fig. 3 | Transcriptome-wide association analysis.** **b**, Number of genes located closest to the lead
 variants (red), identified by TWAS (blue), and overlapped between them (green) in atrial appendage
 and left ventricular tissues. **c**, Distribution of physical distances from the AF-associated variant

included in the prediction model to the TSS of the canonical transcript for each candidate gene
identified by TWAS in atrial appendage (left) and left ventricular tissues (right).

**Reviewer #1, Additional comment #8**

**The authors emphasise that their constructed AF PRS was significantly associated with an**
**increased risk of cardiovascular death, which is pretty much expected. I suggest they also**
**include the association with all-cause death and non-cardiovascular death as a control.**

We apologize for our ambiguous presentation and thank you for your constructive suggestion. Whilst
we showed an association between AF-PRS and all-cause mortality and death from several
categories of cardiovascular disease, the results of the association analysis between AF-PRS and
long-term follow-up data were not clearly presented in our manuscript. Therefore, we first classified
mortality data into three categories: (1) overall mortality as a primary outcome, (2) cardiovascular and
non-cardiovascular death as a secondary outcome, and (3) each disease death in cardiovascular
disease as a tertiary outcome. Then, we examined the association of AF-PRS with each category
separately, and also modified the layout of the corresponding figure for clarity. As a result, there was
no significant association, but a trend was observed between AF-PRS and all-cause death. The
secondary outcome indicates that this trend was highly specific to cardiovascular death, which was
significantly associated with AF-PRS. Furthermore, the tertiary outcome demonstrated that, among
the causes of cardiovascular deaths, stroke showed a leading impact on the association between AF-
PRS and cardiovascular deaths.

We added the corresponding texts and modified Fig. 6d in the revised manuscript.

(Lines 220 – 228 in the main manuscript)

Moreover, Cox regression analysis was performed, and as shown in Fig. 6d, no significant association
between AF-PRS and all-cause death was found, but a trend was observed (hazard ratio [HR] per 1
standard deviation [s.d.] of PRS = 1.02, 95% CI = 0.99 – 1.04, $P = 0.13$). The secondary outcome
indicates that this trend was highly specific to cardiovascular death, which was significantly
associated with AF-PRS (HR [95% CI] = 1.06 [1.02 – 1.11], $P = 4.4 \times 10^{-3}$ for cardiovascular disease;
HR [95% CI] = 1.00 [0.98 – 1.01], $P = 0.50$ for non-cardiovascular disease). Furthermore, the tertiary
outcome suggests stroke death as a leading factor that impacts the association between AF-PRS and
cardiovascular deaths (HR [95% CI] = 1.13 [1.04 – 1.22], $P = 2.7 \times 10^{-3}$).

(Lines 562 – 566 in the main manuscript)

The causes of death were classified into three categories according to ICD-10 codes: (i) primary
outcome for all-cause death, (ii) secondary outcome for cardiovascular death (I00–I99) and non-
cardiovascular death (not I00–I99), and (iii) tertiary outcome for heart failure death (I50), ischemic
heart disease death (I20–I25), and stroke death (I60, I61, I63, and I64).

(Fig. 6d)

**Reviewer #1, Additional comment #9**

**The authors should make sure and confirm there is no sample overlap between FinnGen and**
 **AFGen cohorts.**

Thank you for raising the important point. We used the summary statistics from a previous study by
 Nielsen et al. published in 2018 *Nature Genetics*⁶. According to their paper, the samples in the AFGen
 consortium were derived from another paper by Christophersen et al. in 2017 *Nature Genetics*⁷, which
 did not include the FinnGen cohort. Additionally, we confirmed that in the paper by Larsson et al. in
 2020 *Diabetologia*⁸ mentioned no sample overlap between AFGen and FinnGen.

**Reviewer #1, Additional comment #10**

**For the credible set analysis, how was a locus defined? This is missing from the methods.**

We sincerely apologize for the lack of a detailed explanation regarding the credible set analysis. In
 this analysis, we first focused on 150 AF-associated loci identified by the trans-ancestry meta-GWAS,
 and for each locus, the posterior probabilities (PP) of individual variants were calculated using Bayes
 factors obtained from MANTRA results. Then, the 99% credible set was constructed by adding
 variants in the order of decreasing PP, starting from a lead variant, where the sum of PP included in the
 region is 99%.

Below is the corresponding text.

(Lines 147 – 153 in Supplementary Notes)

To identify sets of variants that likely include causal variants, we constructed a 99% credible set **per**
 **150 AF-associated loci identified by the trans-ancestry meta-analysis**. For each locus, we calculated
 the posterior probability (PP) for the *j*th SNP **using BF obtained from MANTRA results in the trans-**

**ancestry meta-analysis and** the following formula: $PP_j = \frac{BF_j}{\sum_k BF_k}$, where BF_j denotes the BF for the j^{th}
SNP and BF_k denotes all of the variants included in the locus. We then constructed the 99% credible
set **by adding the variants in the order of decreasing PP, starting from a lead variant, where the sum**
**of PP included in the region is 99%.**

**Reviewer #1, Additional comment #11**

**Throughout the manuscript, the r^2 values used vary widely. For instance, in the “pleiotropy”**
**paragraph in the methods, the authors use two different r^2 cutoffs (0.5 and 0.6 for QTLs) with**
**no apparent rationale. The reviewer acknowledges that changing the r^2 value to an uniform**
**value throughout the manuscript is not feasible, but the authors should explain the rationale**
**for using a specific cutoff in the methods (even if the explanation is just because it’s the**
**default setting of the software used).**

We apologize for using these confusing thresholds. In the pre-revised manuscript, we performed
downstream analyses using different r^2 values because we applied the same analytic pipeline used in
a previous study for each. However, as you correctly indicated, using different r^2 values could result in
confusing the readers. Therefore, to improve consistency across analyses, we unified r^2 value of 0.8
and defined AF-associated variants as the lead variants and their proxies with $r^2 > 0.8$ identified by
the trans-ancestry meta-analysis, referring to a previous paper⁶. Specifically, we assessed the
functional consequence and the pleiotropic effects using this definition of AF-associated variants.
Among them, we observed 19 missense variants, which was slightly fewer than when using $r^2 > 0.6$
(Supplementary Table 5, please refer to the response to Reviewer #1 Major comment #5). For
pleiotropic analysis, a search for overlaps with variants reported in the GWAS catalog yielded 112 loci
(74.7%) with at least one overlapping variant and 631 pairs of AF-associated variants and
phenotypes, in which blood pressure-related traits were most frequently observed (13.5%), followed
by electrocardiogram-related (11.7%) and anthropometric traits (6.8%) (Supplementary Table 17).
This result was not much different from that obtained when using $r^2 > 0.6$.

Below are the corresponding revised text.

(Lines 117 – 118 in the main manuscript)

Of the **3,637** variants in LD ($r^2 > 0.8$) with 150 lead variants, **19** missense variants were observed
(Supplementary Table 5).

(Lines 83 – 88 in Supplementary Notes)

To characterize the AF-associated loci, we examined the pleiotropic effects of the identified lead
variants and proxies using the NHGRI-EBI GWAS catalog database. Of the 150 AF-associated loci,
**112 (74.7%) had at least one overlapping variant, and we found 631 pairs of AF-associated variants**
**and phenotypes, where blood pressure-related traits were most frequently observed (13.5%), followed**
**by electrocardiogram-related (11.7%) and anthropometric traits (6.8%) (Supplementary Table 17).**

(Lines 169 – 172 in Supplementary Notes)

To assess the pleiotropic effects of AF-associated variants, we selected the lead variants and their
 proxies with $r^2 > 0.8$ across 150 AF-associated loci from the trans-ancestry meta-analysis, and
 searched if these variants overlapped with other diseases or traits using NHGRI-EBI GWAS catalog
 on February 8, 2019.

[Supplementary Table 17 (partially extracted)]

Supplementary Table 17 | Pleiotropy of AF-associated loci

Lead variant		Variant associated with other trait			Trait or disease		
CHR	POS	CHR	POS	rsID			
Novel loci							
1	918617	1	895706	rs13303327	Heel bone mineral density		
		1	943468	rs3121567	Blood protein levels		
1	16199051	1	16133396	rs12738340	Hair color		
		1	16188681	rs6701290	Heel bone mineral density		
		1	16277082	rs848188	Hair color		
1	39385714	1	39380385	rs4246511	Menopause (age at onset)		
		1	39364617	rs4970634	Age at menopause		
4	83910712	4	83955107	rs12511815	Red cell distribution width		
		4	83896818	rs12649662	Systolic blood pressure		
		4	83895232	rs13103322	Red blood cell count		
		4	83903895	rs2868708	White blood cell count		
		4	83885220	rs60065504	Lymphocyte percentage of white cells		
		4	83890975	rs6535419	Pulse pressure		
		4	83925895	rs6823199	Pulse pressure		
4	83916644	rs6845791	Red blood cell count				
Known loci							
1	10796866	1	10799577	rs12046278	Diastolic blood pressure		
					Pulse pressure		
					Systolic blood pressure		
		1	10796866	1	10796866	rs880315	Systolic blood pressure
							Pulse pressure
							Blood pressure
							Cardiovascular disease
							Diastolic blood pressure
							Hypertension
							Ischemic stroke
Mean arterial pressure							
Pulse pressure							
Stroke							
1	22282619	1	22266330	rs115979533	Systolic blood pressure		
1	22282619	1	22266330	rs115979533	Lung function (FVC)		

**Reviewer #1, Additional comment #12**

**Also, in the MR analysis, the authors use an r^2 cutoff of 0.05 which is exceptionally high (0.01**
 **or 0.001 are used nowadays).**

Thank you very much for pointing out this issue. We initially performed MR analysis using the
 parameters including r^2 of 0.05 as applied in a previous study⁹. However, as per your
 recommendation, we re-performed the MR analysis using an r^2 threshold of 0.01, which was used in a
 more recent study¹⁰. As a result, there were no significant changes in the causal effects of quantitative
 traits on AF development (Supplementary Fig. 9b). Meanwhile, we did not observe statistically
 significant causality of AF to develop cardiovascular diseases such as sinus node dysfunction,
 atrioventricular block, pericarditis, ischemic heart disease, and other cardiac arrhythmias
 (Supplementary Fig. 9a).

Below are the corresponding revised text and figures.

(Lines 245 – 264 in the main manuscript)

Consistent with clinical evidence, we observed significant genetic liability of AF to the development of
several cardiovascular diseases such as heart failure, cardiomyopathy, stroke, and transient ischemic
attack (Supplementary Fig. 9a and Supplementary Table 10). Additionally, we found the causal effect
of AF on valvular disease (OR [95% CI] = 1.139 [1.133–1.630], $P = 9.4 \times 10^{-4}$ for rheumatic valvular
disease; OR [95% CI] = 1.183 [1.112–1.258], $P = 1.1 \times 10^{-7}$ for valvular heart disease), indicating that
hemodynamic instability and structural remodeling underlying AF may contribute to the development
of valvular diseases.

AF is also known as a consequent phenotype accumulated by multiple atherosclerotic- and metabolic-
related traits. Large observational studies have identified these traits as significant risk factors
associated with AF^{23,24}, but the causal relationship between them has not been fully assessed due to
potential mediators or confounders of these associations. Therefore, we performed a MR analysis to
thoroughly investigate the causality of quantitative traits. We represented the exposures as
quantitative traits and selected the distinct variants associated with each trait as instrumental
variables. As expected²⁵, height and BMI were significant predictors for AF (OR [95% CI] = 1.398
[1.164 – 1.679], $P = 3.3 \times 10^{-4}$; OR [95% CI] = 1.133 [1.061 – 1.209], $P = 1.8 \times 10^{-4}$, respectively).
Furthermore, among atherosclerotic- and metabolic-related traits, we found blood pressure as the
only trait with a causal effect on AF development (OR [95% CI] = 1.400 [1.285 – 1.525], $P = 1.2 \times$
10^{-14} for systolic blood pressure; OR [95% CI] = 1.455 [1.330 – 1.591], $P = 2.1 \times 10^{-16}$ for diastolic
blood pressure; OR [95% CI] = 1.267 [1.161 – 1.381], $P = 9.2 \times 10^{-8}$ for pulse pressure;
Supplementary Fig. 9b and Supplementary Table 10).

(Lines 577 – 579 in the main manuscript)

To identify independent variants for exposure, genome-wide significant variants ($P < 5 \times 10^{-8}$) were
pruned ($r^2 < 0.01$; LD window of 10,000 kb; using European samples of 1KG for LD reference) as
reported previously⁵⁰.

(Supplementary Fig. 9)

[Supplementary Table 10 (partially extracted)]

Supplementary Table 10 Mendelian randomization analysis										
Outcome / Exposure	Number of variants (IV)	MR-IVW			MR-Egger			Cochran's Q		
		Beta	SE	P	Beta	SE	P	Intercept	Intercept P	
Causal effect of AF on cardiovascular and cerebrovascular disease										
Ischemic heart disease	33	0.088	0.038	1.99E-02	-0.074	0.119	5.41E-01	0.010	1.65E-01	1.53E-01
Peripheral artery disease	74	0.070	0.037	5.44E-02	0.168	0.061	7.42E-03	-0.010	4.91E-02	6.01E-01
Heart failure	28	0.176	0.033	1.19E-07	0.102	0.104	3.35E-01	0.005	4.62E-01	4.31E-01
Cardiomyopathy	74	0.306	0.064	1.74E-06	0.240	0.107	2.79E-02	0.007	4.40E-01	2.81E-01
Valvular heart disease	74	0.168	0.032	1.09E-07	0.166	0.051	1.74E-03	0.000	9.60E-01	5.13E-01
Rheumatic valvular disease	33	0.307	0.093	9.40E-04	0.639	0.297	4.16E-02	-0.021	2.51E-01	2.39E-01
Aortic aneurysm and dissection	74	0.126	0.056	2.59E-02	0.012	0.094	8.95E-01	0.012	1.36E-01	7.94E-01
Myocarditis	74	0.307	0.233	1.88E-01	-0.116	0.386	7.64E-01	0.043	1.75E-01	9.00E-02
Pericarditis	74	0.064	0.060	2.90E-01	0.147	0.100	1.48E-01	-0.008	3.04E-01	8.10E-01
Endocarditis	74	-0.065	0.083	4.33E-01	0.160	0.138	2.52E-01	-0.023	4.54E-02	5.66E-01
Sinus node dysfunction	33	0.482	0.234	3.99E-02	0.565	0.770	4.70E-01	-0.005	9.10E-01	4.16E-01
Atrioventricular block	33	0.152	0.137	2.69E-01	-0.363	0.437	4.14E-01	0.032	2.27E-01	1.52E-02
Other cardiac arrhythmia	32	0.069	0.033	3.67E-02	-0.024	0.106	8.20E-01	0.006	3.63E-01	6.17E-01
Cardiac arrest	74	0.115	0.070	9.96E-02	0.088	0.117	4.55E-01	0.003	7.72E-01	2.67E-01
Stroke	74	0.165	0.033	6.81E-07	0.154	0.056	7.07E-03	0.001	8.06E-01	4.03E-01
Transient ischemic attack	74	0.159	0.047	6.62E-04	0.159	0.078	4.68E-02	0.000	9.93E-01	2.21E-01
Intracerebral hemorrhage	74	0.138	0.063	2.87E-02	0.022	0.098	8.20E-01	0.013	1.32E-01	5.74E-01
Subarachnoid hemorrhage	74	-0.012	0.084	8.86E-01	-0.014	0.135	9.20E-01	0.000	9.89E-01	8.35E-01
Vascular dementia	74	0.185	0.127	1.46E-01	0.137	0.213	5.22E-01	0.005	7.80E-01	1.73E-01
Systemic embolism	74	0.153	0.077	4.78E-02	0.170	0.130	1.95E-01	-0.002	8.72E-01	3.65E-01
Causal effect of quantitative traits on AF										
Height	9	0.335	0.093	3.34E-04	0.955	0.493	1.01E-01	-0.034	2.48E-01	5.80E-01
Body mass index	823	0.125	0.033	1.76E-04	0.147	0.109	1.79E-01	0.000	8.29E-01	1.17E-04
Systolic blood pressure	620	0.336	0.044	1.17E-14	0.473	0.129	2.76E-04	-0.003	2.62E-01	8.58E-14
Diastolic blood pressure	537	0.375	0.046	2.12E-16	0.388	0.142	6.45E-03	0.000	9.22E-01	1.18E-14
Pulse pressure	558	0.236	0.044	9.16E-08	0.323	0.120	7.70E-03	0.002	4.41E-01	1.41E-11
White blood cell count	548	0.078	0.041	5.59E-02	0.125	0.115	2.75E-01	-0.001	6.57E-01	1.92E-03
Neutrophil count	495	0.043	0.035	2.28E-01	0.128	0.084	1.27E-01	-0.002	2.61E-01	1.13E-02
Eosinophil count	651	-0.019	0.029	5.11E-01	-0.060	0.060	3.16E-01	0.001	4.33E-01	2.14E-05
Basophil count	155	0.012	0.059	8.45E-01	-0.224	0.121	6.71E-02	0.007	2.89E-02	5.02E-01
Monocyte count	599	-0.035	0.026	1.73E-01	-0.012	0.049	8.03E-01	-0.001	5.72E-01	5.40E-04
Lymphocyte count	523	-0.043	0.035	2.16E-01	-0.066	0.082	4.24E-01	0.001	7.64E-01	1.02E-02
Red blood cell count	723	-0.027	0.032	3.92E-01	-0.140	0.066	3.48E-02	0.003	5.26E-02	3.40E-08
Hemoglobin	515	-0.073	0.051	1.48E-01	0.006	0.137	9.63E-01	-0.002	5.30E-01	1.67E-06

Reviewer #1, Additional comment #13

In the TWAS, the authors base their findings on GTEx7 which is outdated. GTEx8 weights should be readily available for analysis and used in the updated analysis.

Thank you very much for the useful advice. As you suggested, GTEx v8 is now the latest version and

includes a larger dataset of genetic associations and gene expression compared to GTEx v7, which

enables us to perform a more powerful analysis for gene prioritization. We performed TWAS using

GTEx v8, which increased the number of significantly associated genes from 86 to 132 for the atrial
 appendage and from 74 to 127 for the left ventricle (Fig. 3a and Supplementary Table 6).

We modified the description and figure in the revised manuscript as follows.

(Lines 136 – 138 in the main manuscript)

TWAS prioritized **132** and **127** candidate causal genes significantly associated with AF in the atrial
 appendage and left ventricle, respectively (Fig. 3a and Supplementary Table 6).

(Lines 442 – 448 in the main manuscript)

Since tissue enrichment analysis demonstrated that AF-associated variants were predominantly
 enriched in heart tissues (Supplementary Notes and Supplementary Methods), we used precomputed
 prediction models of gene expression in atrial appendage and left ventricular tissues with LD
 reference data in GTEx v8 and the summary statistic of the trans-ancestry AF-GWAS as input.
 Bonferroni significance level was set at $P = 4.8 \times 10^{-6}$ ($=0.05/10,414$) for the atrial appendage and $P =$
 5.2×10^{-6} ($=0.05/9,702$) for the left ventricle to account for the number of genes tested in each tissue.

(Fig 3a)

**Fig. 3 | Transcriptome-wide association analysis. a,** Volcano plot showing individual genes with
 the effect size and the $-\log_{10} P$ value for TWAS based on atrial appendage (left) and left ventricular
 tissues (right) from the GTEx project. Significant genes adjusted Bonferroni correction for all tested
 genes in each tissue (P value $< 0.05/10,414$ for atrial appendage and $< 0.05/9,702$ for left ventricle)
 are highlighted with red (positive effect of predicted gene expression on AF) and blue (negative effect
 of predicted gene expression on AF).

[Supplementary Table 6 (partially extracted)]

Supplementary Table 6 AF-associated genes identified by transcriptome-wide association study using GTEx heart tissues															
Gene		Heart atrial appendage							Heart left ventricle						
Gene symbol	Gene	CHR	POS	rsID	Distance to gene (kb)	Effect size	Z score	P	CHR	POS	rsID	Distance to gene (kb)	Effect size	Z score	P
FERM1	ENSG00000187642.9	1	917492	rs28434575	0.065	0.174	5.463	4.68E-09	1	917640	rs1235816	0.143	0.175	5.942	2.81E-09
SCMH1	ENSG0000010803.16	-	-	-	-	-	-	-	1	41705510	rs12126982	2.272	-0.387	-6.023	1.71E-09
RNF11	ENSG00000123091.4	1	51701842	rs72692291	0.1	-9.747	-5.986	2.15E-09	1	51701842	rs72692291	0.1	-12.385	-5.986	2.15E-09
CASQ2	ENSG00000118729.11	1	116311476	rs1266877999	0.074	-0.380	-7.764	8.21E-15	1	116311476	rs1266877999	0.074	-0.554	-7.828	4.95E-15
SNAPIN	ENSG00000143553.10	1	153632852	rs6427640	1.723	-0.414	-4.576	4.73E-06	-	-	-	-	-	-	-
IL6R	ENSG00000160712.12	1	154397933	rs10908837	20.268	0.221	5.986	2.15E-09	-	-	-	-	-	-	-
PMVK	ENSG00000163344.5	1	154909266	rs1109815	0.199	3.766	13.032	8.02E-39	-	-	-	-	-	-	-
PBX1	ENSG00000163346.16	1	154919080	rs2061690	9.5	0.764	12.855	8.08E-38	-	-	-	-	-	-	-
ZBTB7B	ENSG00000160685.13	1	154980351	rs56103503	5.057	-0.697	-11.262	2.03E-29	1	154980351	rs56103503	5.057	-0.793	-11.262	2.03E-29
DCST2	ENSG00000163354.14	1	155006451	rs76306191	0.215	0.227	9.708	2.79E-22	1	155025361	rs2306125	19.125	0.212	9.361	7.89E-21
ADAM15	ENSG00000143537.13	-	-	-	-	-	-	-	1	155023634	rs78266397	0.127	6.474	7.395	1.42E-13
METTL1B	ENSG00000203740.3	-	-	-	-	-	-	-	1	170135510	rs576572	20.369	-0.199	-6.613	3.77E-11
PRRX1	ENSG00000116132.11	1	170631763	rs529234	1.283	-0.645	-16.979	6.39E-64	1	170631763	rs629234	1.283	-0.357	-18.315	6.29E-75
RPI1-TCF4	ENSG00000271811.1	1	170638656	rs10919449	2.137	-0.121	-13.482	2.05E-41	1	170638692	rs1540322	0.461	-0.272	-15.811	6.08E-35
PK3C2B	ENSG00000133056.13	1	204464181	rs2926534	4.829	-0.418	-4.746	2.07E-06	1	204464150	rs3014606	4.598	-0.549	-4.746	2.07E-06

**Reviewer #2:**

**Remarks to the Author:**

**Miyazawa and colleagues describe the results of their genome-wide association study (GWAS)**
**meta-analyses of atrial fibrillation (AF) in the Japanese population and in a trans-ancestry**
**population of Europeans and Japanese samples, identifying five and 35 novel loci,**
**respectively. Downstream in silico analyses of the novel loci show several biologically and**
**clinically relevant genes and pathways, and genetic risk scores and Mendelian randomization**
**analyses show that AF is a (causal) risk factor for a variety of cardiovascular disorders.**
**However, no functional (in vitro or in vivo) work is done to reveal the biological mechanisms**
**underlying AF. The manuscript is well written and used methodologies and presented results**
**are of high quality. Conclusions mostly flow smoothly from described results. I have some**
**suggestions regarding the analyses and manuscript, and would like the authors to clarify a**
**number of points:**

Thank you very much for carefully reading our manuscript, raising important points, and giving us
useful advice on how to improve our research. To strengthen the results, we have added wet
experiments using human induced pluripotent stem cell-derived cardiomyocytes (please refer to
**Major changes 3** at the beginning of this response letter) to assess the importance of the
transcription factor that we identified. We have taken all your comments into consideration and made
corrections to the manuscript accordingly. Please find below the point-by-point responses to each of
your comments.

**Reviewer #2, Comment #1**

**Five novel loci are found in the Japanese ancestry GWAS that have not been found previously.**
**Since some of these have a low frequency in the Japanese population and are virtually absent**
**from other populations, it is difficult to determine whether they may be false positive hits. I**
**would suggest to use an independent (East Asian) replication dataset to confirm these novel**
**loci. Also I would suggest to describe in the Discussion section why you believe these are true**
**positive hits.**

Thank you for pointing out this important issue. In our Japanese GWAS, we found five novel loci, of
which three lead variants had low allele frequencies in the Japanese population: rs202030113 (MAF =
1.2%), rs1055894680 (MAF = 0.1%), and rs778479352 (MAF = 0.2%). In particular, rs202030113 and
rs778479352 were not found in other ethnic groups. Accordingly, we completely agree with the
reviewer that these loci need to be considered carefully. Therefore, we added functional annotations
using epigenome database and validated the novel loci by a replication study using another Japanese
cohort to prove that these association signals are true positives.

First, we annotated the lead variants with transcriptional information by referring to epigenome
databases, and found that rs778479352 was involved in the region of ENCODE¹¹ accession No.
EH38E2771113, where high H3K4me3 and H3K27ac signals are observed. This result suggested that
rs778479352 might function as a candidate cis-regulatory element (cCRE).

Next, we replicated the five novel loci using an independent Japanese cohort. Specifically, we
performed an association study for the lead variants of these novel loci in a replication cohort of
48,677 Japanese (4,602 cases and 44,075 controls), which were enrolled in the Biobank Japan
second cohort. We then confirmed the reproducibility of these association results, where the nominal
level of significance for each lead variant: rs202030113 $P = 6.19 \times 10^{-3}$, rs2930856 $P = 1.85 \times 10^{-5}$,
rs1055894680 $P = 5.82 \times 10^{-16}$, rs73205368 $P = 7.26 \times 10^{-7}$, rs778479352 $P = 4.74 \times 10^{-10}$ were
observed (Supplementary Table 2).

These results provide functional support and replication for the novel loci identified in our Japanese
GWAS.

Below are the corresponding text and table in the revised manuscript.

(Lines 82 – 94 in the main manuscript)

To confirm the replication of the five newly identified loci, we performed genotyping and association
study in an independent Japanese cohort including 4,602 cases and 44,075 controls. Observations
with all of the lead variants were successfully replicated with nominal associations ($P < 0.05$) in the
same direction (Supplementary Table 2). Among the lead variants in the five novel loci, rs202030113
(MAF = 1.2%) and rs778479352 (MAF = 0.25%) were observed only in the East Asian population
according to the Genome Aggregation Database v2.1.1 (gnomAD)⁹. rs202030113 was located in the
intronic region three bases away from an exon-intron boundary of *SYNE1* and was predicted as a
splice donor loss with a spliceAI delta score of 0.33¹⁰. A strong signal (odds ratio [OR] for AF
development = 2.00, 95% confidence interval [CI] = 1.73-2.31, $P = 1.6 \times 10^{-20}$) was displayed by
rs778479352, the lead variant located in an intron of *FGF13*, which was involved in the region of
ENCODE¹¹ accession No. EH38E2771113, where high H3K4me3 and H3K27ac signals were
observed. This result suggested that rs778479352 might function as a candidate cis-regulatory
element.

(Lines 362 – 367 in the main manuscript)

The samples in the replication study were registered in the BBJ second cohort, which comprised DNA
samples and clinical information of approximately 60,000 new patients with the 38 target diseases
collected between 2013 and 2018 to expand research outcomes from the first cohort. We applied the
same inclusion criteria to the clinical information of the participants and excluded related individuals
estimated by PI_HAT and PCA outliers from the East Asian population. Finally, 48,677 individuals
(4,602 cases and 44,075 controls) were included in the replication study.

(Lines 379 – 384 in the main manuscript)

In the replication study, all participants were genotyped using Illumina Asian Screening Array. We
excluded variants meeting any of the following criteria: (i) SNP call rate $< 98\%$, (ii) a minor allele count
of < 5 , and (iii) Hardy–Weinberg equilibrium P value $< 1.0 \times 10^{-6}$. Post-QC genotype data were pre-
phased using Shapeit2, and imputed using minimac4 with the 1KG Phase 3 reference panel and

3,256 Japanese in-house reference panel from BBJ. Pre-phasing and imputation of the X
 chromosome were performed using the same pipeline applied for autosomes.

[Supplementary Table 2]

Supplementary Table 2 Replication of five newly identified loci in an independent Japanese cohort													
CHR	POS	REF	ALT	rsID	Discovery cohort				Replication cohort				
					AAF	Beta	SE	P	AAF	Beta	SE	P	
6	152466619	T	C	rs202030113	0.012	0.352	0.063	1.90E-08	0.012	0.249	0.091	6.19E-03	
12	104471663	C	T	rs2930856	0.579	0.090	0.015	5.80E-09	0.521	0.097	0.023	1.85E-05	
16	30619745	C	T	rs1055894680	0.001	1.215	0.158	1.58E-14	0.002	1.297	0.460	5.82E-16	
X	23399501	T	C	rs73205368	0.285	0.089	0.012	7.50E-13	0.282	0.092	0.186	7.26E-07	
X	137790580	T	C	rs778479352	0.002	0.692	0.075	1.62E-20	0.003	0.712	0.114	4.74E-10	

**Reviewer #2, Comment #2**

**Sex stratified GWAS for AF: The authors describe that several loci show a statistically**
 **stronger association (i.e. much smaller P-values) in males than females. However, the study in**
 **males contains ~2 times more cases. Please incorporate this information in this section. In**
 **addition, I would be interested in the number of novel loci found among the significant loci in**
 **the sex-stratified GWAS.**

Thank you for pointing out the issue. Indeed, the number of samples differed between the male- and
 the female-GWAS, which could result in a difference in detection power. First, we have mentioned this
 point in the main text. In addition, to assess the impact of the sample sizes on the heterogeneity in the
 sex-stratified GWAS, we performed power calculation for the significant loci with heterogeneity using
 Genetic Association Study Power Calculator provided by the University of Michigan
 (https://csg.sph.umich.edu/abecasis/gas_power_calculator/). Please also refer to the response to
 Reviewer #1 Major comment #4. To achieve 80% power, the effective sample size for the *PITX2-*
 *C4orf32* and *CUX2* loci were approximately 450 and 3,000 cases, respectively. This result suggests
 that the sample size of the sex-stratified GWAS has sufficient statistical power to detect the significant
 association signals of the two loci and supports the genetic heterogeneity between sexes in the two
 loci rather than the mere difference in the sample sizes.

Regarding the novel loci in the sex-stratified GWAS, none of the novel loci were identified in the
 female-GWAS. Meanwhile, we identified three novel loci in the male-GWAS, the *HCFC2*, *PTCHD1*,
 and *FGF13* loci, which were also observed in our Japanese GWAS including both sexes, which were
 highlighted in red in Supplementary Fig. 12a.

Below are the corresponding text, table, and figure in the revised manuscript.

(Lines 24 – 33 in Supplementary Notes)

Since the sample sizes in the sex-stratified GWAS were different (6,825 cases in the male-GWAS and
 3,001 cases in the female-GWAS), we calculated the effective sample sizes with sufficient statistical
 power using the Genetic Association Study Power Calculator to examine whether the heterogeneity
 between sexes in these two loci was due to the difference in the sample sizes. To achieve 80%
 power, the effective sample size for the *PITX2-C4orf32* locus was approximately 450, while the *CUX2*
 locus required 3,000 cases. This result shows that the sex-stratified GWAS has enough sample size
 with sufficient statistical power to detect the significant association signals of the two loci

(Supplementary Table 14) and supports the genetic heterogeneity between sexes in the two loci
 rather than the mere difference in the sample sizes.

(Lines 124 – 131 in Supplementary Notes)

To calculate the effective sample sizes with sufficient statistical power, we used the Genetic
 Association Study Power Calculator provided by the University of Michigan
 (https://csg.sph.umich.edu/abecasis/gas_power_calculator/), under the following conditions: a
 significance level of $P < 5 \times 10^{-8}$ and disease prevalence in the Japanese population of 0.82%
 (<https://vizhub.healthdata.org/gbd-compare/>). We also used the allele frequencies and genotype
 relative risks of the lead variants in the *PITX2-C4orf32* and *CUX2* loci as input data (Supplementary
 Table 14). The effective sample size was defined as the minimum number of samples to achieve 80%
 power.

(Supplementary Table 14)

Supplementary Table 14 | Power calculation by GAS Power Calculator

PITX2-C4orf32 locus												
CHR	Start	End	Lead variant							N _{case}		Effective sample size for 80% statistical power
			CHR	POS	rsID	REF	ALT	AAF	OR	Male-GWAS	Female-GWAS	
4	111467478	112007484	4	111716513	rs12644625	C	T	0.455	1.766	6,825	3,001	450
CUX2 locus												
CHR	Start	End	Lead variant							N _{case}		Effective sample size for 80% statistical power
			CHR	POS	rsID	REF	ALT	AAF	OR	Male-GWAS	Female-GWAS	
12	110153675	114812626	12	111609727	rs3809297	G	T	0.235	0.7556	6,825	3,001	3,000

(Supplementary Fig. 12a)

**Reviewer #2, Comment #3**

**Several downstream analyses use not only the index/lead variant for each locus but also SNPs**
**in LD. However, the r^2 used to determine SNPs in LD is not consistent across the manuscript.**
**Please explain why a specific threshold is chosen for a certain analysis.**

We apologize for this confusing threshold usage. In the pre-revised manuscript, we had set the r^2
threshold independently for each analysis, by referring to previous studies that performed the same
kind of downstream analysis. However, we agree that using different r^2 values could result in
confusing the readers. Therefore, to improve consistency across analyses, we unified r^2 value of 0.8
and defined AF-associated variants as the lead variants and their proxies with $r^2 > 0.8$ identified by
the trans-ancestry meta-analysis, referring to a previous paper⁶. Specifically, we assessed the
functional consequence and the pleiotropic effects of these AF-associated variants. Please also refer
to the response to Reviewer #1 Additional comment #11. Among variants in LD ($r^2 > 0.8$) with 150
lead variants, 19 missense variants were observed (Supplementary Table 5), which was slightly fewer
than that observed when using $r^2 > 0.6$. For pleiotropic analysis, a search for overlaps with variants
reported in the GWAS catalog yielded 112 loci (74.7%) with at least one overlapping variant and 631
pairs of AF-associated variants and phenotypes, in which blood pressure-related traits were most
frequently observed (13.5%), followed by electrocardiogram-related (11.7%) and anthropometric traits
(6.8%) (Supplementary Table 17). This result was not much different from that obtained when using r^2
> 0.6 .

Below are the corresponding text and table in the revised manuscript.

(Lines 117 – 118 in the main manuscript)

Of the **3,637** variants in LD ($r^2 > 0.8$) with 150 lead variants, **19** missense variants were observed
(Supplementary Table 5).

(Lines 83 – 88 in Supplementary Notes)

To characterize the AF-associated loci, we examined the pleiotropic effects of the identified lead
variants and proxies using the NHGRI-EBI GWAS catalog database. Of the 150 AF-associated loci,
**112 (74.7%) had at least one overlapping variant, and we found 631 pairs of AF-associated variants**
**and phenotypes, where blood pressure-related traits were most frequently observed (13.5%), followed**
**by electrocardiogram-related (11.7%) and anthropometric traits (6.8%) (Supplementary Table 17).**

(Lines 169 – 172 in Supplementary Notes)

To assess the pleiotropic effects of AF-associated variants, we selected the lead variants and their
proxies with $r^2 > 0.8$ across 150 AF-associated loci from the trans-ancestry meta-analysis, and
searched if these variants overlapped with other diseases or traits using NHGRI-EBI GWAS catalog
on February 8, 2019.

(Supplementary Table 5 (partially extracted))

Supplementary Table 5 | Missense variants in linkage disequilibrium with the lead variants detected in trans-ancestry meta-analysis

CHR	POS	rsID	REF	ALT	Gene	r ²	Lead SNP			Functional impact					
							CHR	POS	rsID	SIFT score	SIFT pred	Polyphen2 HDIV score	Polyphen2 HDIV pred	CADD raw	CADD pred
Novel loci															
1	16255644	rs848208	C	T	SPEN	0.989	1	16199051	rs9782984	0.098	Tolerated	0.006	Benign	-0.317	0.608
4	83838262	rs897945	G	T	THAP9	0.964	4	83910712	rs6841049	0.499	Tolerated	0.906	Possibly damaging	-0.478	0.250
20	36841914	rs3746471	G	A	KIAA1755	1.000	20	36841914	rs3746471	0.059	Tolerated	0.003	Benign	2.905	21.900
Known loci															
2	179397561	rs3829747	C	T	TTN	0.992	2	179413110	rs3731748	0	Deleterious	1.000	Probably damaging	4.497	24.300
2	179457147	rs16866406	G	A	TTN	0.992	2	179413110	rs3731748	0.167	Tolerated	0.008	Benign	2.159	17.240
3	128488222	rs11718898	T	C	CAND2	0.932	3	12841804	rs7650482	0.502	Tolerated	0.000	Benign	1.109	11.260
3	38766675	rs6795970	A	G	SCN10A	0.959	3	38767315	rs6801957	1	Tolerated	0.000	Benign	-0.943	0.021
3	89521893	rs35124509	T	C	EPHA3	0.927	3	89489529	rs6771054	0.229	Tolerated	0.000	Benign	2.291	18.100
7	881668	rs6461378	C	T	SUN1	0.841	7	895285	rs6461461	0.736	Tolerated	0.945	Possibly damaging	3.431	23.000
10	75406912	rs34163229	G	T	SYNPO2L	0.967	10	75414344	rs60212594	0.02	Deleterious	0.986	Probably damaging	2.766	21.200
10	75415677	rs60632610	C	T	SYNPO2L	0.983	10	75414344	rs60212594	0.02	Deleterious	0.997	Probably damaging	6.741	32.000
10	75759483	rs12217245	T	C	CAMK2G	0.844	10	75414344	rs60212594	NA	NA	NA	NA	NA	NA
12	57109792	rs2858149	A	G	NACA	1.000	12	57105938	rs2860482	1	Tolerated	0.000	Benign	1.812	15.080
12	57114100	rs2826743	A	G	NACA	1.000	12	57105938	rs2860482	0.423	Tolerated	0.000	Benign	-1.288	0.005
13	113909339	rs2302757	A	G	CUL4A	0.983	13	113864837	rs2316443	0.204	Tolerated	0.009	Benign	1.207	11.790
17	38028634	rs11557467	G	T	ZBP2	0.984	17	38049589	rs7359623	0.045	Deleterious	0.295	Benign	1.537	13.510
17	38062196	rs2305480	G	A	GSDMB	0.829	17	38049589	rs7359623	0.898	Tolerated	0.877	Possibly damaging	-1.352	0.004
17	38062217	rs2305479	C	T	GSDMB	0.960	17	38049589	rs7359623	0.129	Tolerated	1.000	Probably damaging	3.505	23.100
22	26159289	rs133885	G	A	MYO18B	0.892	22	26164079	rs133902	1	Tolerated	0.000	Benign	1.100	11.220

[Supplementary Table 17 (partially extracted)]

Supplementary Table 17 | Pleiotropy of AF-associated loci

Lead variant		Variant associated with other trait			Trait or disease		
CHR	POS	CHR	POS	rsID			
Novel loci							
1	918617	1	895706	rs13303327	Heel bone mineral density		
		1	943468	rs3121567	Blood protein levels		
1	16199051	1	16133396	rs12738340	Hair color		
		1	16188681	rs6701290	Heel bone mineral density		
		1	16277082	rs848188	Hair color		
1	39385714	1	39380385	rs4246511	Menopause (age at onset)		
		1	39364617	rs4970634	Age at menopause		
4	83910712	4	83955107	rs12511815	Red cell distribution width		
		4	83896818	rs12649662	Systolic blood pressure		
		4	83895232	rs13103322	Red blood cell count		
		4	83903895	rs2868708	White blood cell count		
		4	83885220	rs60065504	Lymphocyte percentage of white cells		
		4	83890975	rs6535419	Pulse pressure		
		4	83925895	rs6823199	Pulse pressure		
4	83916644	rs6845791	Red blood cell count				
Known loci							
1	10796866	1	10799577	rs12046278	Diastolic blood pressure		
					Pulse pressure		
					Systolic blood pressure		
		1	10796547	rs17035646	rs34071855	Systolic blood pressure	
						Pulse pressure	
						Blood pressure	
		1	10796866	1	10796866	rs880315	Cardiovascular disease
							Diastolic blood pressure
							Hypertension
							Ischemic stroke
							Mean arterial pressure
1	22282619	1	22266330	rs115979533	Pulse pressure		
1	22282619	1	22266330	rs115979533	Stroke		
1	22282619	1	22266330	rs115979533	Systolic blood pressure		
1	22282619	1	22266330	rs115979533	Lung function (FVC)		

Reviewer #2, Comment #4
Lines 239-243, distance between the lead SNP and potential causal gene: how do these numbers compare to what has been described in literature?

Thank you very much for raising this important point. In the pre-revised manuscript, we calculated the shortest distance from the lead variant to the canonical transcript of candidate genes identified by TWAS, which showed approximately 20% of the candidate genes 500kb away from the lead variants. This result was consistent with a previous study by Roselli et al. in 2018³, in which 26% of AF-associated genes identified by TWAS were more than 500 kb apart from the lead variants. However,

given the closer relationship between the non-coding variants and the target genes in complex
diseases, which was reported in Nasser et al.'s study published in *Nature* 2021⁴, our results gave the
impression of the longer-acting effect (trans-effect) of AF-associated variants compared to other
complex diseases. Therefore, we decided to revise our method rigorously, in which we first extracted
an AF-associated variant with the lowest association *P* value in the prediction model for each
candidate gene, and calculated the physical distance from the variant to the transcription start site of
the canonical transcript for each candidate gene. We then observed that the median physical
distances were 2.25 kb and 1.14 kb in the atrial appendage and left ventricle, respectively (Fig. 3c),
which demonstrated shorter-acting effects of AF-associated variants compared to those in our
previous result while they were comparable to the results from the Nasser et al.'s study⁴. Please also
refer to the Reviewer#1, Major Comment #7

Below are the corresponding text and figures in the revised manuscript.

(Lines 141 – 151 in the main manuscript)

Furthermore, to assess the cis- and trans-effect of AF-associated variants on the candidate genes, we
calculated the physical distances from the variants to the transcription start site (TSS) of the
candidate genes. Only 34 and 35 genes overlapped between the nearest genes to the lead variants
and the candidate genes identified by TWAS (Fig. 3b), and the median physical distances were 2.25
695 kb and 1.14 kb (Fig. 3c) in the atrial appendage and left ventricle, respectively. This relationship
between the AF-associated variants and the candidate genes is comparable to a previous study in
which the distances from the non-coding GWAS signals to the target genes were assessed based on
chromatin state and three-dimensional contacts¹⁸. Exceptionally, only one gene, *FBN2*, was more
than 500 kb away from the variant in the left ventricle (512 kb). This result indicates that, although
disease-associated genes are not necessarily closest to the lead variants, the majority of candidate
genes are affected by the cis-effect of AF-associated variants.

(Lines 448 – 451 in the main manuscript))

To assess the relationship between AF-associated variants and the candidate genes, we first
extracted an AF-associated variant with the lowest association *P* value among those included in the
prediction model obtained from GTEx PredictDB⁴¹ for each candidate gene, and then calculated the
physical distances from the variant to the TSS of the canonical transcript for each candidate gene.

(Fig. 3b,c)

**Fig. 3 | Transcriptome-wide association analysis.** **b**, Number of genes located closest to the lead
variants (red), identified by TWAS (blue), and overlapped between them (green) in atrial appendage
and left ventricular tissues. **c**, Distribution of physical distances from the AF-associated variant
included in the prediction model to the TSS of the canonical transcript for each candidate gene
identified by TWAS in atrial appendage (left) and left ventricular tissues (right).

**Reviewer #2, Comment #5**

**Lines 258 and 302-303: a P-value threshold of 0.05 is used while a large number of tests have**
**been performed. How do the authors warrant the lack of multiple testing correction?**

We apologize for not performing corrections in these analyses. As you indicated, both analyses
needed multiple testing corrections. Accordingly, Bonferroni correction was applied based on the
number of tests performed to determine the significance level, which is reflected in the revised
manuscript. For the analysis of transcription factors, we removed the description of the enrichment
with a nominal threshold level of $P < 0.05$ and left only the result of ERRg with a Bonferroni-corrected
significance level of $P < 0.05/15,109$ (Fig. 4a). For the association analysis between AF-PRS and
stroke phenotypes, as Reviewer #3 suggested, we adjusted the use of anticoagulants or antiplatelets
as one of the covariates, and then found that atherothrombotic stroke and cerebral hemorrhage did
not reach a statistically significant level. However, as expected, the associations with cerebral
infarction ($P = 4.0 \times 10^{-4}$) and cardioembolic stroke ($P = 1.3 \times 10^{-3}$) were statistically significant even
after Bonferroni correction ($P < 0.05/6$) (Fig. 6b).

We corrected Fig. 6b and added the following descriptions to the manuscript.

(Lines 156 – 160 in the main manuscript)

We performed enrichment analysis using the ChIP-Atlas dataset¹⁹, which comprised several high-
throughput ChIP-seq experiments (15,109 experiments, 1,028 transcription factors). As a result,
**estrogen-related receptor gamma (ERRg)** binding was significantly enriched in AF-associated loci
with a Bonferroni-corrected significance level of $P = 3.3 \times 10^{-6}$ ($0.05/15,109$) (Fig. 4a and
Supplementary Table 7).

(Lines 209 – 213 in the main manuscript)

We performed logistic regression analysis in 121,351 control samples in our dataset, and found
 significant associations of the PRS with increased risks of cerebral infarction (OR [95% CI] = 1.042
 [1.018 – 1.065], $P = 4.0 \times 10^{-4}$) and cardioembolic stroke (OR [95% CI] = 1.355 [1.126 – 1.630], $P =$
 1.3×10^{-3}) after Bonferroni correction (Fig. 6b).

 (Fig. 6b)

**Reviewer #2, Comment #6**

**Mendelian randomization section: the authors refer to clinical causality vs. genetic causality –**
 **I would suggest to rephrase this sentence because MRs use a genetic instrument to determine**
 **the (clinically relevant) causality between traits/diseases. In addition, I wonder if the authors**
 **excluded variants with pleiotropic effects?**

Thank you very much for your accurate remarks. As you mentioned, our manuscript did not properly
 describe MR analysis, a method that uses genetic instruments to determine the causal relationship
 between traits/diseases like a randomized controlled trial. Accordingly, we revised the objective and
 explanation of MR analysis. Additionally, as you pointed out, the exclusion of variants with pleiotropic
 effects is an important procedure to perform MR analysis. Therefore, referring to a previous paper by
 Surendran et al. *Nature Genetics*, 2020¹², we used PhenoScanner V2
 (<http://www.phenoscaner.medschl.cam.ac.uk/>) to exclude variants with pleiotropic effects based on
 the outcome variables. Specifically, to assess the causal effect on cardiovascular diseases, we
 excluded variants associated with cardiovascular risk factors such as hypertension, cholesterol,
 diabetes mellitus, and smoking as well as those associated with cardiovascular diseases from the list
 of instrument variables. Likewise, to assess the causal effect on AF, we excluded variants associated
 with risk factors for AF such as hypertension, diabetes mellitus, and obesity from the list of instrument
 variables, unless exposure was a risk factor itself. As a result, the intercept term from MR-Egger
 regression showed no significant pleiotropy (Supplementary Table 10), and we properly detected

significant causal effects of AF on heart failure and stroke as we expected (Supplementary Fig. 9).
Further, as a previous study demonstrated, a significant causal effect of AF on height was observed,
which indicated the validity of our analysis.
Below are the corresponding revised text, figure, and table.

(Lines 238 – 264 in the main manuscript)

[revised manuscript text omitted]

(Supplementary Fig. 9)

[Supplementary Table 10]

Supplementary Table 10 Mendelian randomization analysis										
Outcome / Exposure	Number of variants (IV)	MR-IVW			MR-Egger			Cochran's Q		
		Beta	SE	P	Beta	SE	P	Intercept	Intercept P	P
Causal effect of AF on cardiovascular and cerebrovascular disease										
Ischemic heart disease	33	0.088	0.038	1.99E-02	-0.074	0.119	5.41E-01	0.010	1.65E-01	1.53E-01
Peripheral artery disease	74	0.070	0.037	5.44E-02	0.168	0.061	7.42E-03	-0.010	4.91E-02	6.01E-01
Heart failure	28	0.176	0.033	1.19E-07	0.102	0.104	3.35E-01	0.005	4.62E-01	4.31E-01
Cardiomyopathy	74	0.306	0.064	1.74E-06	0.240	0.107	2.79E-02	0.007	4.40E-01	2.81E-01
Valvular heart disease	74	0.168	0.032	1.09E-07	0.166	0.051	1.74E-03	0.000	9.60E-01	5.13E-01
Rheumatic valvular disease	33	0.307	0.093	9.40E-04	0.639	0.297	4.16E-02	-0.021	2.51E-01	2.39E-01
Aortic aneurysm and dissection	74	0.126	0.056	2.59E-02	0.012	0.094	8.95E-01	0.012	1.36E-01	7.94E-01
Myocarditis	74	0.307	0.233	1.88E-01	-0.116	0.386	7.64E-01	0.043	1.75E-01	9.00E-02
Pericarditis	74	0.064	0.060	2.90E-01	0.147	0.100	1.48E-01	-0.008	3.04E-01	8.10E-01
Endocarditis	74	-0.065	0.083	4.33E-01	0.160	0.138	2.52E-01	-0.023	4.54E-02	5.66E-01
Sinus node dysfunction	33	0.482	0.234	3.99E-02	0.565	0.770	4.70E-01	-0.005	9.10E-01	4.16E-01
Atrioventricular block	33	0.152	0.137	2.69E-01	-0.363	0.437	4.14E-01	0.032	2.27E-01	1.52E-02
Other cardiac arrhythmia	32	0.069	0.033	3.67E-02	-0.024	0.106	8.20E-01	0.006	3.63E-01	6.17E-01
Cardiac arrest	74	0.115	0.070	9.96E-02	0.088	0.117	4.55E-01	0.003	7.72E-01	2.67E-01
Stroke	74	0.165	0.033	6.81E-07	0.154	0.056	7.07E-03	0.001	8.06E-01	4.03E-01
Transient ischemic attack	74	0.159	0.047	6.62E-04	0.159	0.078	4.68E-02	0.000	9.93E-01	2.21E-01
Intracerebral hemorrhage	74	0.138	0.063	2.87E-02	0.022	0.098	8.20E-01	0.013	1.32E-01	5.74E-01
Subarachnoid hemorrhage	74	-0.012	0.084	8.86E-01	-0.014	0.135	9.20E-01	0.000	9.89E-01	8.35E-01
Vascular dementia	74	0.185	0.127	1.46E-01	0.137	0.213	5.22E-01	0.005	7.89E-01	1.73E-01
Systemic embolism	74	0.153	0.077	4.78E-02	0.170	0.130	1.95E-01	-0.002	8.72E-01	3.65E-01
Causal effect of quantitative traits on AF										
Height	9	0.335	0.093	3.34E-04	0.955	0.493	1.01E-01	-0.034	2.48E-01	5.80E-01
Body mass index	823	0.125	0.033	1.76E-04	0.147	0.109	1.79E-01	0.000	8.29E-01	1.17E-04
Systolic blood pressure	620	0.336	0.044	1.17E-14	0.473	0.129	2.76E-04	-0.003	2.62E-01	8.58E-14
Diastolic blood pressure	537	0.375	0.046	2.12E-16	0.388	0.142	6.45E-03	0.000	9.22E-01	1.18E-14
Pulse pressure	558	0.236	0.044	9.16E-08	0.323	0.120	7.70E-03	0.002	4.41E-01	1.41E-11
White blood cell count	548	0.078	0.041	5.59E-02	0.125	0.115	2.75E-01	-0.001	6.57E-01	1.92E-03
Neutrophil count	495	0.043	0.035	2.28E-01	0.128	0.084	1.27E-01	-0.002	2.61E-01	1.13E-02
Eosinophil count	651	-0.019	0.029	5.11E-01	-0.060	0.060	3.16E-01	0.001	4.33E-01	2.14E-05
Basophil count	155	0.012	0.059	8.45E-01	-0.224	0.121	6.71E-02	0.007	2.89E-02	5.02E-01
Monocyte count	599	-0.035	0.026	1.73E-01	-0.012	0.049	8.03E-01	-0.001	5.72E-01	5.40E-04
Lymphocyte count	523	-0.043	0.035	2.16E-01	-0.066	0.082	4.24E-01	0.001	7.64E-01	1.02E-02
Red blood cell count	723	-0.027	0.032	3.92E-01	-0.140	0.066	3.48E-02	0.003	5.26E-02	3.40E-08
Hemoglobin	515	-0.073	0.051	1.48E-01	0.006	0.137	9.63E-01	-0.002	5.30E-01	1.67E-06

**Reviewer #2, Comment #7**

**Discussion lines 371-373: while I agree with the authors that the PRS shows clear groups with**
 **different AF risks, the conclusion that it will help identifying individuals at risk for precision**
 **medicine is not something I can agree with.**

Thank you for your valuable suggestion. We agree that it would be difficult to directly apply our AF-
 PRS to precision medicine at present. We recognize that there are many obstacles to overcome for
 actual utilization of AF-PRS in clinical practice, such as superiority over the established risk prediction
 models. For instance, several studies have reported conflicting results regarding the addition of PRS
 to clinical risk scores for predicting coronary heart disease^{13,14}. However, we found a statistically
 significant performance in predicting the early onset of AF, the development of cardioembolic stroke,
 and the risk of long-term cardiovascular and stroke mortalities, which could act as a basis for future
 precision medicine in AF. Of course, these results should need to be validated in prospective cohort
 studies. Therefore, we modified the description as follows, discussing these issues with the
 advantages of our results.

(Lines 301 – 324 in the main manuscript)

During the last decade, there has been a growing interest in predicting complex diseases or traits
 using genetic data. PRS is expected to provide a clinical utility to enhance disease risk prediction,
 whereas previous studies have demonstrated comparable or less performance and a weak additive
 effect of PRS to the established risk prediction models^{33,34}. Additionally, the lack of trans-ancestry
 portability of PRS has also been reported due to the predominant proportion of individuals of
 European descent in the current GWASs³⁵. In this study, we found shared allelic effects of AF-
 associated variants and genetic correlations between Japanese and European populations
 (Supplementary Notes and Supplementary methods). Therefore, we exhaustively examined AF-PRS
 using various combinations of GWASs and multiple parameters to maximize the predictive

performance of AF-PRS; AF-PRS achieved (1) a higher performance when applied to the same
population as the derivation-GWAS population regardless of the sample size in the single derivation-
GWAS category, (2) a higher performance when it was derived from a trans-ancestry meta-GWAS
including the Japanese population compared to that derived from a meta-GWAS in a non-Japanese
population even with a similar or smaller sample size of derivation-GWAS, and (3) the best
performance when it was derived from the trans-ancestry meta-GWAS including the Japanese
population and with the largest sample size. Furthermore, recent studies have shown the potential
utility of PRS in a variety of clinical settings, such as diagnostic refinement³⁶ and prediction of disease
progression³⁷. Our study also demonstrated that, in addition to the predictive ability for AF itself, AF-
PRS segregated individuals with AF-related phenotypes, such as early onset of AF and cardioembolic
stroke, as well as those with increased risks of long-term cardiovascular and stroke mortalities. This
indicated that the cumulative genetic risk for AF may be an indicator for early therapeutic intervention,
including anticoagulation in at-risk individuals as primary prevention of stroke. Taken together, **our**
**results have several implications for the clinical utility of AF-PRS, which will be clues for the**
**realization of future precision medicine.**

**Reviewer #2, Comment #8**

**Table 2 shows the novel loci after the trans-ancestry meta-analysis. However, if I understand it**
**correctly, many loci are significant in the EUR samples only, which come from a study that is**
**published few years ago. Why are these loci described as novel?**

We apologize for the confusion caused by our mistake. In Table 2, the summary statistics of EUR
population GWAS were typographical errors. These loci did meet a genome-wide significance level (P
$< 5.0 \times 10^{-8}$) just in the meta-analysis, not in EUR analysis. We also confirmed that all these loci have
not been previously reported, referring to the Cardiovascular Disease Knowledge Portal
(<http://www.broadcvdi.org/>).

We have corrected Table 2 in this regard.

[Table 2 (partially extracted)]

CHR	POS(hg19)	BBJ				EUR				FIN			
		AAF	Beta	SE	P	AAF	Beta	SE	P	AAF	Beta	SE	P
1	918617	0.076	0.062	0.028	2.90×10 ⁻²	0.167	0.044	0.010	7.54×10 ⁻⁶	0.175	0.102	0.025	4.10×10 ⁻⁵
1	16199051	0.723	-0.075	0.017	1.54×10 ⁻³	0.883	-0.055	0.014	8.40×10 ⁻⁵	0.835	-0.035	0.025	1.72×10 ⁻¹
1	39385714	0.060	0.068	0.032	3.49×10 ⁻²	0.077	0.068	0.015	4.35×10 ⁻⁶	0.059	0.106	0.040	8.18×10 ⁻³
1	100149308	0.698	0.034	0.016	3.75×10 ⁻²	0.677	0.036	0.007	5.21×10 ⁻⁷	0.664	0.027	0.020	1.82×10 ⁻¹
4	71776935	0.621	-0.039	0.016	1.35×10 ⁻³	0.623	-0.034	0.007	1.27×10 ⁻⁵	0.542	-0.030	0.019	1.89×10 ⁻²
4	83910712	0.342	-0.044	0.016	5.58×10 ⁻³	0.567	-0.037	0.007	6.01×10 ⁻⁶	0.608	-0.019	0.019	3.19×10 ⁻¹
5	139703286	0.385	0.059	0.015	1.37×10 ⁻⁴	0.276	0.036	0.007	1.86×10 ⁻⁶	0.357	0.043	0.020	3.00×10 ⁻²
6	22598259	0.229	0.040	0.018	2.19×10 ⁻²	0.282	0.031	0.007	2.04×10 ⁻⁵	0.222	0.071	0.023	1.62×10 ⁻³
6	76164589	0.135	0.088	0.021	4.34×10 ⁻⁵	0.108	0.059	0.011	7.42×10 ⁻⁸	0.137	0.082	0.027	2.59×10 ⁻³
135119089	0.829	-0.052	0.020	8.17×10 ⁻³	0.556	-0.037	0.007	7.51×10 ⁻⁸	0.645	-0.036	0.020	7.14×10 ⁻²
105612736	0.570	0.060	0.016	1.13×10 ⁻⁴	0.273	0.030	0.008	5.49×10 ⁻⁵	0.292	0.051	0.021	1.47×10 ⁻²
118863412	0.230	0.027	0.018	1.20×10 ⁻³	0.180	0.040	0.009	4.85×10 ⁻⁵	0.160	0.082	0.026	1.35×10 ⁻³
119181794	0.086	0.035	0.026	1.86×10 ⁻³	0.290	0.034	0.007	5.27×10 ⁻⁶	0.254	0.077	0.022	3.80×10 ⁻⁴
32772734	0.303	-0.048	0.016	3.30×10 ⁻³	0.113	0.051	0.011	1.92×10 ⁻⁶	0.103	-0.084	0.031	7.23×10 ⁻³
10	50485434	0.096	-0.083	0.026	1.43×10 ⁻³	0.053	-0.066	0.015	1.25×10 ⁻⁵	0.057	-0.138	0.041	7.16×10 ⁻⁴
10	80898969	0.715	0.052	0.017	2.72×10 ⁻³	0.490	0.034	0.008	5.38×10 ⁻⁶	0.501	0.062	0.019	1.11×10 ⁻³
11	3890059	0.661	-0.036	0.016	2.27×10 ⁻²	0.489	-0.030	0.007	5.10×10 ⁻⁶	0.484	-0.042	0.019	2.50×10 ⁻²
11	14036189	0.341	0.062	0.016	8.40×10 ⁻⁵	0.376	0.035	0.007	3.45×10 ⁻⁷	0.361	0.081	0.020	3.61×10 ⁻⁵
11	95089882	0.219	-0.022	0.018	2.30×10 ⁻³	0.670	-0.037	0.007	1.80×10 ⁻⁷	0.603	-0.028	0.019	1.47×10 ⁻¹
12	12886027	0.104	0.064	0.026	1.26×10 ⁻²	0.132	0.049	0.010	2.32×10 ⁻⁶	0.134	0.037	0.028	1.77×10 ⁻¹
12	104492003	0.420	-0.088	0.015	6.35×10 ⁻³	0.141	-0.038	0.009	6.71×10 ⁻³	0.152	-0.045	0.026	8.69×10 ⁻²
12	110082115	0.446	-0.097	0.015	1.46×10 ⁻¹⁰	0.537	-0.011	0.007	9.59×10 ⁻²	0.431	-0.012	0.019	5.20×10 ⁻¹
12	113196733	0.417	-0.111	0.017	9.88×10 ⁻¹¹	0.312	-0.009	0.008	2.12×10 ⁻¹	0.252	-0.034	0.022	1.16×10 ⁻¹
13	22111521	0.324	0.026	0.016	1.11×10 ⁻¹	0.404	0.030	0.007	8.82×10 ⁻⁶	0.388	0.064	0.019	8.57×10 ⁻⁴
13	74520186	0.195	0.047	0.018	1.08×10 ⁻²	0.357	0.036	0.007	2.81×10 ⁻⁷	0.360	0.024	0.020	2.29×10 ⁻¹
16	15902715	0.194	-0.057	0.019	3.39×10 ⁻³	0.314	-0.035	0.007	1.22×10 ⁻⁵	0.335	-0.044	0.020	2.92×10 ⁻²
18	77156537	0.305	0.054	0.017	1.18×10 ⁻³	0.488	0.038	0.007	1.33×10 ⁻⁷	0.446	0.010	0.019	6.19×10 ⁻¹
19	48142746	0.818	0.045	0.020	2.23×10 ⁻²	0.660	0.038	0.007	5.93×10 ⁻⁷	0.708	0.051	0.021	1.35×10 ⁻²
20	36841914	0.430	0.050	0.015	7.96×10 ⁻⁴	0.469	0.035	0.007	2.30×10 ⁻⁷	0.437	0.057	0.019	2.94×10 ⁻³
22	21999229	0.360	0.069	0.018	1.35×10 ⁻⁴	0.191	0.036	0.009	1.00×10 ⁻⁴	0.288	0.035	0.021	9.70×10 ⁻²
22	42189407	0.697	-0.025	0.016	1.33×10 ⁻¹	0.676	0.038	0.007	1.69×10 ⁻⁷	0.682	0.070	0.020	5.73×10 ⁻⁴
X	23399501	0.285	0.089	0.012	7.50×10 ⁻¹³	0.051	0.009	0.024	7.12×10 ⁻¹	0.021	0.148	0.054	5.74×10 ⁻³
X	137418967	0.079	0.091	0.020	5.93×10 ⁻⁶	0.180	0.038	0.011	7.50×10 ⁻⁴	0.192	0.086	0.019	9.31×10 ⁻⁶

**Reviewer #2, Comment #9**

**Figure 4: How do these numbers compare to a model including only the covariates/PCs? Also please**
**note the Y-axis label contains a typo.**

Thank you for raising this point. For the assessment of PRS performance, we measured Nagelkerke's
pseudo R^2 and AUCROC using the regression model, which included age, sex, and PCs as
covariates. Regarding the number of PCs, we previously used 10 PCs as covariates in the model, but
to avoid inconsistency between GWAS model (20 PCs) and PRS model (10 PCs), we incorporated 20
PCs into the PRS regression model and re-calculated pseudo R^2 and AUCROC.

As you indicated, we also evaluated the performance of the base model which comprised just age,
sex, and the top 20 PCs as explanatory variables (Supplementary Table 9). As a result, the
performance of the base model was substantially lower than those of the full models that included
PRS. Therefore, subsequent analyses were continued with the full models.

Also, thank you for pointing out the typo in the Y-axis in Fig. 4 in the pre-revised manuscript. We have
corrected this as well (Fig.5. in the revised manuscript).

(Lines 527 – 543 in the main manuscript)

The performance of the PRS was measured as (1) Nagelkerke's pseudo R^2 obtained by modeling
age, sex, **the top 20 PCs** and normalized PRS, and (2) the area under the curve of the receiver
operator curve in the same model as Nagelkerke's pseudo R^2 . The best model/parameter set for each
combination model (BBJ, EUR, FIN, BBJ + EUR, BBJ + FIN, EUR + FIN, BBJ + EUR + FIN) was

determined by averaging Nagelkerke's pseudo R^2 (Supplementary Table 9). Using the best models
 and parameters determined in the derivation and validation cohorts, we calculated the PRSs and
 assessed their performance for the independent test cohort. To evaluate the PRS performance in the
 test cohort, we performed bootstrap over the samples in the test cohort with 5.0×10^4 replicates and
 assessed Nagelkerke's pseudo R^2 and the area under the curve of the receiver operator curve in
 each bootstrap group. Before the comparison of the combination models, we evaluated the
 performance of the base model, which included age, sex, and the top 20 PCs, and found that the
 base model showed the lowest performance among all of the models (Supplementary Table 9). Next,
 to compare the performance of the PRS derived from each combination model, we calculated the
 pairwise difference of Nagelkerke's pseudo R^2 (ΔR^2) between each pair of models (2 out of 7 models;
 21 combinations) and obtained the two-sided bootstrap P value by counting the number of $\Delta R^2 \leq 0$ or
 $\Delta R^2 > 0$ and then multiplied the lower value by 4×10^{-5} . The significance was set at $P = 2.3 \times 10^{-3}$
 (0.05/21).

(Fig. 5)

[Supplementary Table 9]

Summary statistics	Method	Parameter				Number of variants	Pseudo R^2			AUCROC		
		LD reference	P	r^2	Effect model		Median	L95	U95	Median	L95	U95
Base model*	NA	NA	NA	NA	NA	NA	0.069	0.036	0.095	0.662	0.649	0.667
BBJ	P+T	EAS	5.0.E-04	0.8	NA	2,103	0.124	0.092	0.148	0.719	0.707	0.726
EUR	P+T	EUR	5.0.E-06	0.5	NA	1,147	0.122	0.091	0.147	0.717	0.706	0.724
FIN	P+T	EUR	5.0.E-06	0.8	NA	201	0.102	0.071	0.127	0.700	0.688	0.707
BBJ/EUR	P+T	EUR	5.0.E-04	0.5	fixed	4,102	0.144	0.113	0.168	0.737	0.725	0.744
BBJ/FIN	P+T	EAS	5.0.E-06	0.2	fixed	155	0.131	0.100	0.156	0.725	0.713	0.732
EUR/FIN	P+T	EUR	5.0.E-04	0.5	random	3,799	0.127	0.096	0.151	0.722	0.711	0.729
BBJ/EUR/FIN	P+T	EUR	5.0.E-04	0.5	fixed	4,520	0.146	0.115	0.170	0.738	0.726	0.745

**Reviewer #2, Minor comment #1**

**The numbers of samples between the Japanese-only and BBJ in the trans-ancestry meta-**
**analysis differ. Please elaborate on this difference.**

We confirmed that the Japanese samples used in the Japanese GWAS and the trans-ancestry meta-
GWAS consist of the same number of case-controls (9,826 cases and 140,446 controls) as shown in
the pre-revised Supplementary Fig. 1, Supplementary Fig. 2, and manuscript lines 72-73 and 142-
145. However, we agree that the reason for the split of the BBJ samples in the PRS analysis should
be explained. We divided the BBJ control samples into two parts; one with the case samples was for
the PRS development, and the other was for the survival analysis, where we explored the association
of AF-PRS with long-term follow-up data. The samples in the PRS development were further divided
into the cross-validation cohort and test cohort, which comprised 6,890 / 49,451 cases / controls, and
2,953 / 21,194 cases / controls, respectively (Supplementary Fig. 15). The reason for separating the
samples for the PRS analysis was to ensure the independence of PRS derivation, validation, and test
samples, which enabled to avoid overfitting and overestimating the model and to evaluate the
accuracy against unseen data.

**Reviewer #2, Minor comment #2**

**Several analyses show $P < 2.2 \times 10^{-16}$. Please provide the actual P-values for these analyses.**

Thank you for raising this point. In the pre-revised manuscript, we performed a Student's *t*-test to
compare the performance of PRS between each pair of GWAS models, which showed exceedingly
low *P* values (almost 0). However, since we obtained the distribution of the PRS performance by a
bootstrapping procedure, we thought it was appropriate for testing the performance difference to use
the two-sided bootstrap *P* value (5×10^4 bootstrapping, where the minimum estimated *P* value was
$2 * 1 / (5 \times 10^4) = 4.0 \times 10^{-5}$: two-sided). We calculated the pairwise difference of Nagelkerke's
pseudo R^2 (ΔR^2) between each pair of scores (2 out of 7 models; 21 combinations) and obtained the
two-sided bootstrap *P* value by counting the number of $\Delta R^2 \leq 0$ or $\Delta R^2 > 0$ and then multiplied the
lower value by 4×10^{-5} . The significance was set at $P = 2.3 \times 10^{-3}$ ($0.05 / 21$ combinations). As a
result, for the PRS derived from a single population GWAS, as concordant with the population
specificity, the PRS derived from BBJ (pseudo $R^2 = 0.124$) showed a higher performance trend than
those from EUR (pseudo $R^2 = 0.122$, $P = 0.681$) and FIN (pseudo $R^2 = 0.102$, $P < 4 \times 10^{-4}$) despite
the smaller sample size, although there was no statistically significant difference in the PRS
performance between BBJ and EUR. Among the PRS derived from the meta-GWAS, we found
significant superiority of the PRS derived from BBJ + EUR compared to those from FIN + EUR
(pseudo $R^2 = 0.144$ in BBJ+EUR vs 0.131 in FIN+EUR, $P < 4 \times 10^{-4}$) even though the number of
cases was similar. The order of PRS performances did not change from the result in our pre-revised
manuscript and the PRS derived from the three studies with multi-ancestry and the largest sample
size (BBJ + EUR + FIN) showed the highest performance among all models (pseudo $R^2 = 0.146$, 95%
CI = 0.115 – 0.170, area under the curve of receiver operating characteristic = 0.738, 95% CI = 0.726
955 – 0.745).

We modified the corresponding descriptions and added Supplementary Fig. 6.

(Lines 191 – 201 in the main manuscript)

For the PRS derived from a single population GWAS, as concordant with the population specificity,
 the PRS derived from BBJ showed higher performance trend than those from EUR (pseudo $R^2 =$
 0.122 in EUR vs 0.124 in BBJ, $P = 0.681$) and FIN (pseudo $R^2 = 0.102$ in FIN, $P < 4 \times 10^{-4}$) despite
 the smaller sample size, although there was no statistically significant difference in the PRS
 performance between BBJ and EUR. Among the PRS derived from the meta-GWAS, we found
 significant superiority of the PRS derived from BBJ + EUR compared to those from FIN + EUR
 (pseudo $R^2 = 0.144$ in BBJ+EUR vs 0.131 in FIN+EUR, $P < 4 \times 10^{-4}$) even though the number of
 cases was similar. Among all models, the PRS derived from three studies with multi-ancestry and the
 largest sample size (BBJ + EUR + FIN) showed the highest performance (pseudo $R^2 = 0.146$, 95% CI
 $= 0.115 - 0.170$, area under the curve of receiver operating characteristic $= 0.738$, 95% CI $= 0.726 -$
 0.745).

(Lines 538 – 543 in the main manuscript)

Next, to compare the performance of the PRS derived from each combination model, we calculated
 the pairwise difference of Nagelkerke's pseudo R^2 (ΔR^2) between each pair of scores (2 out of 7
 models; 21 combinations) and obtained the two-sided bootstrap P value by counting the number of
 $\Delta R^2 \leq 0$ or $\Delta R^2 > 0$ and then multiplied the lower value by 4×10^{-5} . The significance was set at $P =$
 2.3×10^{-3} ($0.05/21$).

(Supplementary Fig. 6)

**Supplementary Fig. 6 | Pairwise comparison of PRS performance.** The distribution of the pairwise
difference of Nagelkerke's pseudo R^2 (Δ pseudo R^2 : Pseudo $R^2_{\text{Score } Y} - \text{Pseudo } R^2_{\text{Score } X}$, X and Y are
found at the top of the panel) between each pair of GWAS models are shown. The distributions were
obtained by bootstrapping 5.0×10^4 times. Two-sided bootstrap P values are presented.

**Reviewer #2, Minor comment #3**

**Line 197 "739 variants": Please explain where this number comes from, is this the number of**
**lead variants and proxies across the 150 loci?**

We apologize for not having provided information on this number. In the pleiotropy analysis, we
defined AF-associated variants as the lead variants, and their proxies determined by an r^2 value
threshold. Among them, we observed 739 variants overlapped with genome-wide significant variants
reported in the GWAS catalog. In the revised manuscript, as we mentioned in our response to your
major comment #3, we unified r^2 value of 0.8 to improve the consistency across our current analyses
and assessed the pleiotropic effects of the lead variants and their proxies with $r^2 > 0.8$. The results
slightly changed while we identified that 112 of 150 loci (74.7%) had the overlapped variants
associated with any diseases or traits, and we found 631 pairs of AF-associated variants and
phenotypes, where blood pressure-related traits were most frequently observed (13.5%), followed by
electrocardiogram-related (11.7%) and anthropometric traits (6.8%) (Supplementary Table 17).

We have made the following changes in this regard.

(Lines 83 – 88 in Supplementary Notes)

To characterize the AF-associated loci, we examined the pleiotropic effects of the identified lead
variants and proxies using the NHGRI-EBI GWAS catalog database. Of the 150 AF-associated loci,
**112 (74.7%) had at least one overlapping variant, and we found 631 pairs of AF-associated variants**
**and phenotypes, where blood pressure-related traits were most frequently observed (13.5%), followed**
**by electrocardiogram-related (11.7%) and anthropometric traits (6.8%) (Supplementary Table 17).**

(Lines 169 – 172 in Supplementary Notes)

To assess the pleiotropic effects of AF-associated variants, we selected the lead variants and their
proxies with $r^2 > 0.8$ across 150 AF-associated loci from the trans-ancestry meta-analysis, and
searched if these variants overlapped with other diseases or traits using NHGRI-EBI GWAS catalog
on February 8, 2019.

**Reviewer #2, Minor comment #4**

**Lines 207-209: Several clusters are mentioned but I have difficulties to understand why the**
**authors focus specifically on these.**

We apologize for this ambiguous clustering. This was an arbitrary functional classification based on
prior medical knowledge. In the revised manuscript, as we mentioned in the response to Reviewer #1
Major comment #6, we performed colocalization analysis to examine the overlap between AF-
associated variants and QTL and defined significant colocalization of a posterior probability > 0.8 as
applied in a previous study². Colocalization analysis was performed per 150 AF-associated loci, and

we found 25 AF-associated loci that showed significant colocalization with QTL (Supplementary Fig.
16 and Supplementary Table 18). As for clustering, the result of hierarchical clustering using the
posterior probabilities of colocalization analysis was shown in the figure, where QTLs were
categorized according to prior medical knowledge for readers' intuitive understanding. Of 56 pairs of
AF-associated loci and QTL with significant colocalization, 19.6% (11/56) were kidney-related traits,
16.0% (9/56) were blood pressure-related, and 16.0% (9/56) were metabolic traits.

Below are the corresponding texts and figure in the revised manuscript.

(Lines 90 – 97 in Supplementary Notes)

In particular, to explore the biological pathways related to AF development, we assessed
colocalization of AF-associated loci with quantitative trait loci (QTL) for clinical measurements such as
anthropometric, metabolic, kidney-related, and blood pressure data. Colocalization analysis was
performed per 150 AF-associated loci identified by the trans-ancestry meta-GWAS, and we identified
25 AF-associated loci that showed significant colocalization with QTL (Supplementary Fig. 16). Of 56
pairs of AF-associated loci and QTL with significant colocalization, 19.6% (11/56) were kidney-related
traits, 16.0% (9/56) were blood pressure-related, and 16.0% (9/56) were metabolic traits
(Supplementary Table 18).

(Lines 173 – 186 in Supplementary Notes)

For colocalization analysis, we used the summary statistics of QTL from the BBJ dataset¹⁷⁻¹⁹ for
clinical parameters of nine distinct categories; anthropometric (height and body mass index),
metabolic (total cholesterol, high-density-lipoprotein cholesterol, low-density-lipoprotein cholesterol,
triglyceride, blood sugar, and hemoglobin A1c), serum protein (total protein and albumin), kidney-
related (blood urea nitrogen, serum creatinine, estimated glomerular filtration rate, and uric acid),
electrolyte (sodium, potassium, and chloride), liver-related (total bilirubin, aspartate aminotransferase,
alanine aminotransferase, alkaline phosphatase, and γ -glutamyl transferase), other biochemical
(activated partial thromboplastin time, prothrombin time, creatine kinase, lactate dehydrogenase, and
C-reactive protein), hematological (white blood cell count, red blood cell count, hemoglobin,
hematocrit, and platelet), and blood pressure (systolic blood pressure, diastolic blood pressure, mean
arterial pressure, and pulse pressure). We extracted QTL data based on 150 AF-associated loci
identified by the trans-ancestry meta-GWAS, and colocalization was performed for each locus using
Coloc.abf from the Coloc R package²⁶. The threshold of a posterior probability > 0.8 was applied for
significant colocalization by referring to a prior study²⁷.

(Supplementary Fig. 16)

Supplementary Fig. 16 | Colocalization of quantitative trait loci with AF-associated signals.

Heat-map representation of the approximate Bayes factor posterior probability of AF and quantitative

trait loci to share a common causal variant at 150 AF-associated loci (coefficient H4 of the

colocalization analysis, represented on a white-red color scale). The rows show 36 quantitative traits

highlighted in each color of the 9 categories, and the columns show the chromosome and physical

position of the lead variant in AF-associated loci. AF, atrial fibrillation; QTL, quantitative trait loci.

[Supplementary Table 18 (partially extracted)]

Supplementary Table 18 | Colocalization of AF-associated loci with QTL for clinical measurement

Lead SNP			Locus		Phenotype	Posterior probability
CHR	POS	rsID	Start	End		
Novel loci						
1	16199051	rs9782984	15699051	16699051	AST	0.893
5	139703286	rs17118812	139032339	140214751	CK	0.940
					eGFR	0.876
					Height	0.992
6	76164589	rs12209223	75664589	77067954	HbA1c	0.860
11	3890059	rs7126870	3389383	4390223	HDL-C	0.993
12	12886027	rs10845620	12386027	13386027	BUN	0.999
18	77156537	rs8096658	76656537	77656537	eGFR	0.984
					sCr	0.997
					UA	0.837
19	48142746	rs11881441	47628418	48670757	Height	0.874
22	21999229	rs5754508	21499229	22499229	GGT	0.836
					HDL-C	0.898
					RBC	0.823
Known loci						
1	10796866	rs880315	10280727	11302468	AST	0.961
					DBP	0.999
					K	0.998
					MAP	0.999
					Na	0.975
					PP	0.999
2	61676940	rs2694635	61071295	62306999	WBC	0.841
2	65284231	rs2540949	64735333	65886462	PP	0.908
3	12841804	rs7650482	12259080	12841804	CK	0.995

**Reviewer #2, Minor comment #5**

**Line 229: The way this sentence is currently phrased it seems like the mouse knockdown was**
**part of the current study, while this is not the case.**

We apologize for the misleading sentence. As you pointed out, we did not conduct the mice
knockdown experiments in our study. To clarify this, we have rephrased the text as follows.

(Lines 107 – 112 in Supplementary Notes)

Among them, the regulation of cell adhesion was a previously unreported pathway ($P = 1.6 \times 10^{-10}$),
including *ZMIZ1* gene, the nearest for rs1769758, which was reported to regulate the activity of
various transcription factors related to vascular development, androgen receptor coregulation,
SMAD3 regulation, and coactivation of p53²²⁻²⁴. Further, it has been reported that knockdown of
*ZMIZ1* in mice resulted in severe defects in the reorganization of the yolk sac vascular plexus and cell
proliferation²⁵.

**Reviewer #2, Minor comment #6**

**Line 251 TWAS association between AF and IL6R: Please include a bit more information, such as**
**variant, P-value.**

We apologize for not showing this information in our pre-revised manuscript. We have added the
detailed information on the association between *IL6R* and AF identified by TWAS in the text, such as
the TWAS result (the effect size and P value) as well as the data on the variant used in the prediction
model of *IL6R*. Accordingly, we have modified the description as follows.

(Lines 138 – 141 in the main manuscript)

Intriguingly, we found that *IL6R* is one of the candidate genes associated with AF in the atrial
appendage ($\beta_{IL6R} = 0.221$, $P = 2.147 \times 10^{-9}$). The prediction model of *IL6R* expression included
rs10908837 (MAF = 42%, $\log_{10} BF = 7.237$ in the trans-ancestry meta-analysis), which is located in
an intron of *IL6R*.

(Lines 287 – 294 in the main manuscript)

Transcriptome-wide analysis linked AF-associated loci to target genes, and particularly revealed that
*IL6R* is one of the candidate genes associated with AF. Despite increasing evidence for the role of
inflammation in the AF pathophysiology³⁰, only suggestive association between *IL6R* and AF ($P = 5.0$
$\times 10^{-4}$) has so far been reported³¹, and the genetic contribution of inflammatory process to AF
development has not been fully elucidated. Our transcriptome-wide analysis revealed a significant
association between *IL6R* and AF development, shedding light on the inflammatory signaling as a key
pathway in the pathogenesis of AF as well as a therapeutic target.

**Reviewer #2, Minor comment #7**

**Lines 315-317: Low numbers of heart failure deaths are provided as potential reason for not**
**finding a significant effect. How do these numbers compare to CVD and stroke deaths (which**
**both show a significant association)?**

Thank you very much for raising this important point. Among 132,737 individuals for whom mortality
data was available, the number of events for heart failure deaths was 1,093 (0.82%), which was
approximately half of stroke events (n=2,012) and even less than 20% of cardiovascular events
(n=6,877). Thus, it was assumed that the standard deviation for heart failure death was larger due to
the smaller number of events, which resulted in a relatively wide confidence interval that did not reach
statistical significance. Therefore, we added a description of the differences in the number of events
and their impact on the association analysis in the revised manuscript as follows.

(Lines 228 – 235 in the main manuscript)

In contrast to evidence from clinical studies, the association between AF-PRS and heart failure death
did not reach statistical significance in the present study (HR [95% CI] = 1.05 [0.95 – 1.16], $P = 0.37$).
Among 132,737 individuals for whom mortality data was available, the number of events for heart
failure death was 1,093 (0.82%), which was approximately half of stroke events (n=2,012) and even
less than 20% of cardiovascular events (n=6,877). Thus, it was assumed that the standard deviation
for heart failure death was larger due to the smaller number of events in our cohort, which resulted in
a relatively wide confidence interval that might make it difficult to reach statistical significance.

**Reviewer #2, Minor comment #8**

**Methods line 431: this sentence described loci instead of variants being in LD (or not). It is**
**unclear to me what the authors mean with this, please clarify.**

We apologize for this ambiguous explanation. We defined a locus as follows. (1) extracted genome-
wide significant variants ($P < 5 \times 10^{-8}$) from the association result, (2) added 500 Mb to both sides of
these variants, and (3) merged overlapping regions. If the locus did not contain coordinates with
previously reported genome-wide significant variants (i.e., all variants with $P < 5 \times 10^{-8}$ in the
previously reported locus), the region was annotated as being novel.

We modified the description as follows.

(Lines 398 – 402 in the main manuscript)

We defined a locus as follows: (1) extracted genome-wide significant variants ($P < 5 \times 10^{-8}$) from the
association result, (2) added 500 Mb to both sides of these variants, and (3) merged overlapping
regions. If the locus did not contain coordinates with previously reported genome-wide significant
variants (i.e., all variants with $P < 5 \times 10^{-8}$ in the previously reported locus), the region was annotated
as being novel.

**Reviewer #2, Minor comment #9**

**Credible set analysis line 488: please explain why you did not use FIN + BBJ**

Thank you for raising this point. Our aim of the 99% credible set analysis was to examine the fine-
mapping effect in our trans-ancestry meta-analyses. When we compared the fine-mapping effect of

trans-ancestry meta-analysis, the sample size in FIN + BBJ ($n = 213,894$) was much smaller than
those in the other combinations (EUR + BBJ; $n = 1,181,108$, EUR + FIN; $n = 1,094,458$, and BBJ +
1134 EUR + FIN; $n = 1,244,730$), which may lead to difficulty in distinguishing the sample size effect from
1135 the fine-mapping effect of a trans-ancestry meta-analysis. Accordingly, we thought that it was
1136 appropriate to see the effect of a trans-ancestry meta-analysis by comparing three GWASs (EUR +
BBJ, EUR + FIN, and BBJ + EUR + FIN), where the sample sizes were similar.

We modified the description regarding this issue as follows.

(Lines 153 – 162 in Supplementary Notes)

To assess whether the 99% credible set derived from the trans-ancestry meta-analysis narrowed
down the causal variants, we performed two additional meta-analyses (EUR + BBJ and EUR + FIN)
using the MANTRA algorithm and constructed 99% credible sets for 150 AF-associated loci identified
by the trans-ancestry meta-analysis. **When we compared the fine-mapping effect of trans-ancestry
meta-analysis, the sample size in FIN + BBJ ($n = 213,894$) was much smaller than those in the other
combinations (EUR + BBJ; $n = 1,181,108$, EUR + FIN; $n = 1,094,458$, and BBJ + EUR + FIN; $n =$
$1,244,730$), which may lead to difficulty in distinguishing the sample size effect from the fine-mapping
effect of a trans-ancestry meta-analysis. Accordingly, we examined the effect of a trans-ancestry
meta-analysis by comparing three meta-GWASs (EUR + BBJ, EUR + FIN, and BBJ + EUR + FIN).**

**Reviewer #2, Minor comment #10**

**PRS methods section: It is not entirely clear to me how many PRS were constructed. Is it 6**
**subgroups x 5 P-value thresholds x 3 r^2 thresholds = 90, or many more because different**
**subsets of samples were used for test/validation purposes?**

We apologize for not providing this essential information in our pre-revised manuscript. We derived
PRSs for all cohort combinations (BBJ, EUR, FIN, BBJ + EUR, BBJ + FIN, EUR + FIN, and BBJ +
1155 EUR + FIN) using multiple parameters to obtain the maximum performance of a PRS model, but
these parameters are slightly different among cohorts, which resulted in the complicated and
misleading method of the PRS construction. Specifically, in addition to 5 P values and 3 r^2 thresholds
for SNP selection, we used LD information (1KG East Asian (EAS) and 1KG European (EUR)
population) according to each cohort population; (1) 1KG EAS for cohort with only East Asian
population (BBJ), (2) 1KG EUR for cohorts with only European population (EUR, FIN, and EUR +
FIN), (3) both 1KG EAS and 1KG EUR for cohorts with multiple ancestries (BBJ + EUR, BBJ + FIN,
BBJ + EUR + FIN). Furthermore, in four cases of the meta-analysis (BBJ + EUR, BBJ + FIN, EUR +
FIN, and BBJ + EUR + FIN), we set the parameter for the meta-analysis models (i.e., the fixed effect
and the random effect models). We also calculated PRSs using LDpred2 algorithm as Reviewer #3
suggested, in which we ran the LDpred2-grid model with the parameters such as p (proportion of
causal variants) in a sequence of 5 values from 10^{-4} to 1 on a log-scale and sparse option (true or
false). We did not tune the parameter for the SNP heritability h^2 because the different samples in each
derivation cohort did not enable us to determine the optimal h^2 . For the parameters of LD information
and the meta-analysis models, we used the same method as above. Finally, we constructed a total of

376 PRSs (Supplementary Table 8); the PRS with the best performance for each cohort is shown in
Fig. 5.

Below are the corresponding text and figure.

(Lines 503 – 526 in the main manuscript)

First, we divided our dataset into three groups: (i) a discovery group to derive and validate PRS
(6,890 cases and 49,451 controls), (ii) a test group to assess PRS performance (2,953 cases and
21,194 controls), and (iii) a group for the survival analysis (70,645 controls) (Supplementary Fig. 11).
To secure independence between the PRS derivation and validation, we used a ten-fold cross-
validation approach. Next, we randomly split a discovery group into ten subgroups and used nine of
these subgroups for PRS derivation and the remaining one for PRS validation. For each derivation
cohort, we performed GWASs in combinations with two European GWASs: (1) a population-specific
GWAS (BBJ, EUR, FIN), (2) European meta-GWAS (EUR + FIN), and (3) the trans-ancestry meta-
GWAS (BBJ + EUR, BBJ + FIN, BBJ + EUR + FIN). Meta-analyses were performed using the fixed
effect and the random effect models by METASOFT software. We derived PRS using the pruning and
thresholding method and the Ldpred2 algorithm. For the pruning and thresholding method, in addition
to the meta-analysis models, we applied the P value thresholds as 0.5, 5.0×10^{-2} , 5.0×10^{-4} , 5.0×10^{-6} ,
and 5.0×10^{-8} , and the r^2 thresholds as 0.8, 0.5, and 0.2. For Ldpred2, the variants were restricted
to HapMap3 SNPs as recommended⁴⁷, and we ran the Ldpred2-grid model with the parameters: p
(proportion of causal variants) in a sequence of 5 values from 10^{-4} to 1 on a log-scale and sparse
option (true or false). We did not tune the parameter for the SNP heritability h^2 because the different
samples in each derivation cohort did not enable us to determine the optimal h^2 . For the LD reference,
we used 1KG East Asian (EAS) and 1KG European (EUR) populations according to each cohort
population; (1) 1KG EAS for a cohort with only East Asian (BBJ), (2) 1KG EUR for cohorts with only
European (EUR, FIN, and EUR + FIN), (3) both 1KG EAS and 1KG EUR for cohorts with multiple
ancestries (BBJ + EUR, BBJ + FIN, BBJ + EUR + FIN). Subsequently, we calculated PRSs in the
withheld validation cohorts, and repeated this procedure ten times by changing the withheld validation
cohorts. Finally, we constructed 376 PRSs in total; the PRS with the best performance for each cohort
is shown in Fig. 5.

[Supplementary Table 8 (partially extracted)]

Supplementary Table 8 The performance of AF-PRS in the cross-validation cohort											
Summary statistics	Method	Parameter						Median number of variants	Performance		
		P	r ²	p	sparse	LD reference	Effect model		Pseudo R ²	AUCROC	
BBJ	P+T	5.00E-04	0.5	-	-	EAS	NA	1,374	0.128	0.723	
BBJ	P+T	5.00E-04	0.8	-	-	EAS	NA	1,985	0.128	0.723	
BBJ	P+T	5.00E-06	0.2	-	-	EAS	NA	130	0.128	0.722	
BBJ	P+T	5.00E-08	0.2	-	-	EAS	NA	65	0.126	0.721	
BBJ	P+T	5.00E-06	0.5	-	-	EAS	NA	206	0.126	0.720	
BBJ	P+T	5.00E-06	0.8	-	-	EAS	NA	350	0.126	0.720	
BBJ	P+T	5.00E-04	0.2	-	-	EAS	NA	1,070	0.125	0.720	
BBJ	P+T	5.00E-08	0.5	-	-	EAS	NA	111	0.124	0.718	
BBJ	P+T	5.00E-08	0.8	-	-	EAS	NA	205	0.123	0.717	
BBJ	P+T	5.00E-02	0.8	-	-	EAS	NA	86,953	0.109	0.705	
BBJ	P+T	5.00E-02	0.5	-	-	EAS	NA	63,015	0.103	0.700	
BBJ	P+T	5.00E-02	0.2	-	-	EAS	NA	46,787	0.098	0.695	
BBJ	P+T	5.00E-01	0.8	-	-	EAS	NA	739,368	0.096	0.692	
BBJ	P+T	5.00E-01	0.5	-	-	EAS	NA	494,448	0.093	0.689	
BBJ	P+T	5.00E-01	0.2	-	-	EAS	NA	305,173	0.092	0.688	
BBJ	LDpred2	-	-	1.00E-04	TRUE	EAS	NA	923,378	0.078	0.670	
BBJ	LDpred2	-	-	1.00E-04	FALSE	EAS	NA	923,378	0.078	0.670	
BBJ	LDpred2	-	-	1.00E-03	TRUE	EAS	NA	923,378	0.076	0.669	
BBJ	LDpred2	-	-	1.00E-03	FALSE	EAS	NA	923,378	0.075	0.669	
BBJ	LDpred2	-	-	1.00E-02	TRUE	EAS	NA	923,378	0.075	0.669	
BBJ	LDpred2	-	-	1.00E-02	FALSE	EAS	NA	923,378	0.075	0.669	

Reviewer #2, Minor comment #11

I noticed that throughout the manuscript the authors use 10 or 20 principal components. How were they chosen and why are different numbers used?

We apologize for this inconsistency. In the pre-revised manuscript, we utilized 20 PCs for the adjustment of the Japanese GWAS, while 10 PCs were used as covariates in the PRS analyses. As you mentioned, since this difference could be confusing, we re-performed all of these analyses where we standardized the number of PCs to 20. As for the association analysis between AF-PRS and stroke phenotypes, we added the use of anticoagulants or antiplatelets as a covariate in the regression model as suggested by Reviewer #3. As a result, we got the same results as when we used 10 PCs.

Below are the corresponding text in the revised manuscript.

(Lines 527 – 529 in the main manuscript)

The performance of the PRS was measured as (1) Nagelkerke's pseudo R^2 obtained by modeling age, sex, **the top 20 PCs** and normalized PRS, and (2) the area under the curve of the receiver operator curve in the same model as Nagelkerke's pseudo R^2 .

(Lines 545 – 554 in the main manuscript)

To assess the association between AF-PRS and age at AF onset, we extracted AF case samples with **available data on age at AF onset** ($n = 7,458$, the median age of AF onset was 63 years of age (IQR 56 – 71)), and **constructed a linear regression model of age at AF onset including AF-PRS as a dichotomous variable (individual with high PRS [the top 1%, 5%, 10%, and 20%] vs. those with the remaining PRS) to estimate** the difference in the age of AF onset between them adjusted by sex and **the top 20 PCs**. For the association analysis with stroke phenotypes, we **performed a logistic regression analysis adjusted by the use of anticoagulants or antiplatelets in addition to age, sex, and the top 20 PCs, because antithrombotic therapy is associated with a decreased risk of ischemic stroke as well as an increased risk of hemorrhagic stroke.**

(Lines 567 in the main manuscript)

The Cox proportional hazards model was adjusted for sex, age, the top 20 PCs, and disease status.

Reviewer #2, Minor comment #12

Figure 5: panels c and e contain colors (grey vs red) that seem to be significant vs non-significant results but this is not specified.

We apologize for not explaining the colors in pre-revised Fig. 5. We have added an explanation in the corresponding figure legend (Fig. 6 in the revised manuscript), as shown below.

(Fig. 6b,d)

Fig. 6 | Impact of AF-PRS on long-term cardiovascular mortality. b, Association between AF-PRS and stroke phenotypes in individuals without AF. Data are presented as estimated Ors and 95% CI for 1 s.d. increase in AF-PRS. Significant and non-significant associations are shown in orange and grey, respectively. **D,** Effects of AF-PRS on long-term mortality. Data are presented as estimated HRs and

1241 95% Cis for a 1 s.d. increase in AF-PRS. Significant and non-significant associations are shown in
orange and grey, respectively.

**Reviewer #3:**

**Remarks to the Author:**

**A. Summary of key results**

**The authors report results from the largest GWAS of atrial fibrillation (AF) ever performed in**
**the Japanese population, including 9826 cases and 150272 controls. The GWAS identified 5**
**new susceptibility loci. A trans-ancestry meta-analysis including published GWAS for a total**
**of 77690 AF cases identifies a total of 35 novel loci. The investigators then follow-up on the**
**GWAS finding using standard annotation of loci and identify several new genes of potential**
**interest. No functional studies are reported. A polygenic risk score (PRS) is then derived from**
**the trans-ancestry meta-analysis and is shown to be associated with an increased risk of long-**
**term cardiovascular and stroke mortality, and segregated individuals with cardioembolic**
**stroke in undiagnosed AF patients.**

**B. Originality and significance**

**Although prior larger GWAS of AF have been reported, this work reports the largest GWAS in**
**Japanese, and should as such be considered novel given the importance of diversity in**
**genetics research especially with regards to potential clinical use. The trans-ancestry meta-**
**analysis with the current sample size is novel.**

**C. Data & methodology**

**- Methodology. Generally very appropriate and well described, with the following minor**
**exceptions:**

We sincerely appreciate your opinion and feedback and thank you for your detailed evaluation of our
manuscript. We took your concerns seriously, evaluated them, made our utmost efforts to address
them, and prepared a revised manuscript.

As you indicated, several results of less importance have been moved from the main manuscript to
the supplementary note. Meanwhile, leveraging the Biobank Japan second cohort, which is another
Japanese cohort, and GTEx.v8, the latest tissue-specific gene expression database, we have
ensured and expanded our results. Additionally, to strengthen the results, we have added wet
experiments using human induced pluripotent stem cell-derived cardiomyocytes (please refer to
**Major changes 3** at the beginning of this response letter) to assess the importance of the
transcription factor that we identified.

Please find below our point-to-point responses to each of your comments.

**Reviewer #3, Comment #1**

**Assuming sample independence, the authors should test the PRS of Khera et al (PMID**
**30104762;**
**https://personal.broadinstitute.org/ryank/AtrialFibrillation_PRS_LDpred_rho0.003_v3.zip) in**
**their Japanese testing cohort. How does it perform in the Japanese population, compared to**
**the European testing set reported by Khera et al.? Does the PRS derived from the current**
**Japanese GWAS perform better?**

Thank you for your suggestion. We agree that evaluating the performance of this benchmark AF-PRS
by Khera et al. in the Japanese population is essential to prove the superiority of our PRS. However,
we confirmed that the samples in Khera et al.'s GWAS partially overlapped with our samples because
Khera et al. derived their PRS using summary statistics of a previous GWAS published in *Nature*
*Genetics* 2017⁷, in which AF-GWAS was performed using the samples including BioBank Japan (n =
4,130). Although there was an issue of overfitting due to the sample overlap, we evaluated the
performance of Khera et al.'s PRS in our PRS test cohort using the score from the link above, which
demonstrated a relatively high predictive performance (Response Table 1), but did not outperform our
PRS (Supplementary Table 9). Since there was a sample overlap and this result is not conflicting with
our initial results, we did not change the text.

Response Table 1 | The performance of AF-PRS by Khera et al in the test cohort

Summary statistics	Method	Number of variants	Pseudo R ²			AUCROC		
			Median	L95	U95	Median	L95	U95
AFGen	LDpred	6,730,541	0.132	0.101	0.156	0.725	0.713	0.732

AF, atrial fibrillation; PRS, polygenic risk score; Pseudo R², Nagelkerke's pseudo R²; AUCROC, area
under the curve of the receiver operator curve; L95, lower limit of 95% confidence interval; U95, upper
limit of 95% confidence interval

**Reviewer #3, Comment #2**

**PRS derivations have been performed using classic thresholding and pruning. The authors**
**can consider also using LDpred or LDpred2 for PRS derivation.**

Thank you very much for your constructive suggestion. As you mentioned, PRS derivation methods
are currently evolving from a pruning and thresholding method to better ones using a Bayesian
method such as the LDpred algorithm, and we agree that they need to be tested. Here, we added the
evaluation of the PRS performance generated by LDpred2, the latest LDpred algorithm, where
variants were restricted to HapMap3 SNPs as recommended¹⁵, and we ran the LDpred2-grid model
with the parameters such as p (proportion of causal variants) in a sequence of 5 values from 10⁻⁴ to 1
on a log-scale and sparse option (true or false). We did not tune the parameter for the SNP heritability
*h*² because the different samples in each derivation cohort did not enable us to determine the optimal
*h*². As for LD references and the meta-analysis models, we used the same parameters as used in the
pruning and thresholding method. As a result, PRS derivated by LDpred2 did not show the best
performance in any validation cohorts for all combinations of GWAS models (Supplementary Table 8).

In other words, LDPred2 did not show better performance than the pruning and thresholding method
 in the validation cohort. Next, we validated the PRS performance in the test cohort using the best
 performing hyperparameter/method for each PRS-derivation GWAS combination, and found that the
 PRS derived from the trans-ancestry GWAS of BBJ + EUR + FIN showed the best performance.
 Therefore, the subsequent analyses were conducted using this PRS.
 We added the following description to the manuscript.

(Lines 513 – 524 in the main manuscript)

We derived PRS using the pruning and thresholding method and the LDpred2 algorithm. For the
 pruning and thresholding method, in addition to the meta-analysis models, we applied the *P* value
 thresholds as 0.5, 5.0×10^{-2} , 5.0×10^{-4} , 5.0×10^{-6} , and 5.0×10^{-8} , and the r^2 thresholds as 0.8, 0.5,
 and 0.2. For LDpred2, the variants were restricted to HapMap3 SNPs as recommended⁴⁷, and we ran
 the LDpred2-grid model with the parameters: *p* (proportion of causal variants) in a sequence of 5
 values from 10^{-4} to 1 on a log-scale and sparse option (true or false). We did not tune the parameter
 for the SNP heritability h^2 because the different samples in each derivation cohort did not enable us to
 determine the optimal h^2 . For the LD reference, we used 1KG East Asian (EAS) and 1KG European
 (EUR) population according to each cohort population; (1) 1KG EAS for cohort with only East Asian
 (BBJ), (2) 1KG EUR for cohorts with only European (EUR, FIN, and EUR + FIN), (3) both 1KG EAS
 and 1KG EUR for cohorts with multiple ancestries (BBJ + EUR, BBJ + FIN, BBJ + EUR + FIN).

 [Supplementary Table 8 (partially extracted)]

Summary statistics	Method	Parameter						Median number of variants	Performance	
		P	r^2	p	sparse	LD reference	Effect model		Pseudo R ²	AUCROC
BBJ	P+T	5.00E-04	0.5	-	-	EAS	NA	1,374	0.128	0.723
BBJ	P+T	5.00E-04	0.8	-	-	EAS	NA	1,985	0.128	0.723
BBJ	P+T	5.00E-06	0.2	-	-	EAS	NA	130	0.128	0.722
BBJ	P+T	5.00E-08	0.2	-	-	EAS	NA	65	0.126	0.721
BBJ	P+T	5.00E-06	0.5	-	-	EAS	NA	206	0.126	0.720
BBJ	P+T	5.00E-06	0.8	-	-	EAS	NA	350	0.126	0.720
BBJ	P+T	5.00E-04	0.2	-	-	EAS	NA	1,070	0.125	0.720
BBJ	P+T	5.00E-08	0.5	-	-	EAS	NA	111	0.124	0.718
BBJ	P+T	5.00E-08	0.8	-	-	EAS	NA	205	0.123	0.717
BBJ	P+T	5.00E-02	0.8	-	-	EAS	NA	86,953	0.109	0.705
BBJ	P+T	5.00E-02	0.5	-	-	EAS	NA	63,015	0.103	0.700
BBJ	P+T	5.00E-02	0.2	-	-	EAS	NA	46,787	0.098	0.695
BBJ	P+T	5.00E-01	0.8	-	-	EAS	NA	739,368	0.096	0.692
BBJ	P+T	5.00E-01	0.5	-	-	EAS	NA	494,448	0.093	0.689
BBJ	P+T	5.00E-01	0.2	-	-	EAS	NA	305,173	0.092	0.688
BBJ	LDpred2	-	-	1.00E-04	TRUE	EAS	NA	923,378	0.078	0.670
BBJ	LDpred2	-	-	1.00E-04	FALSE	EAS	NA	923,378	0.078	0.670
BBJ	LDpred2	-	-	1.00E-03	TRUE	EAS	NA	923,378	0.076	0.669
BBJ	LDpred2	-	-	1.00E-03	FALSE	EAS	NA	923,378	0.075	0.669
BBJ	LDpred2	-	-	1.00E-02	TRUE	EAS	NA	923,378	0.075	0.669
BBJ	LDpred2	-	-	1.00E-02	FALSE	EAS	NA	923,378	0.075	0.669

**Reviewer #3, Comment #3**

**The association of PRS with hemorrhagic stroke is interesting. What is the mechanism? Can it**
 **be mediated by anticoagulant therapy? Did the authors correct for antithrombotic drug use?**

Thank you very much for your insightful remarks. Atrial fibrillation can also cause hemorrhagic stroke,
 following ischemic stroke, especially acute cardioembolic stroke^{16,17}. As you mentioned,
 anticoagulation therapy is known to be one of the risk factors for the hemorrhagic transition from
 infarction, which may play a key role in the association between AF-PRS and hemorrhagic stroke.

Therefore, we assessed the association between AF-PRS and stroke phenotypes using a logistic
 regression model, in which we added the use of anticoagulants or antiplatelets as one of the
 covariates. Then, we found that the association with hemorrhagic stroke reached a nominal threshold
 ($P = 0.045$), but no statistically significant association with hemorrhage stroke was observed after
 Bonferroni correction ($P < 0.05/6$) (Fig. 6b).

We modified the description regarding this issue in the revised manuscript.

(Lines 209 – 213 in the main manuscript)

We performed logistic regression analysis in 121,351 control samples in our dataset, and found
 significant associations of the PRS with increased risks of cerebral infarction (OR [95% CI] = 1.042
 [1.018 – 1.065], $P = 4.0 \times 10^{-4}$) and cardioembolic stroke (OR [95% CI] = 1.355 [1.126 – 1.630], $P =$
 1.3×10^{-3}) after Bonferroni correction (Fig. 6b).

(Lines 551 – 557 in the main manuscript)

For the association analysis with stroke phenotypes, we performed a logistic regression analysis
 adjusted by the use of anticoagulants or antiplatelets in addition to age, sex, and the top 20 PCs,
 because antithrombotic therapy is associated with a decreased risk of ischemic stroke as well as an
 increased risk of hemorrhagic stroke. We selected the control samples with available data on
 antithrombotic therapy ($n = 121,351$), among whom, we found 14,120 stroke phenotypes: 8,547
 cerebral infarction, 111 cardioembolic stroke, 1,429 atherothrombotic infarction, 1,230 lacunar
 infarction, 1,061 cerebral hemorrhage, and 879 subarachnoid hemorrhage.

(Fig. 6b)

**Reviewer #3, Comment #4**

**How are gene names selected for locus identification in Tables 1 and 2 and Figure 1? More**
**generally, the authors should consider using opentarget to map locus to genes in addition to**
**the described annotation.**

We appreciate this advice to make our manuscript more informative. In the pre-revised manuscript,
we presented nearby genes for each locus conventionally, while the gene that maps to the locus
should be determined more functionally as you indicated. Therefore, in addition to listing the nearby
genes in the “Nearby gene” column, we have also added a column, “Annotated gene”, which contains
putative causal genes proposed by Open Targets.

We modified the following description in the revised text and tables.

(Lines 403 – 405 in the main manuscript)

We mapped variants to the nearby genes as well as the functionally annotated genes using Open
Targets (<https://www.opentargets.org/>), in which the pair of variant and gene with the highest Variant-
to-Gene score was selected.

[Table 1]

CHR	Position (hg19)	rsID	REF	ALT	AAF			Beta	SE	P	Nearby gene	Annotated gene*	Functional consequence
					BBJ	gnomAD							
						EUR (non-Finnish)	EUR (finnish)						
6	152466619	rs202030113	T	C	0.012	0	0	0.352	0.063	1.90×10^{-8}	SYNE1	SYNE1	Intronic
12	104471663	rs2930856	C	T	0.579	0.861	0.854	0.090	0.015	5.80×10^{-9}	HCFC2	HCFC2	Intronic
16	30619745	rs1055894680	C	T	0.001	<0.001	0.001	1.215	0.158	1.58×10^{-14}	ZNF689	ZNF689	Intronic
X	23399501	rs73205368	T	C	0.284	0.045	0.015	0.089	0.012	7.50×10^{-13}	PTCHD1	PTCHD1	Intronic
X	137790580	rs778479352	T	C	0.002	0	0	0.692	0.075	1.62×10^{-20}	FGF13	FGF13	Intronic

[Table 2 (partially extracted)]

CHR	POS(hg19)	REF	ALT	rsID	Nearby gene	Annotated gene*	Functional consequence	log ₁₀ BF
1	918617	G	A	rs4970418	PERM1, HES4	PLEKHN1	intergenic	7.647
1	16199051	T	C	rs9782984	MIR5096	SPEN	ncRNA intronic	6.970
1	39385714	G	A	rs75414548	RHBDL2	NDUF55	intronic	6.439
1	100149308	G	A	rs1933723	PALMD	PALMD	intronic	6.298
4	71776935	A	C	rs12512502	MOB1B	DCK	intronic	6.547
4	83910712	T	G	rs6841049	LIN54	LIN54	intronic	7.628
5	139703286	T	C	rs17118812	PFDN1, HBEGF	PFDN1	intergenic	8.078
6	22598259	C	T	rs7766436	HDGFL1, LOC105374972	HDGFL1	intergenic	6.187
6	76164589	C	A	rs12209223	LOC101928540	FILIP1	ncRNA exonic	10.909
135119089	C	T	rs4896104	LOC101928304, ALDH8A1	ALDH8A1	intergenic	7.975
105612736	A	G	rs2727757	CDHR3	CDHR3	intronic	6.943
118863412	A	T	rs17430357	EXT1	EXT1	intronic	6.245
119181794	G	A	rs17303101	PAPPA, ASTN2	TRIM32	intergenic	6.328
32772734	C	T	rs11527634	CCDC7	CCDC7	intronic	7.692
10	50485434	G	A	rs76460895	C10orf128, C10orf71-AS1	TMEM273	intergenic	7.810
10	80898969	G	T	rs1769758	ZMIZ1	ZMIZ1	intronic	7.505
11	3890059	C	T	rs7126870	STIM1	STIM1	intronic	6.216
11	14036189	G	A	rs10500790	SPON1	SPON1	intronic	10.831
11	95089882	C	T	rs517938	LOC100129203, FAM76B	SESN3	intergenic	6.157
12	12886027	G	A	rs10845620	APOLD1	GPR19	intronic	6.031
12	104492003	A	G	rs2629755	HCFC2	HCFC2	intronic	9.505
12	110082115	T	C	rs1344543	MVK, FAM222A	UBE3B	intergenic	8.675
12	113196733	G	A	rs11614295	RPH3A	RPH3A	intronic	8.669
13	22111521	C	A	rs11841562	MICU2	MICU2	intronic	6.240
13	74520186	T	A	rs1886512	KLF12	KLF12	intronic	6.943
16	15902715	G	A	rs9284324	MYH11	MYH11	intronic	7.200
18	77156537	C	G	rs8096658	NFATC1	NFATC1	intronic	7.120
19	48142746	A	C	rs11881441	BICRA	NOP53	intronic	7.384
20	36841914	G	A	rs3746471	KIAA1755	KIAA1755	exonic	9.429
22	21999229	C	G	rs5754508	SDF2L1	CCDC116	downstream	6.077
22	42189407	T	G	rs139557	MEI1	MEI1	intronic	6.577
X	23399501	T	C	rs73205368	PTCHD1	PTCHD1	intronic	10.514
X	137418967	C	A	rs1891095	ZIC3, LINC00889	ZIC3	intergenic	9.242

Reviewer #3, Comment #5

In the trans-ancestry meta-analysis, the authors included the GWAS of Nielsen et al (PMID 30061737) but not of Roselli et al (PMID 29892015). The latter publication reported a meta-analysis of both Roselli and Nielsen including non-overlapping samples. Do the authors have access to this latter larger meta-analysis summary statistics and can it be included? More importantly, did the authors make sure the 35 novel loci were not identified by this combined meta-analysis of Roselli and Nielsen (supplementary table 16 in PMID 29892015: https://www.ncbi.nlm.nih.gov/pmc/articles/PMC6136836/bin/NIHMS986675-supplement-Supplementary_Table_16.xlsx).

Thank you for raising this point. As you suggested, the study by Roselli et al. reported the largest meta-analysis for AF-GWAS thus far, while their GWAS included the samples from the BioBank Japan. Therefore, to avoid sample overlap, we used the summary statistics from Nielsen et al.'s GWAS, which included only individuals of European descent. Please also refer to reviewer #1, additional comment #9.

Further, we confirmed that the 35 novel loci in our study were not previously identified by the combined meta-analyses of Roselli and Nielsen's GWASs by checking Supplementary Table 16 in PMID 29892015.

Also, we have to apologize for the confusion in Table 2, in which the summary statistics of EUR and
 FIN GWAS included genome-wide significant variants, but these were typographical errors. These loci
 actually did meet a genome-wide significance level ($P < 5.0 \times 10^{-8}$) just in the meta-analysis.
 Furthermore, to make sure of these novel loci, we referred to another database, the Cardiovascular
 Disease Knowledge Portal (<http://www.broadcvdi.org/>), and confirmed that all of these loci were not
 previously reported.
 We corrected the summary statistics for each population GWAS in Table 2 as shown below.

[Table 2 (partially extracted)]

CHR	POS(hg19)	BBJ				EUR				FIN			
		AAF	Beta	SE	P	AAF	Beta	SE	P	AAF	Beta	SE	P
1	918617	0.076	0.062	0.028	2.90×10 ⁻²	0.167	0.044	0.010	7.54×10 ⁻⁶	0.175	0.102	0.025	4.10×10 ⁻⁵
1	16199051	0.723	-0.075	0.017	1.54×10 ⁻⁵	0.883	-0.055	0.014	8.40×10 ⁻⁵	0.835	-0.035	0.025	1.72×10 ⁻¹
1	39385714	0.060	0.068	0.032	3.49×10 ⁻²	0.077	0.068	0.015	4.35×10 ⁻⁶	0.059	0.106	0.040	8.18×10 ⁻³
1	100149308	0.698	0.034	0.016	3.75×10 ⁻²	0.677	0.036	0.007	5.21×10 ⁻⁷	0.664	0.027	0.020	1.82×10 ⁻¹
4	71776935	0.621	-0.039	0.016	1.35×10 ⁻²	0.623	-0.034	0.007	1.27×10 ⁻⁶	0.542	-0.030	0.019	1.89×10 ⁻²
4	83910712	0.342	-0.044	0.016	5.58×10 ⁻³	0.567	-0.037	0.007	6.01×10 ⁻⁸	0.608	-0.019	0.019	3.19×10 ⁻³
5	139703286	0.385	0.059	0.015	1.37×10 ⁻⁴	0.276	0.036	0.007	1.86×10 ⁻⁶	0.357	0.043	0.020	3.00×10 ⁻²
6	22598259	0.229	0.040	0.018	2.19×10 ⁻²	0.282	0.031	0.007	2.04×10 ⁻⁵	0.222	0.071	0.023	1.62×10 ⁻³
6	76164589	0.135	0.088	0.021	4.34×10 ⁻⁵	0.108	0.059	0.011	7.42×10 ⁻⁸	0.137	0.082	0.027	2.59×10 ⁻³
135119089	0.829	-0.052	0.020	8.17×10 ⁻³	0.556	-0.037	0.007	7.51×10 ⁻⁸	0.645	-0.036	0.020	7.14×10 ⁻²
105612736	0.570	0.060	0.016	1.13×10 ⁻⁴	0.273	0.030	0.008	5.49×10 ⁻⁵	0.292	0.051	0.021	1.47×10 ⁻²
118863412	0.230	0.027	0.018	1.20×10 ⁻¹	0.180	0.040	0.009	4.85×10 ⁻⁶	0.160	0.082	0.026	1.35×10 ⁻³
119181794	0.086	0.035	0.026	1.86×10 ⁻¹	0.290	0.034	0.007	5.27×10 ⁻⁶	0.254	0.077	0.022	3.80×10 ⁻⁴
32772734	0.303	-0.048	0.016	3.30×10 ⁻³	0.113	0.051	0.011	1.92×10 ⁻⁶	0.103	-0.084	0.031	7.23×10 ⁻³
10	50485434	0.096	-0.083	0.026	1.43×10 ⁻³	0.053	-0.066	0.015	1.25×10 ⁻⁵	0.057	-0.138	0.041	7.16×10 ⁻⁴
10	80898969	0.715	0.052	0.017	2.72×10 ⁻³	0.490	0.034	0.008	5.38×10 ⁻⁶	0.501	0.062	0.019	1.11×10 ⁻³
11	3890059	0.661	-0.036	0.016	2.27×10 ⁻²	0.489	-0.030	0.007	5.10×10 ⁻⁶	0.484	-0.042	0.019	2.50×10 ⁻²
11	14036189	0.341	0.062	0.016	8.40×10 ⁻⁵	0.376	0.035	0.007	3.45×10 ⁻⁷	0.361	0.081	0.020	3.61×10 ⁻⁵
11	95089882	0.219	-0.022	0.018	2.30×10 ⁻¹	0.670	-0.037	0.007	1.80×10 ⁻⁷	0.603	-0.028	0.019	1.47×10 ⁻¹
11	12886027	0.104	0.064	0.026	1.26×10 ⁻²	0.132	0.049	0.010	2.32×10 ⁻⁶	0.134	0.037	0.028	1.77×10 ⁻¹
12	104492003	0.420	-0.088	0.015	6.35×10 ⁻⁹	0.141	-0.038	0.009	6.71×10 ⁻⁵	0.152	-0.045	0.026	8.69×10 ⁻²
12	110082115	0.446	-0.097	0.015	1.46×10 ⁻¹⁰	0.537	-0.011	0.007	9.59×10 ⁻²	0.431	-0.012	0.019	5.20×10 ⁻¹
12	113196733	0.417	-0.111	0.017	9.88×10 ⁻¹¹	0.312	-0.009	0.008	2.12×10 ⁻¹	0.252	-0.034	0.022	1.16×10 ⁻¹
13	22111521	0.324	0.026	0.016	1.11×10 ⁻¹	0.404	0.030	0.007	8.82×10 ⁻⁶	0.388	0.064	0.019	8.57×10 ⁻⁴
13	74520186	0.195	0.047	0.018	1.08×10 ⁻²	0.357	0.036	0.007	2.81×10 ⁻⁷	0.360	0.024	0.020	2.29×10 ⁻¹
16	15902715	0.194	-0.057	0.019	3.39×10 ⁻³	0.314	-0.035	0.007	1.22×10 ⁻⁶	0.335	-0.044	0.020	2.92×10 ⁻²
18	77156537	0.305	0.054	0.017	1.18×10 ⁻³	0.488	0.038	0.007	1.33×10 ⁻⁷	0.446	0.010	0.019	6.19×10 ⁻¹

**Reviewer #3, Comment #6**

**Data presentation should be improved, as follows: The authors should be more concise in the**
 **results section and consider modifying the manuscript to a letter format. Important findings**
 **seem to be flooded by less interesting results and lengthy descriptions. Specifically, the**
 **following 3 sections should be shortened or shifted to the supplement: "Sex-stratified GWAS**
 **for AF", "Shared allelic effects between Japanese and European populations and fine**
 **mapping/credible set analyses in the trans-ancestry meta-analysis", "Pleiotropic effects and**
 **functional pathways of AF-associated loci"**

We appreciate your helpful advice on the content of our manuscript. As you mentioned, our pre-
 revised manuscript was rather redundant and included less interesting results. Therefore, to clearly
 present our key results, we moved several analyses including those that you pointed out to
 Supplementary Notes, such as the sex-stratified GWAS, comparison of variant effects among
 ancestries, 99% credible set analysis, pleiotropic analysis using colocalization, tissue and gene-sets
 enrichment analysis. Meanwhile, we performed additional important analyses such as a replication
 study for the novel loci identified in the Japanese GWAS and wet experiments to demonstrate the

function of the transcription factor using human induced pluripotent stem cell-derived cardiomyocytes
(iPSCMs). We, therefore, decided to take an original article format to include these results while
keeping the content brief.

**Reviewer #3, Comment #7**

**Figure 2d: Are the P-values corrected for multiple testing? It seems strengths of association**
**are otherwise very modest (-log₁₀ P ranging from 1.5 to 2). Move to supplement?**

Thank you for pointing out the issue. As you mentioned, the ingenuity pathway analysis using the
candidate genes identified by TWAS showed no statistically significant pathways after multiple-testing
correction likely due to the modest number of associated genes using GTEx v7. Therefore, as
Reviewer #1 suggested, we performed a TWAS using GTEx v8, which demonstrated the increase in
the number of associated genes from 86 to 132 and from 74 to 127 in the atrial appendage and left
ventricle, respectively. Although no statistically significant pathways were observed in the ingenuity
pathway analysis using these candidate genes, GO enrichment analysis using FUMA web application
v1.3.7¹⁸ identified several pathways enriched in the associated genes such as cardiac developmental,
conduction, and cardiomyocyte contractile or structure after false discovery rate correction
considering the number of gene-sets tested per category. However, since these pathways were
already suggested in previous studies, the result from this pathway analysis has been moved to
Supplementary Fig. 3.

We added the following description to the revised manuscript.

(Lines 152 – 154 in the main manuscript)

Finally, we performed Gene Ontology enrichment analysis using the candidate genes identified by
TWAS and found several significantly enriched pathways, such as cardiac developmental,
conduction, and cardiomyocyte contractile or structure (Supplementary Fig. 3).

(Lines 452 – 453 in the main manuscript)

Furthermore, we performed Gene Ontology enrichment analysis using FUMA web application v1.3.7⁴²
with false discovery rate correction considering the number of gene sets tested per category.

(Supplementary Fig. 3)

**Reviewer #3, Comment #8**

**Figure 4: Describe what the outcome is in the figure and/or legend.**

We apologize for not adequately describing the outcome. In Figure 4 in the pre-revised manuscript,
 we wanted to highlight the following points. In the Japanese population, PRS showed (1) higher
 performance when applied to the same population as the derivation-GWAS population regardless of
 the sample size in the single derivation-GWAS category, (2) higher performance when it was derived
 from a trans-ancestry meta-GWAS including the Japanese population compared to that derived from
 a meta-GWAS without the Japanese population even when the sample size of derivation-GWAS was
 similar or smaller, and we observed (3) the best performance in the PRS derived from the trans-
 ancestry meta-GWAS including the Japanese population and with the largest sample size.

To make these points clear, we modified the figure and its legend as follows.

(Lines 308 – 316 in the main manuscript)

Therefore, we exhaustively examined AF-PRS using various combinations of GWASs and multiple
 parameters to maximize the predictive performance of AF-PRS; AF-PRS achieved (1) a higher
 performance when applied to the same population as the derivation-GWAS population regardless of
 the sample size in the single derivation-GWAS category, (2) a higher performance when it was
 derived from a trans-ancestry meta-GWAS including the Japanese population compared to that
 derived from a meta-GWAS in a non-Japanese population even with a similar or smaller sample size
 of derivation-GWAS, and (3) the best performance when it was derived from the trans-ancestry meta-
 GWAS including the Japanese population and with the largest sample size.

(Fig. 5)

**Fig. 5 | The performance of PRS (Nagelkerke's pseudo R^2) in the Japanese test cohort (2,953**
 **cases and 21,194 controls).** The results of three PRS models derived from a single GWAS are
 shown on the left panel. The results of four PRS models derived from a meta-GWAS are shown on
 the right panel. The distribution of pseudo R^2 was estimated from 5×10^4 times bootstrapping. **The**
 **box plot center line represents the median, the bounds represent the first and third quartile, and the**
 **whiskers reach to 1.5 times the interquartile range.**

**Reviewer #3, Comment #9**

**Figure 5b as represented here is not easy to understand. If it adds to panel A, then please**
 **describe further and clarify visually. If it does not add to panel A, simply remove or move to**
 **supplement.**

We apologize for presenting such a confusing figure. In the figure in our pre-revised manuscript, we
 constructed a linear regression model of the onset age of AF including PRS as a dichotomous
 variable (individual with high PRS [the top 1%, 5%, 10%, and 20%] vs. those with the remaining
 PRS]. Then, we showed the coefficients of PRS in the regression model, i.e., the difference in the
 onset age of AF between them, as well as its confidence intervals and P values. Since this figure
 might be confusing, we moved it to the supplementary figures (Supplementary Fig. 7a) and modified
 Figure 6a.

(Lines 546 – 551 in the main manuscript)

To assess the association between AF-PRS and age at AF onset, we extracted AF case samples with
 available data on age at AF onset (n = 7,458, the median age of AF onset was 63 years of age [IQR
 56 – 71]), and constructed a linear regression model of age at AF onset including AF-PRS as a

dichotomous variable (individual with high PRS [the top 1%, 5%, 10%, and 20%] vs. those with the
 remaining PRS) to estimate the difference in the age of AF onset between them adjusted by sex and
 the top 20 PCs.

 (Fig. 6a)

 **Fig. 6 | Impact of AF-PRS on relevant phenotypes and long-term mortality. a,** Association
 analysis between AF-PRS and onset age of AF. The onset age of AF is shown based on AF-PRS
 quintiles. The center line of the box plot represents the median, the bounds represent the first and
 third quartile, the whiskers reach to 1.5 times the interquartile range.

 (Supplementary Fig. 7a)

 **Supplementary Fig. 7 | Impact of AF-PRS on the age of AF onset and long-term mortality. a,**
 Difference in the age of AF onset between individuals with high PRS (the top 1%, 5%, 10%, and 20%)
 and those with the remaining PRS. We constructed linear regression models by adjusting for sex and

the top 20 PCs. Each point and error bar represent the estimated beta and 95% CI obtained from the
linear regression models.

**Reviewer #3, Comment #10**

**D. Appropriate use of statistics and treatment of uncertainties**

**No further comment**

**E. Conclusions: robustness, validity, reliability**

**The discussion section is rather short compared to the results section. I suggest shortening**
**the results section (or modifying to a letter format for conciseness)**

**No further comment**

Thank you for your advice to improve our manuscript. To present our results clearly and concisely, we
shortened the result section and moved several analyses to Supplementary Notes as we mentioned
in the response to your Comment #6. Meanwhile, we performed a replication study for the novel loci
identified in the Japanese GWAS and the functional analyses for the transcription factor using human
iPSCMs. Thus, although the content has increased, we have attempted to keep it concise and
improve clarity and overall structure.

**Reviewer #3, Comment #11**

**F. Suggested improvements: experiments, data for possible revision**

**No further comment**

**G. References: appropriate credit to previous work?**

**Appropriate.**

**H. Clarity and context: lucidity of abstract/summary, appropriateness of abstract, introduction**
**and conclusions**

**Good abstract. Suggest moving supplementary figure 1 (Overview of the study design) to the**
**main text. Thank you ##**

Thank you for this suggestion. Accordingly, we moved Supplementary Fig. 1, which contained the
summary of our study design, to the main figure 1. We think this will allow the readers to see the
overall study design first and will help them understand our research better.

We also modified this figure in the revised manuscript.

(Fig. 1)

Fig. 1 | Overview of the study design. Flowchart of the study, which encompasses the Japanese GWAS with the BioBank Japan 1st cohort (9,826 AF cases and 140,446 controls), a replication study using the BioBank Japan 2nd cohort (4,602 cases and 44,075 controls), a trans-ancestry meta-analysis with large-scale European GWASs (77,690 cases and 1,167,040 controls in total), and the downstream analysis. AF, atrial fibrillation; GWAS, genome-wide association study; iPSCMs, induced pluripotent stem cell-derived cardiomyocytes; MR, Mendelian randomization; PRS, polygenic risk score; TWAS, transcriptome-wide association study.

**References**

- 1. Giambartolomei, C. *et al.* Bayesian test for colocalisation between pairs of genetic association
studies using summary statistics. *PLoS Genet* **10**, e1004383 (2014).
- 2. Wightman, D.P. *et al.* A genome-wide association study with 1,126,563 individuals identifies new
risk loci for Alzheimer's disease. *Nat Genet* **53**, 1276-1282 (2021).
- 3. Roselli, C. *et al.* Multi-ethnic genome-wide association study for atrial fibrillation. *Nat Genet* **50**, 1225-
1233 (2018).
- 4. Nasser, J. *et al.* Genome-wide enhancer maps link risk variants to disease genes. *Nature* **593**, 238-243
(2021).
- 5. Das, S. *et al.* Next-generation genotype imputation service and methods. *Nat Genet* **48**, 1284-1287
(2016).
- 6. Nielsen, J.B. *et al.* Biobank-driven genomic discovery yields new insight into atrial fibrillation
biology. *Nat Genet* **50**, 1234-1239 (2018).
- 7. Christophersen, I.E. *et al.* Large-scale analyses of common and rare variants identify 12 new loci
associated with atrial fibrillation. *Nat Genet* **49**, 946-952 (2017).
- 8. Larsson, S.C., Michaelsson, K. & Burgess, S. IGF-1 and cardiometabolic diseases: a Mendelian
randomisation study. *Diabetologia* **63**, 1775-1782 (2020).
- 9. Shah, S. *et al.* Genome-wide association and Mendelian randomisation analysis provide insights into
the pathogenesis of heart failure. *Nat Commun* **11**, 163 (2020).
- 10. Tadros, R. *et al.* Shared genetic pathways contribute to risk of hypertrophic and dilated
cardiomyopathies with opposite directions of effect. *Nat Genet* **53**, 128-134 (2021).
- 11. Consortium, E.P. *et al.* Expanded encyclopaedias of DNA elements in the human and mouse
genomes. *Nature* **583**, 699-710 (2020).
- 12. Surendran, P. *et al.* Discovery of rare variants associated with blood pressure regulation through
meta-analysis of 1.3 million individuals. *Nat Genet* **52**, 1314-1332 (2020).
- 13. Mosley, J.D. *et al.* Predictive Accuracy of a Polygenic Risk Score Compared With a Clinical Risk Score
for Incident Coronary Heart Disease. *JAMA* **323**, 627-635 (2020).
- 14. Elliott, J. *et al.* Predictive Accuracy of a Polygenic Risk Score-Enhanced Prediction Model vs a Clinical
Risk Score for Coronary Artery Disease. *JAMA* **323**, 636-645 (2020).
- 15. Prive, F., Arbel, J. & Vilhjalmsson, B.J. LDpred2: better, faster, stronger. *Bioinformatics* (2020).
- 16. Paciaroni, M. *et al.* Early hemorrhagic transformation of brain infarction: rate, predictive factors, and
influence on clinical outcome: results of a prospective multicenter study. *Stroke* **39**, 2249-56 (2008).

- 17. Kerenyi, L. *et al.* Factors influencing hemorrhagic transformation in ischemic stroke: a
clinicopathological comparison. *Eur J Neurol* **13**, 1251-5 (2006).
- 18. Watanabe, K., Taskesen, E., van Bochoven, A. & Posthuma, D. Functional mapping and annotation of
genetic associations with FUMA. *Nat Commun* **8**, 1826 (2017).

Decision Letter, first revision:

20th June 2022

Dear Kaoru,

Your revised Article "Trans-ancestry genome-wide analysis of atrial fibrillation provides new insights into disease biology and enables polygenic prediction of cardioembolic risk" has been seen by the original referees. You will see from their comments below that, while they find the study substantially improved, Reviewer #1 has requested some additional changes. We remain interested in the possibility of publishing your study in Nature Genetics, but we would like to consider your response to these points in the form of a further revision before we make a final decision on publication.

As before, to guide the scope of the revisions, the editors discuss the referee reports in detail within the team, including with the chief editor, with a view to identifying key priorities that should be addressed in revision. In this case, we ask that you re-run the co-localization analyses using the thresholds suggested by Reviewer #1 and that you contact the curators of the CHIP-Atlas database and ask them to correct the error in the antigen field for SRX4003759. We again hope you will find this prioritized set of referee points to be useful when revising your study. Please do not hesitate to get in touch if you would like to discuss these issues further.

We therefore invite you to revise your manuscript again taking into account all reviewer and editor comments. Please highlight all changes in the manuscript text file. At this stage, we will need you to upload a copy of the manuscript in MS Word .docx or similar editable format.

*2) If you have not done so already please begin to revise your manuscript so that it conforms to our Article format instructions, available [here](http://www.nature.com/ng/authors/article_types/index.html).

*3) Include a revised version of any required Reporting Summary:

[redacted]

We hope to receive your revised manuscript within 4-8 weeks. If you cannot send it within this time, please let us know.

Sincerely,
Kyle

Kyle Vogan, PhD
Senior Editor
Nature Genetics
<https://orcid.org/0000-0001-9565-9665>

Referee expertise:

Referee #1: Genetics, cardiovascular diseases

Referee #2: Genetics, cardiovascular diseases

Referee #3: Genetics, cardiovascular diseases

Reviewers' Comments:

Reviewer #1:
Remarks to the Author:

The authors are to be congratulated for strengthening the manuscript and adding important data. Especially the independent replication lends more credibility to the findings and the novel results on "Analyses of AF-associated transcription factor" offer valuable insight into the transcriptional network controlled by ERG.

However, for the sake of clarity, the authors should provide more information on several items:

- The dataset reachable at <https://chip-atlas.org/view?id=SRX4003759> still has "ERG" as the antigen. This curation error needs to be addressed to ensure replicability of the result.

- The authors apply a coloc cutoff of posterior probability > 0.8 . I assume that the authors use the H4 probability for this. However, a more commonly used cutoff is $\text{coloc H3} + \text{H4} \geq 0.8$ and $\text{H4}/\text{H3} \geq 2$ (see PMID 30824768). The authors should re-run the analysis using these cutoffs.

Reviewer #2:

Remarks to the Author:

I want to thank the authors for their extensive response, which made it very clear which parts of the manuscript were updated. I am impressed by the amount of additional work that has been performed. I do not have additional comments other than two minor suggestions:

Line 85: “in the same direction” I would suggest to add the word “effect”.

Line 255/256: a MR analysis -> an MR analysis

Reviewer #3:

Remarks to the Author:

The authors properly addressed my comments/concerns.

Author Rebuttal, first revision:

Dear Editor and Reviewers,

We appreciate the comments from the editors and the reviewers on our revised manuscript. We are also grateful for your evaluation that our study has been substantially improved, and for the opportunity to further revise and resubmit our manuscript.

According to the comments from the reviewers, we have performed additional analyses and corrected our manuscript. Especially, to resolve Reviewer #1’s first concern, we contacted the administrator of the CHIP-Atlas database again and asked them to correct the information for SRX4003759. We then confirmed that the issue has been resolved; you can now find “ERRg” (whose gene symbol is *ESRRG*), instead of the previously listed “ERG”, on the website <https://chip-atlas.org/view?id=SRX4003759>. We also re-performed the colocalization analyses using the thresholds provided by Reviewer #1 and obtained more meaningful results. We addressed the suggestions from Reviewer #2 to clarify the context and revised our manuscript accordingly. We hope you will find that these revisions have made our manuscript worth considering for publication in *Nature Genetics*. The point-by-point responses to all comments from the reviewers are given below.

Referee expertise:

Referee #1: Genetics, cardiovascular diseases

Referee #2: Genetics, cardiovascular diseases

Referee #3: Genetics, cardiovascular diseases

We thank again all the reviewers for their evaluation and comments on our revised manuscript.

Reviewers' Comments:**Reviewer #1:****Remarks to the Author:**

The authors are to be congratulated for strengthening the manuscript and adding important data. Especially the independent replication lends more credibility to the findings and the novel results on "Analyses of AF-associated transcription factor" offer valuable insight into the transcriptional network controlled by ERRg.

However, for the sake of clarity, the authors should provide more information on several items:

We are grateful for your positive feedback. We also appreciate your additional advice and comments to enhance the clarity and validity of our study. The following is a point-by-point response to each of your comments.

Reviewer #1, Major comment #1

The dataset reachable at <https://chip-atlas.org/view?id=SRX4003759> still has "ERG" as the antigen. This curation error needs to be addressed to ensure replicability of the result.

Thank you for raising this important point. We agree that this must be corrected to ensure replicability of our results. Accordingly, we asked again the curators of the ChIP-Atlas database to confirm and correct the information for SRX4003759 as well as SRX4003760, both of which are ChIP-seq experiments for ERRg antigen. Then, the antigen of these experiments is correctly assigned to "ERRg" now. Please refer to <https://chip-atlas.org/view?id=SRX4003759> and <https://chip-atlas.org/view?id=SRX4003760>.

Reviewer #1, Major comment #2

The authors apply a coloc cutoff of posterior probability > 0.8. I assume that the authors use the H4 probability for this. However, a more commonly used cutoff is $\text{coloc H3} + \text{H4} \geq 0.8$ and $\text{H4}/\text{H3} \geq 2$ (see PMID 30824768). The authors should re-run the analysis using these cutoffs.

We appreciate your constructive suggestion. While we previously defined a significant colocalization as a $\text{coloc H4} > 0.8$ by referring to a previous study¹, we applied the thresholds of $\text{coloc H3} + \text{H4} \geq 0.8$ and $\text{H4}/\text{H3} \geq 2$ in this revised manuscript as per your recommendation². Then, we found the number of colocalizations that met the thresholds increased from 56 to 76 pairs, resulting in showing the relationship between atrial fibrillation and related traits more clearly.

The revised text, figure and table are shown below.

(Lines 94 – 98 in Supplementary Notes)

, and we identified 29 AF-associated loci that showed significant colocalization with QTL (Supplementary Fig. 16). Of 76 pairs of AF-associated loci and QTL with significant colocalization, 22.4% (17/76) were kidney-related traits, 15.8% (12/76) were blood pressure-related, 13.2% (10/76) were metabolic traits, and 13.2% (10/76) were anthropometric traits (Supplementary Table 18).

(Lines 186 – 187 in Supplementary Notes)

The thresholds of $\text{coloc H3} + \text{H4} \geq 0.8$ and $\text{H4}/\text{H3} \geq 2$ were applied for significant colocalization by referring to a prior study²⁷.

(Supplementary Fig. 16)

Supplementary Fig. 16 | Colocalization of quantitative traits with AF-associated signals. Heat-map representation of the approximate Bayes factor posterior probability of AF and quantitative traits to share a common causal variant at 150 AF-associated loci (coloc H3 + H4 represented on a white-red color scale and H4/H3 < 2 in grey). The rows show 36 quantitative traits highlighted in each color of the 9 categories, and the columns show the chromosome and physical position of the lead variant in AF-associated loci. AF, atrial fibrillation; QTL, quantitative trait loci.

[Supplementary Table 18 (partially extracted)]

Supplementary Table 18 | Colocalization of AF-associated loci with QTL for clinical measurements

Lead SNP			Locus		Phenotype	H3 + H4	H4 / H3
CHR	POS	rsID	Start	End			
Novel loci							
1	16199051	rs9782984	15699051	16699051	AST	0.979	10.333
5	139703286	rs17118812	139032339	140214751	CK	1.000	15.728
					eGFR	0.995	7.383
					LDL-C	0.898	6.701
					sCr	0.901	7.548
6	76164589	rs12209223	75664589	77067954	Height	1.000	87.036
11	3890059	rs7126870	3389383	4390223	HbA1c	0.889	30.452
12	12886027	rs10845620	12386027	13386027	HDL-C	0.994	2968.740
12	113196733	rs11614295	112696733	114024987	BUN	0.898	4.671
					DBP	0.928	7.340
					MAP	0.932	7.788
					SBP	0.930	7.490
					UA	0.933	7.909
18	77156537	rs8096658	76656537	77656537	BUN	1.000	1019.110
					eGFR	1.000	62.190
					sCr	1.000	347.176
					UA	0.841	263.956
19	48142746	rs11881441	47628418	48670757	BMI	0.812	19.429
					Height	0.963	9.800
22	21999229	rs5754508	21499229	22499229	GGT	0.934	8.559
					HDL-C	0.998	8.947
					RBC	0.925	8.035
					TP	0.908	7.043
Known loci							
1	10796866	rs880315	10280727	11302468	AST	0.964	383.472
					DBP	1.000	1166.800
					K	0.999	882.599
					MAP	1.000	1281.450
					Na	0.980	222.464
					PP	1.000	1179.540
2	61676940	rs2694635	61071295	62306999	SBP	1.000	1335.590
2	65284231	rs2540949	64735333	65886462	WBC	0.977	6.179
2	148799710	rs12992231	148292665	149306433	PP	0.952	20.596
2	148799710	rs12992231	148292665	149306433	BUN	0.912	4.634
3	12841804	rs7650482	12259080	12841804	CK	1.000	200.285
3	135814009	rs1278493	135314009	137000733	TBil	1.000	2.127

Reviewer #2:**Remarks to the Author:**

I want to thank the authors for their extensive response, which made it very clear which parts of the manuscript were updated. I am impressed by the amount of additional work that has been performed. I do not have additional comments other than two minor suggestions:

We are grateful for your positive comment. We also appreciate your additional suggestions to improve our manuscript. The following corrections have been made in our manuscript accordingly.

Reviewer #2, Comment #1

Line 85: "in the same direction" I would suggest to add the word "effect".

Thank you for this important suggestion. As you pointed out, "in the same direction" alone may not convey our intent correctly, and we agree that adding "effect" makes it more accurate. Accordingly, we have revised the text as follows.

(Lines 85 – 86 in the main manuscript)

Observations with all of the lead variants were successfully replicated with nominal associations ($P < 0.05$) in the same effect direction (Supplementary Table 2).

Reviewer #2, Comment #2

Line 255/256: a MR analysis -> an MR analysis

We apologize for the mistake. We have corrected this accordingly.

(Lines 256 – 257 in the main manuscript)

Therefore, we performed an MR analysis to thoroughly investigate the causality of quantitative traits.

Reviewer #3:**Remarks to the Author:**

The authors properly addressed my comments/concerns.

We sincerely thank the reviewer for having provided valuable and constructive advice, which was of great help to improve our manuscript.

Reference

1. Wightman, D.P. *et al.* A genome-wide association study with 1,126,563 individuals identifies new risk loci for Alzheimer's disease. *Nat Genet* **53**, 1276-1282 (2021).
2. Li, Y.I., Wong, G., Humphrey, J. & Raj, T. Prioritizing Parkinson's disease genes using population-scale transcriptomic data. *Nat Commun* **10**, 994 (2019).

Decision Letter, second revision:
--

30th September 2022

Dear Kaoru,

Thank you for submitting your revised manuscript "Trans-ancestry genome-wide analysis of atrial fibrillation provides new insights into disease biology and enables polygenic prediction of cardioembolic risk" (NG-A58386R1). In light of the changes made in response to the referees' comments, we will be happy in principle to publish your study in Nature Genetics as an Article pending final revisions to comply with our editorial and formatting guidelines.

We are now performing detailed checks on your paper and we will send you a checklist detailing our editorial and formatting requirements soon. Please do not upload the final materials or make any revisions until you receive this additional information from us.

Thank you again for your interest in Nature Genetics. Please do not hesitate to contact me if you have any questions.

Sincerely,
Kyle

Kyle Vogan, PhD
Senior Editor
Nature Genetics
<https://orcid.org/0000-0001-9565-9665>

Final Decision Letter: